# Closing the Gap Between the Upper Bound and the Lower Bound of Adam's Iteration Complexity

**Bohan Wang**[1][*], **Jingwen Fu**[2][*], **Huishuai Zhang**[3][†], **Nanning Zheng**[2][†], **Wei Chen**[4][†]

bhwangfy@gmail.com
fu1371252069@stu.xjtu.edu.cn
huzhang@microsoft.com
nnzheng@mail.xjtu.edu.cn
chenwei2022@ict.ac.cn

[1]University of Science and Technology of China, [2]Xi'an Jiaotong University,
[3]Microsoft Research, [4]Institute of Computing Technology, Chinese Academy of Sciences

## Abstract

Recently, Arjevani et al. [1] establish a lower bound of iteration complexity for the first-order optimization under an $L$-smooth condition and a bounded noise variance assumption. However, a thorough review of existing literature on Adam's convergence reveals a noticeable gap: none of them meet the above lower bound. In this paper, we close the gap by deriving a new convergence guarantee of Adam, with only an $L$-smooth condition and a bounded noise variance assumption. Our results remain valid across a broad spectrum of hyperparameters. Especially with properly chosen hyperparameters, we derive an upper bound of iteration complexity of Adam and show that it meets the lower bound for first-order optimizers. To the best of our knowledge, this is the first to establish such a tight upper bound for Adam's convergence. Our proof utilizes novel techniques to handle the entanglement between momentum and adaptive learning rate and to convert the first-order term in the Descent Lemma to the gradient norm, which may be of independent interest.

## 1 Introduction

First-order optimizers, also known as gradient-based methods, make use of gradient (first-order derivative) information to find the minimum of a function. They have become a cornerstone of many machine learning algorithms due to the efficiency as only gradient informaiton is required, and the flexibility as gradients can be easily computed for any function represented as directed acyclic computational graph via auto-differentiation [2, 25].

Therefore, it is fundamental to theoretically understand the properties of these first-order methods. Recently, Arjevani et al. [1] establish a lower bound on the iteration complexity of stochastic first-order methods. Formally, for a well-studied setting where the objective is $L$-smooth and a stochastic oracle can query the gradient unbiasly with bounded variance (see Assumption 1 and 2), any stochastic first-order algorithm requires at least $\varepsilon^{-4}$ queries (in the worst case) to find an $\varepsilon$-stationary point, i.e., a point with gradient norm at most $\varepsilon$. Arjevani et al. [1] further show that the above lower bound is tight as it matches the existing upper bound of iteration complexity of SGD [15].

On the other hand, among first-order optimizers, Adam [20] becomes dominant in training state-of-the-art machine learning models [3, 18, 4, 11]. Compared to vanilla stochastic gradient descent (SGD), Adam consists of two more key components: (i) momentum to accumulate historical gradient

---

[*]Equal Contribution
[†]Corresponding Authors

37th Conference on Neural Information Processing Systems (NeurIPS 2023).

information and (ii) adaptive learning rate to rectify coordinate-wise step sizes. The psedo-code of Adam is given as Algorithm 1. While the sophisticated design of Adam enables its empirical superiority, it brings great challenges for the theoretical analysis. After examining a series of theoretical works on the upper bound of iteration complexity of Adam [33, 9, 10, 36, 16, 27, 34], we find that none of them match the lower bound for first-order optimizers: they not only consume more queries than the lower bound to reach $\varepsilon$-stationary iterations but also requires additional assumptions (see Section 3 for a detailed discussion).

This theoretical mismatch becomes even more unnatural given the great empirical advantage of Adam over SGD, which incites us to think:

*Is the gap between the upper and lower bounds for Adam a result of the inherent complexity induced by Adam's design, or could it be attributed to the proof techniques not being sharp enough?*

This paper answers the above question, validating the latter hypothesis, by establishing a new upper bound on iteration complexity of Adam for a wide range of hyperparameters that cover typical choices. Specifically, our contribution can be summarized as follows:

- We examine existing works that analyze the iteration complexity of Adam, and find that none of them meets the lower bound of first-order optimization algorithms;

- We derive a new convergence guarantee of Adam with only assuming $L$-smooth condition and bounded variance assumption (Theorem 1), which holds for a wide range of hyperparameters covering typical choices;

- With chosen hyperparameters, we further tighten Theorem 1 and show that the upper bound on the iteration complexity of Adam meets the lower bound, closing the gap (Theorem 2). Our upper bound is tighter than existing results by a logarithmic factor, in spite of weaker assumption.

To the best of our knowledge, this work provides the first upper bound on the iteration complexity of Adam without additional assumptions other than $L$-smooth condition and bounded variance assumption. It is also the first upper bound matching the lower bound of first-order optimizers.

**Organization of this paper.** The rest of the paper is organized as follows: in Section 2, we first present the notations and settup of analysis in this paper ; in Section 3, we revisit the existing works on the iteration complexity of Adam; in Section 4, we present a convergence analysis of Adam with general hyperparameters (Theorem 1); in Section 5, we tighten Theorem 1 with a chosen hyperparameter, and derive an upper bound of Adam's iteration complexity which meets the lower bound; in Section 6, we discuss the limitation of our results.

## 2    Preliminary

The Adam algorithm is restated in Agorithm 1 for convenient reference. Note that compared to the orignal version of Adam in Kingma and Ba [20], the bias-correction terms are omitted to simplify the analysis, and our analysis can be immediately extended to the original version of Adam because the effect of bias-correction term decays exponentially. Also, in the original version of Adam, the adaptive learning rate is $\frac{\eta}{\sqrt{\boldsymbol{\nu}_t}+\lambda \mathbb{1}_d}$ instead of $\frac{\eta}{\sqrt{\boldsymbol{\nu}_t}}$. However, our setting is more challenging and our result can be easily extend to the original version of Adam, since the $\lambda$ term makes the adaptive learning rate upper bounded and eases the analysis.

---

**Algorithm 1** Adam

---

**Input:** Stochastic oracle $\boldsymbol{O}$, learning rate $\eta > 0$, initial point $\boldsymbol{w}_1 \in \mathbb{R}^d$, initial conditioner $\boldsymbol{\nu}_0 \in \mathbb{R}^+$, initial momentum $\boldsymbol{m}_0$, momentum parameter $\beta_1$, conditioner parameter $\beta_2$, number of epoch $T$

1: Sample $r \sim \mathrm{Unif}\{1, \cdots, T\}$
2: **For** $t = 1 \rightarrow T$:
3:      Generate a random $z_t$, and query stochastic oracle $\boldsymbol{g}_t = \boldsymbol{O}_f(\boldsymbol{w}_t, z_t)$
4:      Calculate $\boldsymbol{\nu}_t = \beta_2 \boldsymbol{\nu}_{t-1} + (1 - \beta_2) \boldsymbol{g}_t^{\odot 2}$
5:      Calculate $\boldsymbol{m}_t = \beta_1 \boldsymbol{m}_{t-1} + (1 - \beta_1) \boldsymbol{g}_t$
6:      Update $\boldsymbol{w}_{t+1} = \boldsymbol{w}_t - \eta \frac{1}{\sqrt{\boldsymbol{\nu}_t}} \odot \boldsymbol{m}_t$
7: **EndFor**

**Output:** $\boldsymbol{w}_r$

---

**Notations.** For $a, b \in \mathbb{Z}^{\geq 0}$ and $a \leq b$, denote $[a, b] = \{a, a+1, \cdots, b-1, b\}$. For any two vectors $\boldsymbol{w}, \boldsymbol{v} \in \mathbb{R}^d$, denote $\boldsymbol{w} \odot \boldsymbol{v}$ as the Hadamard product (i.e., coordinate-wise multiplication) between $\boldsymbol{w}$ and $\boldsymbol{v}$. When analyzing Adam, we denote the true gradient at iteration $t$ as $\boldsymbol{G}_t = \nabla f(\boldsymbol{w}_t)$, and the sigma algebra before iteration $t$ as $\mathcal{F}_t = \sigma(\boldsymbol{g}_1, \cdots, \boldsymbol{g}_{t-1})$. We denote conditional expectation as $\mathbb{E}^{|\mathcal{F}_t}[*] = \mathbb{E}[*|\mathcal{F}_t]$. We also use asymptotic notations $o, \mathcal{O}, \Omega$, and $\Theta$, where $h_2(x) = \boldsymbol{o}_{x \rightarrow x_0}(h_1(x))$ means that $\lim_{x \rightarrow x_0} \frac{h_2(x)}{h_1(x)} = 0$ (when the context is clear, we abbreviate $x \rightarrow x_0$ and only use $\boldsymbol{o}(h_1(x))$); $h_2(x) = \mathcal{O}(h_1(x))$ means that there exists constant $\gamma$ independent of $x$ such that $h_2(x) \leq \gamma h_1(x)$; $h_2(x) = \Omega(h_1(x))$ means that $h_1(x) = \mathcal{O}(h_2(x))$; and $h_2(x) = \Theta(h_1(x))$ means that $h_2(x) = \mathcal{O}(h_1(x))$ and $h_2(x) = \Omega(h_1(x))$.

**Objective function.** In this paper, we consider solving the following optimization problem: $\min_{\boldsymbol{w} \in \mathbb{R}^d} f(\boldsymbol{w})$. We make the following assumption on the objective function $f$.

**Assumption 1** (On objective function). *We assume $f$ to be non-negative. We further assume that $f$ satisfies $L$-smooth condition, i.e., $f$ is differentiable, and the gradient of $f$ is $L$-Lipschitz.*

We denote the set of all objective functions satisfying Assumption 1 as $\mathcal{F}(L)$.

**Stochastic oracle.** As $f$ is differentiable, we can utilize the gradient of $f$ (i.e., $\nabla f$) to solve the above optimization problem. However, the $\nabla f$ is usually expensive to compute. Instead, we query a stochastic estimation of $\nabla f$ through a stochastic oracle $\boldsymbol{O}$. Specifically, the stochastic oracle $\boldsymbol{O}$ consists of a distribution $\mathcal{P}$ over a measurable space $\mathcal{Z}$ and a mapping $\boldsymbol{O}_f : \mathbb{R}^d \times \mathcal{Z} \rightarrow \mathbb{R}^d$. We make the following asssumption on $\boldsymbol{O}$.

**Assumption 2** (On stochastic oracle). *We assume that $\boldsymbol{O}$ is unbiased, i.e., $\forall \boldsymbol{w} \in \mathbb{R}^d$, $\mathbb{E}_{z \sim \mathcal{P}} \boldsymbol{O}_f(\boldsymbol{w}, z) = \nabla f(\boldsymbol{w})$. We further assume $\boldsymbol{O}$ has bounded variance, i.e., $\forall \boldsymbol{w} \in \mathbb{R}^d$, $\mathbb{E}_{z \sim \mathcal{P}}[\|\boldsymbol{O}_f(\boldsymbol{w}, z) - \nabla f(\boldsymbol{w})\|^2] \leq \sigma^2$.*

We denote the set of all stochastic oracles satisfying Assumption 2 with variance bound $\sigma^2$ as $\mathfrak{O}(\sigma^2)$.

**Algorithm.** Adam belongs to first-order optimization algorithms, which is defined as follows:

**Definition 1** (First-order optimization algorithm). *An algorithm $\boldsymbol{A}$ is called a first-order optimization algorithm, if it takes an input $\boldsymbol{w}_1$ and hyperparameter $\theta$, and produces a sequence of parameters as follows: first sample a random seed $r$ from some distribution $\mathcal{P}_r$*, set $\boldsymbol{w}_1^{\boldsymbol{A}(\theta)} = \boldsymbol{w}_1$ and then update the parameters as*

$$\boldsymbol{w}_{t+1}^{\boldsymbol{A}(\theta)} = \boldsymbol{A}_\theta^t(r, \boldsymbol{w}_1^{\boldsymbol{A}(\theta)}, \boldsymbol{O}_f(\boldsymbol{w}_1^{\boldsymbol{A}(\theta)}, z_1), \cdots, \boldsymbol{O}_f(\boldsymbol{w}_t^{\boldsymbol{A}(\theta)}, z_t)),$$

*where $z_1, z_2, \cdots, z_t$ are sampled i.i.d. from $\mathcal{P}$.*

**Iteration complexity.** Denote the set of all first-order optimization algorithms as $\mathcal{A}_{\mathrm{first}}$. We next introduce *iteration complexity* to measure the convergence rate of optimization algorithms.

**Definition 2** (Iteration complexity). *The iteration complexity of first-order optimization algorithm $\boldsymbol{A}$ is defined as*

$$\mathcal{C}_\varepsilon(\boldsymbol{A}, \Delta, L, \sigma^2) = \sup_{\boldsymbol{O} \in \mathfrak{O}(\sigma^2)} \sup_{f \in \mathcal{F}(L)} \sup_{\boldsymbol{w}_1 : f(\boldsymbol{w}_1) = \Delta} \inf_\theta \{T : \mathbb{E}\|\nabla f(\boldsymbol{w}_T^{\boldsymbol{A}(\theta)})\| \leq \varepsilon\}.$$

---

*Such a random seed allows sampling from all iterations to generate the final output of the optimization algorithm. As an example, Algorithm 1 sets $\mathcal{P}_r$ as a uniform distribution over $[T]$.

*Furthermore, the iteration complexity of the family of first-order optimization algorithms $\mathcal{A}_{\mathrm{first}}$ is*

$$\mathcal{C}_\varepsilon(\Delta, L, \sigma^2) = \sup_{\boldsymbol{O} \in \mathfrak{O}(\sigma^2)} \sup_{f \in \mathcal{F}(L)} \sup_{\boldsymbol{w}_1 : f(\boldsymbol{w}_1) = \Delta} \inf_{\boldsymbol{A} \in \mathcal{A}_{\mathrm{first}}} \inf_\theta \{T : \mathbb{E}\|\nabla f(\boldsymbol{w}_T^{\boldsymbol{A}(\theta)})\| \le \varepsilon\}.$$

It should be noticed that the iteration complexity of the family of first-order optimization algorithms is a lower bound of the iteration complexity of a specific first-order optimization algorithm, i.e., $\forall \boldsymbol{A} \in \mathcal{A}_{\mathrm{first}}, \mathcal{C}_\varepsilon(\boldsymbol{A}, \Delta, L, \sigma^2) \ge \mathcal{C}_\varepsilon(\Delta, L, \sigma^2)$.

## 3  Related works: none of existing upper bounds match the lower bound

In this section, we examine existing works that study the iteration complexity of Adam, and defer a discussion of other related works to Appendix A. Specifically, we find that none of them match the lower bound for first-order algorithms provided in [1] (restated as follows).

**Proposition 1** (Theorem 3, [1])**.** $\forall L, \Delta, \sigma^2 > 0$, *we have* $\mathcal{C}_\varepsilon(\Delta, L, \sigma^2) = \Omega(\frac{1}{\varepsilon^4})$.

Note that in the above bound, we omit the dependence of the lower bound over $\Delta$, $L$, and $\sigma^2$, which is a standard practice in existing works (see Cutkosky and Mehta [8], Xie et al. [32], Faw et al. [13] as examples) because the dependence over the accuracy $\varepsilon$ can be used to derive how much additional iterations is required for a smaller target accuracy and is thus of more interest. In this paper, when we say "match the lower bound", we always mean that the upper bound has the same order of $\varepsilon$ as the lower bound.

Generally speaking, existing works on the iteration complexity of Adam can be divided into two categories: they either (i) assume that gradient is universally bounded or (ii) make stronger assumptions on smoothness. Below we respectively explain how these two categories of works do not match the lower bound in [1].

The first line of works, including Zaheer et al. [33], De et al. [9], Défossez et al. [10], Zou et al. [36], Guo et al. [16], assume that the gradient norm of $f$ is universally bounded, i.e., $\|\nabla f(\boldsymbol{w})\| \le G$, $\forall \boldsymbol{w} \in \mathbb{R}^d$. In other words, what they consider is another iteration complexity defined as follows:

$$\mathcal{C}_\varepsilon(\boldsymbol{A}, \Delta, L, \sigma^2, G) \triangleq \sup_{\boldsymbol{O} \in \mathfrak{O}(\sigma^2)} \sup_{f \in \mathcal{F}(L), \|\nabla f\| \le G} \sup_{\boldsymbol{w}_1 : f(\boldsymbol{w}_1) = \Delta} \inf_\theta \{T : \mathbb{E}\|\nabla f(\boldsymbol{w}_T^{\boldsymbol{A}(\theta)})\| \le \varepsilon\}.$$

This line of works do not match the lower bound due to the following two reasons: First of all, the upper bound they derive is $O(\frac{\log 1/\varepsilon}{\varepsilon^4})$, which has an additional $\log 1/\varepsilon$ factor more than the lower bound; secondly, the bound they derive is for $\mathcal{C}_\varepsilon(\boldsymbol{A}, \Delta, L, \sigma^2, G)$. Note that $\mathcal{F}(L) \cap \{f : \|\nabla f\| \le G\}$ is a proper subset of $\mathcal{F}(L)$ for any $G$, where a simple example in $\mathcal{F}(L)$ but without bounded gradient is the quadratic function $f(x) = \|x\|^2$. Therefore, we have that

$$\mathcal{C}_\varepsilon(\boldsymbol{A}, \Delta, L, \sigma^2) \ge \mathcal{C}_\varepsilon(\boldsymbol{A}, \Delta, L, \sigma^2, G), \quad \forall G \ge 0, \tag{1}$$

and thus the upper bound on $\mathcal{C}_\varepsilon(\boldsymbol{A}, \Delta, L, \sigma^2, G)$ does not apply to $\mathcal{C}_\varepsilon(\boldsymbol{A}, \Delta, L, \sigma^2)$. Moreover, their upper bound of $\mathcal{C}_\varepsilon(\boldsymbol{A}, \Delta, L, \sigma^2, G)$ tends to $\infty$ as $G \to \infty$, which indicates that if following their analysis, the upper bound of $\mathcal{C}_\varepsilon(\boldsymbol{A}, \Delta, L, \sigma^2)$ would be infinity based on Eq. (1).

The second line of works [27, 34, 30] additionally assume a mean-squared smoothness property besides Assumption 1 and 2, i.e., $\mathbb{E}_{z \sim \mathcal{P}} \|\boldsymbol{O}_f(\boldsymbol{w}, z) - \boldsymbol{O}_f(\boldsymbol{v}, z)\|^2 \le L\|\boldsymbol{w} - \boldsymbol{v}\|^2$. Denote $\tilde{\mathfrak{O}}(\sigma^2, L) \triangleq \{\boldsymbol{O} : \mathbb{E}_{z \sim \mathcal{P}} \|\boldsymbol{O}_f(\boldsymbol{w}, z) - \boldsymbol{O}_f(\boldsymbol{v}, z)\|^2 \le L\|\boldsymbol{w} - \boldsymbol{v}\|^2, \forall \boldsymbol{w}, \boldsymbol{v} \in \mathbb{R}^d\} \cap \mathfrak{O}(\sigma^2)$. The iteration complexity that they consider is defined as follows:

$$\tilde{\mathcal{C}}_\varepsilon(\boldsymbol{A}, \Delta, L, \sigma^2) = \sup_{\boldsymbol{O} \in \tilde{\mathfrak{O}}(\sigma^2, L)} \sup_{f \in \mathcal{F}(L)} \sup_{\boldsymbol{w}_1 : f(\boldsymbol{w}_1) = \Delta} \inf_\theta \{T : \mathbb{E}\|\nabla f(\boldsymbol{w}_T^{\boldsymbol{A}(\theta)})\| \le \varepsilon\}.$$

The rate derived in [27, 34, 30] is $O(\frac{\log 1/\varepsilon}{\varepsilon^6})$, which is derived by minimizing the upper bounds in [27, 34, 30] with respect to the hyperparameter of adaptive learning rate $\beta_2$. According to Arjevani et al. [1], the lower bound of iteration complexity of $\tilde{\mathcal{C}}_\varepsilon(\boldsymbol{A}, \Delta, L, \sigma^2)$ is $\Omega(\frac{1}{\varepsilon^3})$ and smaller than the original lower bound $\Omega(\frac{1}{\varepsilon^4})$, resulting in an even larger gap between the upper and lower bounds.

Recently, there is a concurrent work [21] which does not require bounded gradient assumption and mean-squared smoothness property but poses a stronger assumption on the stochastic oracle: the set of stochastic oracles they consider is $\tilde{\mathfrak{O}} = \{ \boldsymbol{O} : \forall \boldsymbol{w} \in \mathbb{R}^d, \mathbb{E}_{z \sim \mathcal{P}} \boldsymbol{O}_f(\boldsymbol{w}, z) = \nabla f(\boldsymbol{w}), \mathbb{P} \left( \| \boldsymbol{O}_f(\boldsymbol{w}, z) - \nabla f(\boldsymbol{w}) \|^2 \leq \sigma^2 \right) = 1 \}$. $\tilde{\mathfrak{O}}$ is a proper subset of $\mathfrak{O}$ because a simple example is that $\boldsymbol{O}_f(\boldsymbol{w}, z) = \nabla f(\boldsymbol{w}) + z$ where $z$ is a standard gaussian variable. Therefore, their result does not provide a valid upper bound of $\mathcal{C}_\varepsilon(\boldsymbol{A}, \Delta, L, \sigma^2)$.

# 4 Convergence analysis of Adam with only Assumptions 1 and 2

As discussed in Section 3, existing works on analyzing Adam require additional assumptions besides Assumption 1 and 2. In this section, we provide the first convergence analysis of Adam with only Assumption 1 and 2, which naturally gives an upper bound on the iteration complexity $\mathcal{C}_\varepsilon(\boldsymbol{A}, \Delta, L, \sigma^2)$. In fact, our analysis even holds when the stochastic oracle satisfies the following more general assumption.

**Assumption 3** (Coordinate-wise affine noise variance). *We assume that $\boldsymbol{O}$ is unbiased, i.e., $\forall \boldsymbol{w} \in \mathbb{R}^d$, $\mathbb{E}_{z \sim \mathcal{P}} \boldsymbol{O}_f(\boldsymbol{w}, z) = \nabla f(\boldsymbol{w})$. We further assume $\boldsymbol{O}$ has coordinate-wise affine variance, i.e., $\forall \boldsymbol{w} \in \mathbb{R}^d$ and $\forall i \in [d]$, $\mathbb{E}_{z \sim \mathcal{P}}[|(\boldsymbol{O}_f(\boldsymbol{w}, z))_i|^2] \leq \sigma_0^2 + \sigma_1^2 \partial_i f(\boldsymbol{w})^2$.*

One can easily observe that Assumption 3 is more general than Assumption 2 since Assumption 2 immediately indicates Assumption 3 with $\sigma_0 = \sigma$ and $\sigma_1 = 1$. We consider Assumption 3 not only because it is more general but also because it allows the noise to grow with the norm of the true gradient, which is usually the case in machine learning practice [14, 19].

Our analysis under Assumption 1 and Assumption 3 is then given as follows.

**Theorem 1.** *Let $\boldsymbol{A}$ be by Adam (Algorithm 1) and $\theta = (\eta, \beta_1, \beta_2)$ are the hyperparameters of $\boldsymbol{A}$. Let Assumption 1 and 2 hold. Then, if $0 \leq \beta_1 \leq \sqrt{\beta_2} - 8\sigma_1^2(1 - \beta_2)\beta_2^{-2}$ and $\beta_2 < 1$, we have*

$$
\mathbb{E} \sum_{t=1}^{T} \| \nabla f(\boldsymbol{w}_t) \| \leq \sqrt{ C_2 + 2C_1 \sum_{i=1}^{d} \left( \ln \left( 2(T+1) \sum_{i=1}^{d} \sqrt{\boldsymbol{\nu}_{0,i} + \sigma_0^2} + 24d \frac{\sigma_1^2 C_1}{\sqrt{\beta_2}} \ln d \frac{\sigma_1^2 C_1}{\sqrt{\beta_2}} + \frac{12\sigma_1^2}{\sqrt{\beta_2}} C_2 \right) \right) }
$$
$$
\times \sqrt{ 2(T+1) \sum_{i=1}^{d} \sqrt{\boldsymbol{\nu}_{0,i} + \sigma_0^2} + 24d \frac{\sigma_1^2 C_1}{\sqrt{\beta_2}} \ln d \frac{\sigma_1^2 C_1}{\sqrt{\beta_2}} + \frac{12\sigma_1^2}{\sqrt{\beta_2}} C_2 }. \tag{2}
$$

*where $\boldsymbol{\nu}_{0,i}$ is the $i$-th coordinate of $\boldsymbol{\nu}_0$,*

$$
C_1 = \frac{32 L \eta \left( 1 + \frac{\beta_1}{\sqrt{\beta_2}} \right)^3}{(1 - \beta_2) \left( 1 - \frac{\beta_1}{\sqrt{\beta_2}} \right)^3} + \frac{16 \beta_1^2 \sigma_0 (1 - \beta_1)}{\beta_2 \sqrt{1 - \beta_2} \left( 1 - \frac{\beta_1}{\sqrt{\beta_2}} \right)^3} + \frac{64(1 + \sigma_1^2) \sigma_1^2 L^2 \eta^2 d}{\beta_2^2 \left( 1 - \frac{\beta_1}{\sqrt{\beta_2}} \right)^4 \sigma_0 (1 - \beta_2)^{\frac{3}{2}}},
$$
$$
C_2 = \frac{1 - \frac{\beta_1}{\sqrt{\beta_2}}}{1 - \beta_1} \frac{8}{\eta} f(\boldsymbol{u}_1) + \frac{32}{\beta_2 \left( 1 - \frac{\beta_1}{\sqrt{\beta_2}} \right)^2} \sum_{i=1}^{d} \mathbb{E} \frac{\boldsymbol{G}_{1,i}^2}{\sqrt{\tilde{\boldsymbol{\nu}}_{1,i}}} + 2C_1 \sum_{i=1}^{d} \left( \ln \left( \frac{1}{\sqrt{\beta_2 \boldsymbol{\nu}_{0,i}}} \right) - T \ln \beta_2 \right).
$$

A proof sketch is given in Section 4.2 and the full proof is deferred to Appendix.

The right-hand side in Eq. (2) looks messy at the first glance. We next explain Theorem 1 in detail and make the upper bound's dependence over hyperparameters crystally clear.

## 4.1 Discussion on Theorem 1

**Required assumptions and conditions.** As mentioned previously, Theorem 1 only requires Assumption 1 and 2, which aligns with the setting of the lower bound (Proposition 1). To our best knowledge, this is the first analysis of Adam without additional assumptions.

As for the range of $\beta_1$ and $\beta_2$, one can immediately see that the condition $\beta_1 \leq \sqrt{\beta_2} - 8\sigma_1^2(1 - \beta_2)\beta_2^{-2}$ degenerates to $\beta_1 \leq \sqrt{\beta_2}$ in the bounded gradient case (i.e., $\sigma_1 = 0$), the weakest condition required in existing literature [36]. When $\sigma_1 \neq 0$, such a condition is stronger than $\beta_1 \leq \sqrt{\beta_2}$. We point out that this is not due to technical limitations but instead agrees with existing counterexamples for Adam: Reddi et al. [26], Zhang et al. [34] show that when $\sigma_1 \neq 0$, there exists a counterexample

satisfying Assumption 1 and Assumption 3 and a pair of $(\beta_1, \beta_2)$ with $\beta_1 < \sqrt{\beta_2}$ and Adam with $(\beta_1, \beta_2)$ diverges over such a counterexample.

**Dependence over $\beta_2$, $\eta$, and $T$.** Here we consider the influence of $\beta_2$, $\eta$, and $T$ while fixing $\beta_1$ constant (we will discuss the effect of $\beta_1$ in Section 6). With logarithmic factors ignored and coefficients hidden, $C_1$, $C_2$ and the right-hand-side of Eq. (2) can be rewritten with asymptotic notations as

$$C_1 = \tilde{\mathcal{O}}\left(\frac{1}{\sqrt{1-\beta_2}} + \frac{\eta^2}{\sqrt{(1-\beta_2)^3}}\right), C_2 = \tilde{\mathcal{O}}\left(\frac{1}{\sqrt{1-\beta_2}} + \frac{\eta^2}{\sqrt{(1-\beta_2)^3}} + \frac{1}{\eta} + T\sqrt{1-\beta_2} + \frac{\eta^2}{\sqrt{1-\beta_2}}T\right),$$

$$\mathbb{E}\sum_{t=1}^{T}\|\nabla f(\boldsymbol{w}_t)\| = \tilde{\mathcal{O}}\left(C_1 + C_2 + \sqrt{TC_1} + \sqrt{TC_2}\right),$$

where $\tilde{\mathcal{O}}$ denotes $\mathcal{O}$ with logarithmic terms ignored. Consequently, the dependence of Eq. (2) over $\beta_2, \eta$ and $T$ becomes

$$\mathbb{E}\sum_{t=1}^{T}\|\nabla f(\boldsymbol{w}_t)\| = \tilde{\mathcal{O}}\left(\frac{1}{\sqrt{1-\beta_2}} + \frac{\eta^2}{\sqrt{(1-\beta_2)^3}} + \frac{1}{\eta} + T\sqrt{1-\beta_2} + \frac{\eta^2}{\sqrt{1-\beta_2}}T\right)$$
$$+ \tilde{\mathcal{O}}\left(\frac{\sqrt{T}}{\sqrt[4]{1-\beta_2}} + \frac{\eta\sqrt{T}}{\sqrt[4]{(1-\beta_2)^3}} + \frac{\sqrt{T}}{\sqrt{\eta}} + T\sqrt[4]{1-\beta_2} + \frac{\eta}{\sqrt[4]{1-\beta_2}}T\right).$$

Here we consider two cases: (i). $\beta_2$ and $\eta$ are independent over $T$, and (ii). $\beta_2$ and $\eta$ are dependent over $T$. For case (i), based on the above equation, one can easily observe that the averaged gradient norm $\frac{1}{T}\mathbb{E}\sum_{t=1}^{T}\|\nabla f(\boldsymbol{w}_t)\|$ will converge to the threshold $\mathcal{O}(\frac{\eta^2}{\sqrt{1-\beta_2}} + \sqrt[4]{1-\beta_2} + \frac{\eta}{\sqrt[4]{1-\beta_2}})$ with rate $\mathcal{O}(1/\sqrt{T})$. This aligns with the observation in [27, 34] that Adam will not converge to the stationary point with constant $\beta_2$.

For case (ii), in order to ensure convergence, i.e., $\min_{t\in[T]}\mathbb{E}\|\boldsymbol{G}_t\|_1 \to 0$ as $T \to \infty$, a sufficient condition is that the right-hand-side of the above equation is $\boldsymbol{o}(T)$. Specifically, by choosing $\eta = \Theta(T^{-a})$ and $1 - \beta_2 = \Theta(T^{-b})$, we obtain that

$$\frac{1}{T}\mathbb{E}\sum_{t=1}^{T}\|\nabla f(\boldsymbol{w}_t)\| = \tilde{\mathcal{O}}\left(T^{\frac{b}{2}-1} + T^{-2a+\frac{3b}{2}-1} + T^{a-1} + T^{-\frac{b}{2}} + T^{-2a+\frac{b}{2}}\right)$$
$$+ \tilde{\mathcal{O}}\left(T^{-\frac{1}{2}+\frac{b}{4}} + T^{-\frac{1}{2}-a+\frac{3b}{4}} + T^{-\frac{1}{2}+\frac{a}{2}} + T^{-\frac{b}{4}} + T^{-a+\frac{b}{4}}\right).$$

By simple calculation, we obtain that the right-hand side of the above inequality is $\boldsymbol{o}(1)$ as $T \to \infty$ if and only if $b > 0, 1 > a > 0$ and $b - a < 1$. Moreover, the minimum of the right-hand side of the above inequality is $\tilde{\mathcal{O}}(1/T^{\frac{1}{4}})$, which is achieved at $a = \frac{1}{2}$ and $b = 1$. Such a minimum implies an upper bound of the iteration complexity which at most differs from the lower bound by logarithmic factors as solving $\tilde{\mathcal{O}}(1/T^{\frac{1}{4}}) = \varepsilon$ gives $T = \tilde{\mathcal{O}}(\frac{1}{\varepsilon^4})$. In Theorem 2, we will further remove the logarithmic factor by giving a refined proof when $a = \frac{1}{2}$ and $b = 1$ and close the gap between the upper and lower bounds.

**Dependence over $\lambda$.** Our analysis allows $\lambda = 0$ in the adaptive learning rate $\eta\frac{1}{\sqrt{\boldsymbol{\nu}_t}+\lambda\mathbb{1}_d}$. In contrast, some existing works [16, 21] require non-zero $\lambda$ and their iteration complexity has polynomial dependence over $\frac{1}{\lambda}$, which is less desired as $\lambda$ can be as small as $10^{-8}$ in practice (e.g., in PyTorch's default setting). Furthermore, compared to their setting, our setting is more challenging as non-zero $\lambda$ immediately provides an upper bound of the adaptive learning rate.

### 4.2 Proof Sketch of Theorem 1

In this section, we demonstrate the proof idea of Theorem 1. Generally speaking, our proof is inspired by (i). the construction of the Lyapunov function for SGDM [22] and (ii) the construction of auxiliary function and the conversion from regret bound to gradient bound for AdaGrad [31], but the adaptation of these techniques to Adam is highly non-trivial, as SGDM does not hold an adaptive learning rate, and the adaptive learning rate of AdaGrad is monotonously decreasing. Below we sketch the proof by identifying three key challenges in the proof and provide our solutions respectively.

**Challenge I: Disentangle the stochasticity in stochastic gradient and adaptive learning rate.** For simplicity, let us first consider the case where $\beta_1 = 0$, i.e., where the momentum $\boldsymbol{m}_t$ degenerates to

the stochastic gradient $\boldsymbol{g}_t$. According to the standard descent lemma, we have that

$$\mathbb{E}f(\boldsymbol{w}_{t+1}) \leq f(\boldsymbol{w}_t) + \mathbb{E}\left[\langle \boldsymbol{G}_t, \boldsymbol{w}_{t+1} - \boldsymbol{w}_t \rangle + \frac{L}{2}\|\boldsymbol{w}_{t+1} - \boldsymbol{w}_t\|^2\right]$$

$$\leq \mathbb{E}f(\boldsymbol{w}_t) + \underbrace{\mathbb{E}\left[\left\langle \boldsymbol{G}_t, -\eta\frac{1}{\sqrt{\boldsymbol{\nu}_t}} \odot \boldsymbol{g}_t \right\rangle\right]}_{\text{First Order}} + \underbrace{\frac{L}{2}\eta^2\mathbb{E}\left\|\frac{1}{\sqrt{\boldsymbol{\nu}_t}} \odot \boldsymbol{m}_t\right\|^2}_{\text{Second Order}} \qquad (3)$$

The first challenge arises from bounding the "First Order" term above. To facilitate the understanding of the difficulty, we compare the "First Order" term of Adam to the corresponding "First Order" term of SGD, i.e., $-\eta\mathbb{E}\langle \boldsymbol{G}_t, \boldsymbol{g}_t \rangle$. By directly applying $\mathbb{E}^{|\mathcal{F}_t} g_t = \boldsymbol{G}_t$, we obtain that the "First-Order" term of SGD equals to $-\eta\mathbb{E}\|\boldsymbol{G}_t\|^2$. However, as for Adam, we do not even know what $\mathbb{E}^{|\mathcal{F}_t}\frac{1}{\sqrt{\boldsymbol{\nu}_t}} \odot \boldsymbol{g}_t$ is given that the stochasticity in $\boldsymbol{g}_t$ and $\boldsymbol{\nu}_t$ entangles. A common practice is to use a *surrogate adaptive learning rate* $\widetilde{\boldsymbol{\nu}}_t$ measurable with respect to $\mathcal{F}_t$, to approximate the real adaptive learning rate $\boldsymbol{\nu}_t$. This leads to the following equation:

$$\underbrace{\mathbb{E}\left[\left\langle \boldsymbol{G}_t, -\eta\frac{1}{\sqrt{\boldsymbol{\nu}_t}} \odot \boldsymbol{g}_t \right\rangle\right]}_{\text{First Order}} = \underbrace{\mathbb{E}\left[\left\langle \boldsymbol{G}_t, -\eta\frac{1}{\sqrt{\widetilde{\boldsymbol{\nu}}_t}} \odot \boldsymbol{g}_t \right\rangle\right]}_{\text{First Order Main}} + \underbrace{\mathbb{E}\left[\left\langle \boldsymbol{G}_t, -\eta\left(\frac{1}{\sqrt{\boldsymbol{\nu}_t}} - \frac{1}{\sqrt{\widetilde{\boldsymbol{\nu}}_t}}\right) \odot \boldsymbol{g}_t \right\rangle\right]}_{\text{Error}}.$$

One can immediately see that "First Order Main" terms equals to $\mathbb{E}[\langle \boldsymbol{G}_t, -\eta\frac{1}{\sqrt{\widetilde{\boldsymbol{\nu}}_t}} \odot \boldsymbol{G}_t \rangle] < 0$, but now we need to handle the "Error" term. In existing literature, such a term is mostly bypassed by applying the bounded gradient assumption [10, 36], which, however, we do not assume.

**Solution to Challenge I.** Inspired by recent advance in the analysis of AdaGrad [31], we consider the auxiliary function $\xi_t = \mathbb{E}[\eta\langle \boldsymbol{G}_t, -\frac{1}{\sqrt{\widetilde{\boldsymbol{\nu}}_{t+1}}} \odot \boldsymbol{G}_t \rangle]$, where we choose $\widetilde{\boldsymbol{\nu}}_t = \beta_2\boldsymbol{\nu}_{t-1} + (1 - \beta_2)\sigma_0^2\mathbb{1}_d$. In the following lemma, we show that the error term can be controlled using $\xi_t$, parallel to (Lemma 4. [31]).

**Lemma 1** (Informal version of Lemma 7 with $\beta_1 = 0$)**.** *Let all conditions in Theorem 1 hold. Then,*

$$Error \leq \frac{5}{8}\mathbb{E}\left[\eta\left\langle \boldsymbol{G}_t, -\frac{1}{\sqrt{\widetilde{\boldsymbol{\nu}}_t}} \odot \boldsymbol{G}_t \right\rangle\right] + \mathcal{O}\left(\frac{1}{\sqrt{\beta_2}}\xi_{t-1} - \xi_t\right) + Small\ Error. \qquad (4)$$

In the right-hand-side of inequality (4), one can easily observe that the first term can be controlled by "First Order Main" term, and the third term is as small as the "Second Order" term. However, the second term seems annoying – in the analysis of AdaGrad [31], there is no $1/\sqrt{\beta_2}$ factor, making the corresponding term a telescoping, but this is no longer true due to the existence of the $1/\sqrt{\beta_2}$ factor. We resolve this difficulty by looking at the sum of $\frac{1}{\sqrt{\beta_2}}\xi_{t-1} - \xi_t$ over $t$ from 1 to $T$, which gives $\mathcal{O}((1 - \beta_2)\sum_{t=1}^{T-1}\xi_t)$. By further noticing that $\widetilde{\boldsymbol{\nu}}_{t+1} \geq \beta_2\widetilde{\boldsymbol{\nu}}_t$, we have

$$\sum_{t=1}^{T}\left(\frac{1}{\sqrt{\beta_2}}\xi_{t-1} - \xi_t\right) \leq \mathcal{O}\left((1 - \beta_2)\sum_{t=1}^{T-1}\mathbb{E}\left[\eta\left\langle \boldsymbol{G}_t, -\frac{1}{\sqrt{\widetilde{\boldsymbol{\nu}}_t}} \odot \boldsymbol{G}_t \right\rangle\right]\right).$$

The right-hand-side term can thus be controlled by the "First Order Main" term when $\beta_2$ is close to 1.

**Remark 1.** *Compared to the analysis of AdaGrad in [31], our proof technique has two-fold novelties. First, our auxiliary function has an additional $(1 - \beta_2)\sigma_0^2\mathbb{1}_d$ term, which is necessary for the analysis of Adam as it makes $\widetilde{\boldsymbol{\nu}}_t$ lower bounded from 0 (AdaGrad does not need this, as $\boldsymbol{\nu}_{t-1}$ of AdaGrad itself is lower bounded). Secondly, as discussed above, the "AdaGrad version" of second term in the right-hand-side of inequality (4) is a telescoping, the sum of which can be bounded straightforwardly.*

**Challenge II: Handle the mismatch between stochastic gradient and momentum.** In the analysis above, we assume $\beta_1 = 0$. Additional challenges arise when we move to the case where $\beta_1 \neq 0$. Specifically, following the same routine, the "First Order Main" term now becomes $\mathbb{E}\left[\langle \boldsymbol{G}_t, -\eta\frac{1}{\sqrt{\widetilde{\boldsymbol{\nu}}_t}} \odot \boldsymbol{m}_t \rangle\right]$. It is hard to even estimate whether such a term is negative or not, given that $\boldsymbol{m}_t$ and $\widetilde{\boldsymbol{\nu}}_t$ still has entangled stochasticity, and the conditional expectation of $\boldsymbol{m}_t$ also differs from $\boldsymbol{G}_t$, both due to the existence of historical gradient.

**Solution to Challenge II.** Inspired by the state-of-art analysis of SGDM [22], which leverage the potential function $f(v_t)$ with $v_t = \frac{\boldsymbol{w}_t - \beta \boldsymbol{w}_{t-1}}{1-\beta}$, we propose to use the potential function $f(\boldsymbol{u}_t)$ with $\boldsymbol{u}_t = \frac{\boldsymbol{w}_t - \frac{\beta_1}{\sqrt{\beta_2}} \boldsymbol{w}_{t-1}}{1 - \frac{\beta_1}{\sqrt{\beta_2}}}$. Applying descent lemma to $f(\boldsymbol{u}_t)$, we obtain that

$$\mathbb{E}[f(\boldsymbol{u}_{t+1})] \leq \mathbb{E}f(\boldsymbol{u}_t) + \underbrace{\mathbb{E}\left[\langle \nabla f(\boldsymbol{u}_t), \boldsymbol{u}_{t+1} - \boldsymbol{u}_t \rangle\right]}_{\text{First Order}} + \underbrace{\frac{L}{2}\mathbb{E}\|\boldsymbol{u}_{t+1} - \boldsymbol{u}_t\|^2}_{\text{Second Order}}. \tag{5}$$

We again focus on the "First Order" term, which can be written as

$$
\begin{aligned}
\mathbb{E}\left[\langle \nabla f(\boldsymbol{u}_t), \boldsymbol{u}_{t+1} - \boldsymbol{u}_t \rangle\right] &= \mathbb{E}\left[\left\langle \nabla f(\boldsymbol{u}_t), \frac{\boldsymbol{w}_{t+1} - \boldsymbol{w}_t}{1 - \frac{\beta_1}{\sqrt{\beta_2}}} - \frac{\beta_1}{\sqrt{\beta_2}}\frac{\boldsymbol{w}_t - \boldsymbol{w}_{t-1}}{1 - \frac{\beta_1}{\sqrt{\beta_2}}}\right\rangle\right] \\
&\overset{(*)}{\approx} \mathbb{E}\left[\left\langle \nabla f(\boldsymbol{w}_t), -\frac{\eta}{1 - \frac{\beta_1}{\sqrt{\beta_2}}}\frac{1}{\sqrt{\boldsymbol{\nu}_t}}\odot \boldsymbol{m}_t + \frac{\eta}{1 - \frac{\beta_1}{\sqrt{\beta_2}}}\frac{\beta_1}{\sqrt{\beta_2 \boldsymbol{\nu}_{t-1}}}\odot \boldsymbol{m}_{t-1}\right\rangle\right] \\
&\overset{(\circ)}{\approx} \mathbb{E}\left[\left\langle \nabla f(\boldsymbol{w}_t), -\frac{\eta}{1 - \frac{\beta_1}{\sqrt{\beta_2}}}\frac{1}{\sqrt{\widetilde{\boldsymbol{\nu}}_t}}\odot \boldsymbol{m}_t + \frac{\eta}{1 - \frac{\beta_1}{\sqrt{\beta_2}}}\frac{\beta_1}{\sqrt{\widetilde{\boldsymbol{\nu}}_t}}\odot \boldsymbol{m}_{t-1}\right\rangle\right] \\
&= \mathbb{E}\left[\left\langle \boldsymbol{G}_t, -\frac{\eta(1-\beta_1)}{1 - \frac{\beta_1}{\sqrt{\beta_2}}}\frac{1}{\sqrt{\widetilde{\boldsymbol{\nu}}_t}}\odot \boldsymbol{g}_t\right\rangle\right] = \mathbb{E}\left[\left\langle \boldsymbol{G}_t, -\frac{\eta(1-\beta_1)}{1 - \frac{\beta_1}{\sqrt{\beta_2}}}\frac{1}{\sqrt{\widetilde{\boldsymbol{\nu}}_t}}\odot \boldsymbol{G}_t\right\rangle\right].
\end{aligned}
$$

Here approximate equation $(*)$ is due to Assumption 1 and that $\boldsymbol{w}_t$ is close to $\boldsymbol{u}_t$, and approximate equation $(\circ)$ is due to Lemma 1 and $\widetilde{\boldsymbol{\nu}}_t = \beta_2 \boldsymbol{\nu}_{t-1} + (1-\beta_2)\sigma_0^2 \approx \beta_2 \boldsymbol{\nu}_{t-1}$ (of course, these are informal statements. Please refer to Appendix C for the detailed proof). With the above methodology, we arrive at the following lemma.

**Lemma 2** (Informal Version of Lemma 8). *Let all conditions in Theorem 1 holds. Then,*

$$\mathbb{E}f(\boldsymbol{u}_{t+1}) \leq \mathbb{E}f(\boldsymbol{u}_t) - \Omega\left(\mathbb{E}\left[\eta\left\langle \boldsymbol{G}_t, -\frac{1}{\sqrt{\widetilde{\boldsymbol{\nu}}_t}}\odot \boldsymbol{G}_t\right\rangle\right]\right) + \mathcal{O}\left(\frac{1}{\sqrt{\beta_2}}\xi_{t-1} - \xi_t\right) + \text{Small Error}.$$

Summing the above lemma over $t$ from $1$ to $T$, we obtain

$$\sum_{t=1}^{T}\mathbb{E}\left[\left\|\frac{1}{\sqrt[4]{\widetilde{\boldsymbol{\nu}}_t}}\odot \boldsymbol{G}_t\right\|^2\right] \leq \mathcal{O}(1) + \sum_{l=1}^{d}\mathcal{O}\left(\mathbb{E}\ln\left(\frac{\boldsymbol{\nu}_{t,i}}{\boldsymbol{\nu}_{0,l}}\right) - T\ln\beta_2\right). \tag{6}$$

We then encounter the second challenge.

**Challenge III: Convert Eq. (6) to a bound of gradient norm.** Although we have derived a regret bound, i.e., a bound of $\sum_{t=1}^{T}\mathbb{E}[\|\frac{1}{\sqrt[4]{\widetilde{\boldsymbol{\nu}}_t}}\odot \boldsymbol{G}_t\|^2]$, we need to convert it into a bound of $\mathbb{E}[\|\boldsymbol{G}_t\|^2]$. In existing works [36, 10, 16] which assumes bounded gradient, such a conversion is straightforward because (their version of) $\widetilde{\boldsymbol{\nu}}_t$ is upper bounded. However, we do not assume bounded gradient and $\widetilde{\boldsymbol{\nu}}_t$ can be aribitrarily large, making $\mathbb{E}[\|\frac{1}{\sqrt[4]{\widetilde{\boldsymbol{\nu}}_t}}\odot \boldsymbol{G}_t\|^2]$ arbitrarily small than $\mathbb{E}[\|\boldsymbol{G}_t\|^2]$.

**Solution to Challenge III.** As this part involves coordinate-wise analysis, we define $\boldsymbol{g}_{t,i}, \boldsymbol{G}_{t,i}, \boldsymbol{\nu}_{t,i}$, and $\widetilde{\boldsymbol{\nu}}_{t,i}^1$ respectively as the $l$-th coordinate of $\boldsymbol{g}_t, \boldsymbol{G}_t, \boldsymbol{\nu}_t$, and $\widetilde{\boldsymbol{\nu}}_t^1$. To begin with, note that due to Cauchy's inequality and Hölder's inequality,

$$\left(\mathbb{E}\sum_{t=1}^{T}\|\boldsymbol{G}_t\|\right)^2 \leq \left(\sum_{t=1}^{T}\mathbb{E}\left[\left\|\frac{1}{\sqrt[4]{\widetilde{\boldsymbol{\nu}}_t}}\odot \boldsymbol{G}_t\right\|^2\right]\right)\left(\sum_{t=1}^{T}\mathbb{E}\left[\left\|\sqrt[4]{\widetilde{\boldsymbol{\nu}}_t}\right\|^2\right]\right). \tag{7}$$

Therefore, we only need to derive an upper bound of $\sum_{t=1}^{T}\mathbb{E}[\|\sqrt[4]{\widetilde{\boldsymbol{\nu}}_t}\|^2]$, which is achieved by the following divide-and-conque methodology. Firstly, when $|\boldsymbol{G}_{t,i}| \geq \frac{\sigma_0}{\sigma_1}$, we can show $2\mathbb{E}^{|\mathcal{F}_t}|\boldsymbol{g}_{t,i}|^2 \geq 2|\boldsymbol{G}_{t,i}|^2 \geq \mathbb{E}^{|\mathcal{F}_t}|\boldsymbol{g}_{t,i}|^2$. Then, through a direct calculation, we obtain that

$$\mathbb{E}\left[\frac{|\boldsymbol{G}_{t,i}|^2}{\sqrt{\widetilde{\boldsymbol{\nu}}_{t,i}}}\mathbf{1}_{|G_{t,i}|\geq\frac{\sigma_0}{\sigma_1}}\right] \geq \frac{\sqrt{\beta_2}}{3(1-\beta_2)\sigma_1^2}\mathbb{E}\left[\left(\sqrt{\widetilde{\boldsymbol{\nu}}_{t+1,i}} - \sqrt{\beta_2\widetilde{\boldsymbol{\nu}}_{t,i}}\right)\mathbf{1}_{|G_{t,i}|\geq\frac{\sigma_0}{\sigma_1}}\right],$$

and thus

$$\sum_{t=1}^{T}\mathbb{E}\left[\frac{|\boldsymbol{G}_{t,i}|^2}{\sqrt{\widetilde{\boldsymbol{\nu}}_{t,i}}}\right] \geq \frac{\sqrt{\beta_2}}{3(1-\beta_2)\sigma_1^2}\sum_{t=1}^{T}\mathbb{E}\left[\left(\sqrt{\widetilde{\boldsymbol{\nu}}_{t+1,i}} - \sqrt{\beta_2\widetilde{\boldsymbol{\nu}}_{t,i}}\right)\mathbf{1}_{|G_{t,i}|\geq\frac{\sigma_0}{\sigma_1}}\right].$$

Secondly, when $|\boldsymbol{G}_{t,i}| < \frac{\sigma_0}{\sigma_1}$, define $\{\bar{\boldsymbol{\nu}}_{t,i}\}_{t=0}^{\infty}$ as $\bar{\boldsymbol{\nu}}_{0,l} = \boldsymbol{\nu}_{0,l}$, $\bar{\boldsymbol{\nu}}_{t,i} = \bar{\boldsymbol{\nu}}_{t-1,i} + |g_{t,i}|^2 \mathbf{1}_{|G_{t,i}| < \frac{\sigma_0}{\sigma_1}}$. One can easily observe that $\bar{\boldsymbol{\nu}}_{t,i} \le \boldsymbol{\nu}_{t,i}$, and thus

$$
\sum_{t=1}^{T} \mathbb{E}\left[ \left( \sqrt{\widetilde{\boldsymbol{\nu}}_{t+1,i}} - \sqrt{\beta_2 \widetilde{\boldsymbol{\nu}}_{t,i}} \right) \mathbf{1}_{|G_{t,i}| < \frac{\sigma_0^2}{\sigma_1^2}} \right]
$$

$$
\le \sum_{t=1}^{T} \mathbb{E}\left( \sqrt{\beta_2 \bar{\boldsymbol{\nu}}_{t,i} + (1-\beta_2)\sigma_0^2} - \sqrt{\beta_2(\beta_2 \bar{\boldsymbol{\nu}}_{t-1,i} + (1-\beta_2)\sigma_0^2)} \right)
$$

$$
= \mathbb{E}\sqrt{\beta_2 \bar{\boldsymbol{\nu}}_{t,i} + (1-\beta_2)\sigma_0^2} + (1 - \sqrt{\beta_2}) \sum_{t=1}^{T-1} \mathbb{E}\sqrt{\beta_2 \bar{\boldsymbol{\nu}}_{t,i} + (1-\beta_2)\sigma_0^2} - \mathbb{E}\sqrt{\beta_2(\beta_2 \bar{\boldsymbol{\nu}}_{0,i} + (1-\beta_2)\sigma_0^2)}.
$$

Putting the above two estimations together, we derive that

$$
(1 - \sqrt{\beta_2}) \sum_{t=1}^{T+1} \mathbb{E}\sqrt{\widetilde{\boldsymbol{\nu}}_{t,i}} \le \frac{3(1-\beta_2)\sigma_1^2}{\sqrt{\beta_2}} \sum_{t=2}^{T} \mathbb{E}\left[ \frac{|\boldsymbol{G}_{t,i}|^2}{\sqrt{\widetilde{\boldsymbol{\nu}}_{t,i}}} \right] + (1 - \sqrt{\beta_2})(T+1)\sqrt{\sigma_0^2 + \boldsymbol{\nu}_{0,i}}.
$$

The above methodology can be summarized as the following lemma.

**Lemma 3.** *Let all conditions in Theorem 1 hold. Then,*

$$
\sum_{t=1}^{T+1} \sum_{i=1}^{d} \mathbb{E}\sqrt{\widetilde{\boldsymbol{\nu}}_{t,i}} \le 2(T+1) \sum_{i=1}^{d} \sqrt{\boldsymbol{\nu}_{0,i} + \sigma_0^2} + 24 d \frac{\sigma_1^2 C_1}{\sqrt{\beta_2}} \ln d \frac{\sigma_1^2 C_1}{\sqrt{\beta_2}} + C_2.
$$

Based on Lemma 3, we can derive the estimation of $\sum_{t=1}^{T} \mathbb{E}[\|\sqrt[4]{\widetilde{\boldsymbol{\nu}}_t}\|^2]$ since $\widetilde{\boldsymbol{\nu}}_t$ is close to $\boldsymbol{\nu}_t$. The proof is then completed by combining the estimation of $\sum_{t=1}^{T} \mathbb{E}[\|\sqrt[4]{\widetilde{\boldsymbol{\nu}}_t}\|^2]$ (Eq. (6)) and Eq. (7).

## 5 Gap-closing upper bound on the iteration complexity of Adam

In this section, based on a refined proof of Stage II of Theorem 1 (see Appendix C) under the specific case $\eta = \Theta(1/\sqrt{T})$ and $\beta_2 = 1 - \Theta(1/T)$, we show that the logarithmic factor in Theorem 1 can be removed and the lower bound can be achieved. Specifically, we have the following theorem.

**Theorem 2.** *Let Assumption 1 and Assumption 2 hold. Then, select the hyperparameters of Adam as $\eta = \frac{a}{\sqrt{T}}$, $\beta_2 = 1 - \frac{b}{T}$ and $\beta_1 = c\sqrt{\beta_2}$, where $a, b > 0$ and $0 \le c < 1$ are independent of $T$. Then, let $\boldsymbol{w}_\tau$ be the output of Adam in Algorithm 1, and we have*

$$
\mathbb{E}\|\nabla f(\boldsymbol{w}_r)\| \le \left( 2\sum_{i=1}^{d} \sqrt{\boldsymbol{\nu}_{0,i} + 3b\sigma_0^2} + \frac{4D_2\sigma_1^2 b}{\sqrt{T}} + \frac{256\sigma_1^2 b}{(1-c)^2 T} \sum_{i=1}^{d} \mathbb{E}\frac{\boldsymbol{G}_{1,i}^2}{\sqrt{\widetilde{\boldsymbol{\nu}}_{1,i}}} + \frac{16D_1\sigma_1^2 b}{\sqrt{T}} \ln\left( e + \frac{4\tilde{D}\sigma_1^2 b}{\sqrt{T}} \right) \right)
$$

$$
\times \left( \frac{2D_1}{\sqrt{T}} \sum_{i=1}^{d} \ln\left( 2\sum_{i=1}^{d} \sqrt{\boldsymbol{\nu}_{0,i} + 3b\sigma_0^2} + \frac{4D_2\sigma_1^2 b}{\sqrt{T}} + \frac{256\sigma_1^2 b}{(1-c)^2 T} \sum_{i=1}^{d} \mathbb{E}\frac{\boldsymbol{G}_{1,i}^2}{\sqrt{\widetilde{\boldsymbol{\nu}}_{1,i}}} + \frac{16D_1\sigma_1^2 b}{\sqrt{T}} \ln\left( e + \frac{4\tilde{D}\sigma_1^2 b}{\sqrt{T}} \right) \right) \right.
$$

$$
\left. + \frac{64}{(1-c)^2 T} \sum_{i=1}^{d} \mathbb{E}\frac{\boldsymbol{G}_{1,i}^2}{\sqrt{\widetilde{\boldsymbol{\nu}}_{1,i}}} + \frac{D_2}{\sqrt{T}} \right)^{1/2},
$$

*where*

$$
D_1 \triangleq \frac{32La}{b} \frac{(1+c)^3}{(1-c)^3} + \frac{32\sigma_0}{\sqrt{b}(1-c)^3} + \frac{(1+\sigma_1^2)\sigma_1^2 L^2 da^2}{(1-c)^4 \sigma_0 \sqrt{b^3}}, D_2 \triangleq \frac{8}{a} f(\boldsymbol{u}_1) + D_1 \left( bd - \sum_{i=1}^{d} \ln \boldsymbol{\nu}_{0,i} \right).
$$

*As a result, let $\boldsymbol{A}$ be Adam in Algorithm 1, we have $\mathcal{C}_\varepsilon(\boldsymbol{A}, \Delta, L, \sigma^2) = \mathcal{O}(\frac{1}{\varepsilon^4})$.*

The proof of Theorem 2 is based on a refined solution of Challenge II in the proof of Theorem 1 under the specific hyperparameter settings, and we defer the concrete proof to Appendix D. Below we discuss on Theorem 2, comparing it with practice, with Theorem 1 and existing convergence rate of Adam, and with the convergence rate of AdaGrad.

**Alignment with the practical hyperparameter choice.** The hyperparameter setting in Theorem 2 indicates that to achieve the lower bound of iteration complexity, we need to select small $\eta$ and close-to-1 $\beta_2$, with less requirement over $\beta_1$. This agrees with the hyperparameter setting in deep learning libaries, for example, $\eta = 10^{-3}$, $\beta_2 = 0.999$, and $\beta_1 = 0.9$ in PyTorch.

**Comparison with Theorem 1 and existing works.** To our best knowledge, Theorem 2 is the first to derive the iteration complexity $\mathcal{O}(\frac{1}{\varepsilon^4})$. Previously, the state-of-art iteration complexity is $\mathcal{O}(\frac{\log 1/\varepsilon}{\varepsilon^4})$ [10] where they additionally assume bounded gradient. Theorem 2 is also tighter than Theorem 1 (while Theorem 1 holds for more general hyperparameter settings). As discussed in Section 4.1, if applying the hyperparameter setting in Theorem 2 (i.e., $\eta = \frac{a}{\sqrt{T}}$, $\beta_2 = 1 - \frac{b}{T}$ and $\beta_1 = c\sqrt{\beta_2}$) to Theorem 1, we will obtain that $\mathbb{E}\|\nabla f(\boldsymbol{w}_\tau)\| \leq \mathcal{O}(\mathrm{poly}(\log T)/\sqrt[4]{T})$ and $\mathcal{C}_\varepsilon(\boldsymbol{A}, \Delta, L, \sigma^2) = \mathcal{O}(\frac{\log 1/\varepsilon}{\varepsilon^4})$, worse than the upper bound in Theorem 2 and the lower bound in Proposition 1 by a logarithmic factor.

**Comparison with AdaGrad.** AdaGrad [12] is another popular adaptive optimizer. Under Assumptions 1 and 2, the state-of-art iteration complexity of AdaGrad is $\mathcal{O}(\frac{\log 1/\varepsilon}{\varepsilon^4})$ [13], which is worse than Adam by a logarithmic factor. Here we show that such a gap may be not due to the limitation of analysis, and can be explained by analogizing AdaGrad to Adam without momentum as SGD with diminishing learning rate to SGD with constant learning rate. To start with, the update rule of AdaGrad is given as

$$\boldsymbol{\nu}_t = \boldsymbol{\nu}_{t-1} + \boldsymbol{g}_t^{\odot 2}, \boldsymbol{w}_{t+1} = \boldsymbol{w}_t - \eta\frac{1}{\sqrt{\boldsymbol{\nu}_t}} \odot \boldsymbol{g}_t. \tag{8}$$

We first show that in Algorithm 1, if we allow the hyperparameters to be dynamical, i.e.,

$$\boldsymbol{\nu}_t = \beta_{2,t}\boldsymbol{\nu}_{t-1} + (1 - \beta_{2,t})\boldsymbol{g}_t^{\odot 2}, \boldsymbol{m}_t = \beta_{1,t}\boldsymbol{m}_{t-1} + (1 - \beta_{1,t})\boldsymbol{g}_t, \boldsymbol{w}_{t+1} = \boldsymbol{w}_t - \eta_t\frac{1}{\sqrt{\boldsymbol{\nu}_t}} \odot \boldsymbol{m}_t, \tag{9}$$

then Adam is equivalent to AdaGrad by setting $\eta_t = \frac{\eta}{\sqrt{t}}$, $\beta_{1,t} = 0$, and $\beta_{2,t} = 1 - \frac{1}{t}$. Specifically, by setting $\boldsymbol{\mu}_t = t\boldsymbol{\nu}_t$ in Eq. (9), we have Eq. (9) is equivalent to with Eq. (8) (by replacing $\boldsymbol{\nu}_t$ by $\boldsymbol{\mu}_t$ in Eq. (8)). Comparing the above hyperparameter setting with that in Theorem 2, we see that the above hyperparameter setting can be obtained by changing $T$ to $t$ and setting $c = 0$ in Theorem 2. This is similar to the relationship between SGD with diminishing learning rate $\Theta(1/\sqrt{t})$ and SGD with diminishing learning rate $\Theta(1/\sqrt{T})$. Recall that the iteration complexity of SGD with diminishing learning rate $\Theta(1/\sqrt{t})$ also has an additional logarithmic factor than SGD with constant learning rate, which may explain the gap between AdaGrad and Adam.

## 6  Limitations

Despite that our work provides the first result closing the upper bound and lower bound of the iteration complexity of Adam, there are several limitations listed as follows:

**Dependence over the dimension $d$.** The bounds in Theorem 1 and Theorem 2 is monotonously increasing with respect to $d$. This is undesired since the upper bound of iteration complexity of SGD is invariant with respect to $d$. Nevertheless, removing such an dependence over $d$ is technically hard since we need to deal with every coordinate separately due to coodinate-wise learning rate, while the descent lemma does not hold for a single coordinate but combines all coordinates together. To our best knowledge, all existing works on the convergene of Adam also suffers from the same problem. We leave removing the dependence over $d$ as an important future work.

**No better result with momentum.** It can be observed that in Theorem 1 and Theorem 2, the tightest bound is achieved when $\beta_1 = 0$ (i.e., no momentum is applied). This contradicts with the common wisdom that momentum helps to accelerate. Although the benefit of momentum is not very clear even for simple optimizer SGD with momentum, we view this as a limitation of our work and defer proving the benefit of momentum in Adam as a future work. Also, our result does not imply that setting $\beta_1$ is not as critical as setting $\beta_2$. The primary objective of this paper is to characterize the dependence on $\varepsilon$, and the importance of setting $\beta_1$ might be justified in other ways or characterizations. To help readers gain a deeper understanding of this issue, we include experiments to illustrate the dependence of performance on $\beta_1$ in Appendix E.

## Acknowledgments and Disclosure of Funding

This work was founded by the CAS Project for Young Scientists in Basic Research under Grant No. YSBR-034 and the Innovation Funding of ICT, CAS under Grant No.E000000.

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

## A Other Related works

Section 3 has provided a detailed discussion over existing convergence analysis of Adam. In this section, we briefly review other related works. Adam is proposed with a convergence analysis in online optimization [20]. The proof, however, is latter shown to be flawed in Reddi et al. [26] as it requires the adaptive learning rate of Adam to be non-increasing. This motivates a line of works modifying Adam to ensure convergence. The modifications include enforcing the adaptive learning rate to be non-increasing [26, 5], imposing upper bound and lower bound of the adaptive learning rate [23], and using different approach to estimate second-order momentum [35, 7]. Recently, Chen et al. [6] discover a new optimizer Lion through Symbolic Discovery, which uses sign operation to replace the adaptive learning rate in Adam, achieving comparable performance of Adam with less memory costs.

## B Auxilliary Lemmas

The following two lemmas are useful when bounding the second-order term.

**Lemma 4.** *Assume we have $0 < \beta_2 < 1$ and a sequence of real numbers $(a_n)_{n=1}^\infty$. Let $b_0 > 0$ and $b_n = \beta_2 b_{n-1} + (1 - \beta_2)a_n^2$. Then, we have*

$$\sum_{n=1}^T \frac{a_n^2}{b_n} \le \frac{1}{1 - \beta_2} \left( \ln \left( \frac{b_T}{b_0} \right) - T \ln \beta_2 \right).$$

**Lemma 5.** *Assume we have $0 < \beta_1^2 < \beta_2 < 1$ and a sequence of real numbers $(a_n)_{n=1}^\infty$. Let $b_0 > 0$, $b_n = \beta_2 b_{n-1} + (1 - \beta_2)a_n^2$, $c_0 = 0$, and $c_n = \beta_1 c_{n-1} + (1 - \beta_1)a_n$. Then, we have*

$$\sum_{n=1}^T \frac{|c_n|^2}{b_n} \le \frac{(1 - \beta_1)^2}{(1 - \frac{\beta_1}{\sqrt{\beta_2}})^2 (1 - \beta_2)} \left( \ln \left( \frac{b_T}{b_0} \right) - T \ln \beta_2 \right).$$

*Proof.* To begin with,

$$\frac{|c_n|}{\sqrt{b_n}} \le (1 - \beta_1) \sum_{i=1}^n \frac{\beta_1^{n-i} |a_i|}{\sqrt{b_n}} \le (1 - \beta_1) \sum_{i=1}^n \frac{\beta_1^{n-i} |a_i|}{\sqrt{b_n}} \le (1 - \beta_1) \sum_{i=1}^n \left( \frac{\beta_1}{\sqrt{\beta_2}} \right)^{n-i} \frac{|a_i|}{\sqrt{b_i}}.$$

Applying Cauchy's inequality, we obtain

$$\frac{|c_n|^2}{b_n} \le (1 - \beta_1)^2 \left( \sum_{i=1}^n \left( \frac{\beta_1}{\sqrt{\beta_2}} \right)^{n-i} \frac{|a_i|}{\sqrt{b_i}} \right)^2$$

$$\le (1 - \beta_1)^2 \left( \sum_{i=1}^n \left( \frac{\beta_1}{\sqrt{\beta_2}} \right)^{n-i} \right) \left( \sum_{i=1}^n \left( \frac{\beta_1}{\sqrt{\beta_2}} \right)^{n-i} \frac{|a_i|^2}{b_i} \right) \le \frac{(1 - \beta_1)^2}{1 - \frac{\beta_1}{\sqrt{\beta_2}}} \left( \sum_{i=1}^n \left( \frac{\beta_1}{\sqrt{\beta_2}} \right)^{n-i} \frac{|a_i|^2}{b_i} \right).$$

Summing the above inequality over $n$ from $1$ to $T$ then leads to

$$\sum_{n=1}^T \frac{|c_n|^2}{b_n} \le \frac{(1 - \beta_1)^2}{1 - \frac{\beta_1}{\sqrt{\beta_2}}} \sum_{n=1}^T \left( \sum_{i=1}^n \left( \frac{\beta_1}{\sqrt{\beta_2}} \right)^{n-i} \frac{|a_i|^2}{b_i} \right) = \frac{(1 - \beta_1)^2}{1 - \frac{\beta_1}{\sqrt{\beta_2}}} \sum_{n=1}^T \frac{|a_n|^2}{b_n} \left( \sum_{i=0}^{T-n} \left( \frac{\beta_1}{\sqrt{\beta_2}} \right)^i \right)$$

$$\le \frac{(1 - \beta_1)^2}{(1 - \frac{\beta_1}{\sqrt{\beta_2}})^2} \sum_{n=1}^T \frac{|a_n|^2}{b_n} \le \frac{(1 - \beta_1)^2}{(1 - \frac{\beta_1}{\sqrt{\beta_2}})^2 (1 - \beta_2)} \left( \ln \left( \frac{b_T}{b_0} \right) - T \ln \beta_2 \right).$$

The proof is completed. □

The following lemma bound the update norm of Adam.

**Lemma 6.** *We have $\forall t \ge 1$, $|\boldsymbol{w}_{t+1,i} - \boldsymbol{w}_{t,i}| \le \eta \frac{1 - \beta_1}{\sqrt{1 - \beta_2} \sqrt{1 - \frac{\beta_1^2}{\beta_2}}} \le \eta \frac{1 - \beta_1}{\sqrt{1 - \beta_2} \sqrt{1 - \frac{\beta_1}{\sqrt{\beta_2}}}}.$*

*Proof.* We have that

$$|\boldsymbol{w}_{t+1,i} - \boldsymbol{w}_{t,i}| = \eta \left| \frac{\boldsymbol{m}_{t,i}}{\sqrt{\boldsymbol{\nu}_{t,i}}} \right| \leq \eta \frac{\sum_{i=0}^{t-1}(1-\beta_1)\beta_1^i |\boldsymbol{g}_{t-i,l}|}{\sqrt{\sum_{i=0}^{t-1}(1-\beta_2)\beta_2^i |\boldsymbol{g}_{t-i,l}|^2 + \beta_2^t \boldsymbol{\nu}_{0,i}}}$$

$$\leq \eta \frac{1-\beta_1}{\sqrt{1-\beta_2}} \frac{\sqrt{\sum_{i=0}^{t-1}\beta_2^i |\boldsymbol{g}_{t-i,l}|^2}\sqrt{\sum_{i=0}^{t-1}\frac{\beta_1^{2i}}{\beta_2^i}}}{\sqrt{\sum_{i=0}^{t-1}\beta_2^i |\boldsymbol{g}_{t-i,l}|^2}} \leq \eta \frac{1-\beta_1}{\sqrt{1-\beta_2}\sqrt{1-\frac{\beta_1^2}{\beta_2}}}.$$

Here the second inequality is due to Cauchy's inequality. The proof is completed. $\square$

## C Proof of Theorem 1

This section collects the proof of Theorem 1. As a part of the proof, we first provide formal descriptions of Lemma 1, Lemma 2, and Lemma 3, and their corresponding proofs. We then proceed to prove Theorem 1 leveraging these lemmas.

### C.1 Formal description of Lemma 1, Lemma 2, and Lemma 3 and their proof

**Lemma 7** (Formal version of Lemma 1). *Let all conditions in Theorem 1 hold. Then, we have*

$$\mathbb{E}\left[\left\langle \boldsymbol{G}_t, -\frac{\eta}{1-\frac{\beta_1}{\sqrt{\beta_2}}}\left(\frac{1}{\sqrt{\boldsymbol{\nu}_t}} - \frac{1}{\sqrt{\tilde{\boldsymbol{\nu}}_t}}\right) \odot \boldsymbol{m}_t\right\rangle\right] \leq \frac{5}{8}\sum_{i=1}^d \eta \frac{1-\beta_1}{1-\frac{\beta_1}{\sqrt{\beta_2}}}\mathbb{E}\frac{|\boldsymbol{G}_{t,i}|^2}{\sqrt{\tilde{\boldsymbol{\nu}}_{t,i}}} + \frac{2\eta\sqrt{1-\beta_2}\sigma_0}{\left(1-\frac{\beta_1^2}{\beta_2}\right)^2}\sum_{i=1}^d \mathbb{E}\frac{\boldsymbol{g}_{t,i}^2}{\boldsymbol{\nu}_{t,i}}$$

$$+ \eta\frac{4(1-\beta_1)}{(1-\frac{\beta_1}{\sqrt{\beta_2}})^2\sqrt{\beta_2}}\sigma_1^2\sum_{i=1}^d \mathbb{E}\left(\frac{\boldsymbol{G}_{t-1,i}^2}{\sqrt{\beta_2\tilde{\boldsymbol{\nu}}_{t,i}}} - \frac{\boldsymbol{G}_{t,i}^2}{\sqrt{\tilde{\boldsymbol{\nu}}_{t+1,i}}}\right) + \sum_{i=1}^d \frac{2\eta\sqrt{1-\beta_2}\sigma_0}{(1-\beta_1)(1-\frac{\beta_1}{\sqrt{\beta_2}})}\mathbb{E}\left[\left(\frac{|\boldsymbol{m}_{t,i}|^2}{\boldsymbol{\nu}_{t,i}}\right)\right]$$

$$+ \frac{64(1+\sigma_1^2)\sigma_1^2 L^2 \eta^3 d}{\beta_2^2(1-\frac{\beta_1}{\sqrt{\beta_2}})^3(1-\beta_1)\sigma_0\sqrt{1-\beta_2}}\mathbb{E}\left\|\frac{1}{\sqrt{\boldsymbol{\nu}_{t-1}}} \odot \boldsymbol{m}_{t-1}\right\|^2.$$

*Proof.* To start with,

$$\mathbb{E}^{|\mathcal{F}_t}\left[\left\langle \boldsymbol{G}_t, -\frac{\eta}{1-\frac{\beta_1}{\sqrt{\beta_2}}}\left(\frac{1}{\sqrt{\boldsymbol{\nu}_t}} - \frac{1}{\sqrt{\tilde{\boldsymbol{\nu}}_t}}\right) \odot \boldsymbol{m}_t\right\rangle\right]$$

$$=\mathbb{E}^{|\mathcal{F}_t}\left[\left\langle \boldsymbol{G}_t, -\frac{\eta}{1-\frac{\beta_1}{\sqrt{\beta_2}}}\left(\frac{(1-\beta_2)(\sigma_0^2 \mathbb{1}_d - \boldsymbol{g}_t^{\odot 2})}{\sqrt{\boldsymbol{\nu}_t}\sqrt{\tilde{\boldsymbol{\nu}}_t}(\sqrt{\boldsymbol{\nu}_t} + \sqrt{\tilde{\boldsymbol{\nu}}_t})}\right) \odot \boldsymbol{m}_t\right\rangle\right]$$

$$\leq \sum_{i=1}^d \frac{\eta}{1-\frac{\beta_1}{\sqrt{\beta_2}}}\mathbb{E}^{|\mathcal{F}_t}\left[|\boldsymbol{G}_{t,i}|\left(\frac{(1-\beta_2)(\sigma_0^2 + \boldsymbol{g}_{t,i}^2)}{\sqrt{\boldsymbol{\nu}_{t,i}}\sqrt{\tilde{\boldsymbol{\nu}}_{t,i}}(\sqrt{\boldsymbol{\nu}_{t,i}} + \sqrt{\tilde{\boldsymbol{\nu}}_{t,i}})}\right)|\boldsymbol{m}_{t,i}|\right]$$

$$=\underbrace{\sum_{i=1}^d \frac{\eta}{1-\frac{\beta_1}{\sqrt{\beta_2}}}\mathbb{E}^{|\mathcal{F}_t}\left[|\boldsymbol{G}_{t,i}|\left(\frac{(1-\beta_2)\boldsymbol{g}_{t,i}^2}{\sqrt{\boldsymbol{\nu}_{t,i}}\sqrt{\tilde{\boldsymbol{\nu}}_{t,i}}(\sqrt{\boldsymbol{\nu}_{t,i}} + \sqrt{\tilde{\boldsymbol{\nu}}_{t,i}})}\right)|\boldsymbol{m}_{t,i}|\right]}_{\text{I.1.1}}$$

$$+\underbrace{\sum_{i=1}^d \frac{\eta}{1-\frac{\beta_1}{\sqrt{\beta_2}}}\mathbb{E}^{|\mathcal{F}_t}\left[|\boldsymbol{G}_{t,i}|\left(\frac{(1-\beta_2)\sigma_0^2}{\sqrt{\boldsymbol{\nu}_{t,i}}\sqrt{\tilde{\boldsymbol{\nu}}_{t,i}}(\sqrt{\boldsymbol{\nu}_{t,i}} + \sqrt{\tilde{\boldsymbol{\nu}}_{t,i}})}\right)|\boldsymbol{m}_{t,i}|\right]}_{\text{I.1.2}}.$$

As for I.1.1, we have

$$\sum_{i=1}^{d} \frac{\eta}{1 - \frac{\beta_1}{\sqrt{\beta_2}}} \mathbb{E}^{|\mathcal{F}_t} \left[ |\boldsymbol{G}_{t,i}| \left( \frac{(1-\beta_2)\boldsymbol{g}_{t,i}^2}{\sqrt{\boldsymbol{\nu}_{t,i}}\sqrt{\tilde{\boldsymbol{\nu}}_{t,i}}(\sqrt{\boldsymbol{\nu}_{t,i}} + \sqrt{\tilde{\boldsymbol{\nu}}_{t,i}})} \right) |\boldsymbol{m}_{t,i}| \right]$$

$$\overset{(*)}{\leq} \sum_{i=1}^{d} \frac{\eta(1-\beta_1)}{\left(\sqrt{1 - \frac{\beta_1}{\sqrt{\beta_2}}}\right)^3} \mathbb{E}^{|\mathcal{F}_t} \left[ |\boldsymbol{G}_{t,i}| \left( \frac{\sqrt{1-\beta_2}\boldsymbol{g}_{t,i}^2}{\sqrt{\tilde{\boldsymbol{\nu}}_{t,i}}(\sqrt{\boldsymbol{\nu}_{t,i}} + \sqrt{\tilde{\boldsymbol{\nu}}_{t,i}})} \right) \right]$$

$$\overset{(\circ)}{\leq} \sum_{i=1}^{d} \frac{\eta(1-\beta_1)}{\left(\sqrt{1 - \frac{\beta_1}{\sqrt{\beta_2}}}\right)^3} \frac{|\boldsymbol{G}_{t,i}|}{\sqrt{\tilde{\boldsymbol{\nu}}_{t,i}}} \sqrt{\mathbb{E}^{|\mathcal{F}_t} \boldsymbol{g}_{t,i}^2} \sqrt{\mathbb{E}^{|\mathcal{F}_t} \frac{\boldsymbol{g}_{t,i}^2}{(\sqrt{\boldsymbol{\nu}_{t,i}} + \sqrt{\tilde{\boldsymbol{\nu}}_{t,i}})^2}}$$

$$\overset{(\bullet)}{\leq} \sum_{i=1}^{d} \frac{\eta(1-\beta_1)\sqrt{1-\beta_2}}{\left(\sqrt{1 - \frac{\beta_1}{\sqrt{\beta_2}}}\right)^3} \frac{|\boldsymbol{G}_{t,i}|}{\sqrt{\tilde{\boldsymbol{\nu}}_{t,i}}} \sqrt{\sigma_0^2 + \sigma_1^2 \boldsymbol{G}_{t,i}^2} \sqrt{\mathbb{E}^{|\mathcal{F}_t} \frac{\boldsymbol{g}_{t,i}^2}{(\sqrt{\boldsymbol{\nu}_{t,i}} + \sqrt{\tilde{\boldsymbol{\nu}}_{t,i}})^2}}$$

$$\leq \sum_{i=1}^{d} \frac{\eta(1-\beta_1)\sqrt{1-\beta_2}}{\left(\sqrt{1 - \frac{\beta_1}{\sqrt{\beta_2}}}\right)^3} \frac{|\boldsymbol{G}_{t,i}|}{\sqrt{\tilde{\boldsymbol{\nu}}_{t,i}}} (\sigma_0 + \sigma_1|\boldsymbol{G}_{t,i}|) \sqrt{\mathbb{E}^{|\mathcal{F}_t} \frac{\boldsymbol{g}_{t,i}^2}{(\sqrt{\boldsymbol{\nu}_{t,i}} + \sqrt{\tilde{\boldsymbol{\nu}}_{t,i}})^2}},$$

where inequality $(*)$ uses Lemma 6, inequality $(\circ)$ is due to Holder's inequality, and inequality $(\bullet)$ is due to Assumption 3. Applying mean-value inequality respectively to $\sum_{i=1}^{d} \frac{\eta(1-\beta_1)\sqrt{1-\beta_2}}{\left(\sqrt{1 - \frac{\beta_1}{\sqrt{\beta_2}}}\right)^3} \mathbb{E}^{|\mathcal{F}_t} \frac{|\boldsymbol{G}_{t,i}|}{\sqrt{\tilde{\boldsymbol{\nu}}_{t,i}}} \sigma_0 \sqrt{\mathbb{E}^{|\mathcal{F}_t} \frac{\boldsymbol{g}_{t,i}^2}{(\sqrt{\boldsymbol{\nu}_{t,i}} + \sqrt{\tilde{\boldsymbol{\nu}}_{t,i}})^2}}$ and

$\sum_{i=1}^{d} \frac{\eta(1-\beta_1)\sqrt{1-\beta_2}}{\left(\sqrt{1 - \frac{\beta_1}{\sqrt{\beta_2}}}\right)^3} \mathbb{E}^{|\mathcal{F}_t} \frac{|\boldsymbol{G}_{t,i}|}{\sqrt{\tilde{\boldsymbol{\nu}}_{t,i}}} \sigma_1 |\boldsymbol{G}_{t,i}| \sqrt{\mathbb{E}^{|\mathcal{F}_t} \frac{\boldsymbol{g}_{t,i}^2}{(\sqrt{\boldsymbol{\nu}_{t,i}} + \sqrt{\tilde{\boldsymbol{\nu}}_{t,i}})^2}}$, we obtain that the right-hand-side of the above inequality can be bounded by

$$\frac{1}{8} \sum_{i=1}^{d} \eta \frac{1-\beta_1}{1 - \frac{\beta_1}{\sqrt{\beta_2}}} \sqrt{1-\beta_2} \sigma_0 \frac{|\boldsymbol{G}_{t,i}|^2}{\tilde{\boldsymbol{\nu}}_{t,i}} + \frac{2\eta\sqrt{1-\beta_2}\sigma_0}{\left(1 - \frac{\beta_1}{\sqrt{\beta_2}}\right)^2} \sum_{i=1}^{d} \mathbb{E}^{|\mathcal{F}_t} \frac{\boldsymbol{g}_{t,i}^2}{(\sqrt{\boldsymbol{\nu}_{t,i}} + \sqrt{\tilde{\boldsymbol{\nu}}_{t,i}})^2}$$

$$+ \frac{1}{8} \sum_{i=1}^{d} \eta \frac{1-\beta_1}{1 - \frac{\beta_1}{\sqrt{\beta_2}}} \frac{|\boldsymbol{G}_{t,i}|^2}{\sqrt{\tilde{\boldsymbol{\nu}}_{t,i}}} + 2\eta \frac{(1-\beta_2)(1-\beta_1)}{(1 - \frac{\beta_1}{\sqrt{\beta_2}})^2} \sigma_1^2 \frac{|\boldsymbol{G}_{t,i}|^2}{\sqrt{\tilde{\boldsymbol{\nu}}_{t,i}}} \mathbb{E}^{|\mathcal{F}_t} \sum_{i=1}^{d} \frac{\boldsymbol{g}_{t,i}^2}{(\sqrt{\boldsymbol{\nu}_{t,i}} + \sqrt{\tilde{\boldsymbol{\nu}}_{t,i}})^2}$$

$$\leq \frac{1}{8} \sum_{i=1}^{d} \eta \frac{1-\beta_1}{1 - \frac{\beta_1}{\sqrt{\beta_2}}} \frac{|\boldsymbol{G}_{t,i}|^2}{\sqrt{\tilde{\boldsymbol{\nu}}_{t,i}}} + \frac{2\eta\sqrt{1-\beta_2}\sigma_0}{\left(1 - \frac{\beta_1}{\sqrt{\beta_2}}\right)^2} \sum_{i=1}^{d} \mathbb{E}^{|\mathcal{F}_t} \frac{\boldsymbol{g}_{t,i}^2}{\boldsymbol{\nu}_{t,i}}$$

$$+ \frac{1}{8} \sum_{i=1}^{d} \eta \frac{1-\beta_1}{1 - \frac{\beta_1}{\sqrt{\beta_2}}} \frac{|\boldsymbol{G}_{t,i}|^2}{\sqrt{\tilde{\boldsymbol{\nu}}_{t,i}}} + 2\eta \frac{(1-\beta_2)(1-\beta_1)}{(1 - \frac{\beta_1}{\sqrt{\beta_2}})^2} \sigma_1^2 \frac{|\boldsymbol{G}_{t,i}|^2}{\sqrt{\tilde{\boldsymbol{\nu}}_{t,i}}} \mathbb{E}^{|\mathcal{F}_t} \sum_{i=1}^{d} \frac{\boldsymbol{g}_{t,i}^2}{(\sqrt{\boldsymbol{\nu}_{t,i}} + \sqrt{\tilde{\boldsymbol{\nu}}_{t,i}})^2}. \quad (10)$$

Here the inequality is due to $\tilde{\boldsymbol{\nu}}_{t,i} = (1-\beta_2)\sigma_0^2 + \beta_2 \boldsymbol{\nu}_{t-1,i} \geq (1-\beta_2)\sigma_0^2$. Meanwhile, we have

$$\left( \frac{1}{\sqrt{\beta_2 \tilde{\boldsymbol{\nu}}_{t,i}}} - \frac{1}{\sqrt{\tilde{\boldsymbol{\nu}}_{t+1,i}}} \right) \boldsymbol{G}_{t,i}^2$$

$$= \frac{\boldsymbol{G}_{t,i}^2((1-\beta_2)^2\sigma_0^2 + \beta_2(1-\beta_2)\boldsymbol{g}_{t,i}^2)}{\sqrt{\beta_2\tilde{\boldsymbol{\nu}}_{t,i}}\sqrt{\tilde{\boldsymbol{\nu}}_{t+1,i}}(\sqrt{\beta_2\tilde{\boldsymbol{\nu}}_{t,i}} + \sqrt{\tilde{\boldsymbol{\nu}}_{t+1,i}})} \geq \frac{\boldsymbol{G}_{t,i}^2\beta_2(1-\beta_2)\boldsymbol{g}_{t,i}^2}{\sqrt{\beta_2\tilde{\boldsymbol{\nu}}_{t,i}}\sqrt{\tilde{\boldsymbol{\nu}}_{t+1,i}}(\sqrt{\beta_2\tilde{\boldsymbol{\nu}}_{t,i}} + \sqrt{\tilde{\boldsymbol{\nu}}_{t+1,i}})}$$

$$\geq \frac{\sqrt{\beta_2}}{2} \frac{\boldsymbol{G}_{t,i}^2(1-\beta_2)\boldsymbol{g}_{t,i}^2}{\sqrt{\tilde{\boldsymbol{\nu}}_{t,i}}(\sqrt{\boldsymbol{\nu}_{t,i}} + \sqrt{\tilde{\boldsymbol{\nu}}_{t,i}})^2}.$$

Applying the above inequality back to Eq. (10), we obtain that

$$
\sum_{i=1}^{d} \frac{\eta}{1-\beta_1} \mathbb{E}^{|\mathcal{F}_t} \left[ |\boldsymbol{G}_{t,i}| \left( \frac{(1-\beta_2)\boldsymbol{g}_{t,i}^2}{\sqrt{\boldsymbol{\nu}_{t,i}}\sqrt{\tilde{\boldsymbol{\nu}}_{t,i}}(\sqrt{\boldsymbol{\nu}_{t,i}}+\sqrt{\tilde{\boldsymbol{\nu}}_{t,i}})} \right) |\boldsymbol{m}_{t,i}| \right]
$$

$$
\leq \frac{1}{4} \sum_{i=1}^{d} \eta \frac{1-\beta_1}{1-\frac{\beta_1}{\sqrt{\beta_2}}} \frac{|\boldsymbol{G}_{t,i}|^2}{\sqrt{\tilde{\boldsymbol{\nu}}_{t,i}}} + \frac{2\eta\sqrt{1-\beta_2}\sigma_0}{\left(1-\frac{\beta_1^2}{\beta_2}\right)^2} \sum_{i=1}^{d} \mathbb{E}^{|\mathcal{F}_t} \frac{\boldsymbol{g}_{t,i}^2}{\boldsymbol{\nu}_{t,i}}
$$

$$
+ \eta \frac{4(1-\beta_1)}{(1-\frac{\beta_1}{\sqrt{\beta_2}})^2\sqrt{\beta_2}} \sigma_1^2 \sum_{i=1}^{d} \mathbb{E}^{|\mathcal{F}_t} \left( \frac{1}{\sqrt{\beta_2\tilde{\boldsymbol{\nu}}_{t,i}}} - \frac{1}{\sqrt{\tilde{\boldsymbol{\nu}}_{t+1,i}}} \right) \boldsymbol{G}_{t,i}^2. \tag{11}
$$

Furthermore, due to Assumption 1, we have (we define $G_0 \triangleq G_1$)

$$
\boldsymbol{G}_{t,i}^2 \leq \boldsymbol{G}_{t-1,i}^2 + 2|\boldsymbol{G}_{t,i}||\boldsymbol{G}_{t,i} - \boldsymbol{G}_{t-1,i}| + 2(\boldsymbol{G}_{t,i} - \boldsymbol{G}_{t-1,i})^2
$$
$$
\leq \boldsymbol{G}_{t-1,i}^2 + 2L|\boldsymbol{G}_{t,i}|\|\boldsymbol{w}_t - \boldsymbol{w}_{t-1}\| + 2L^2\|\boldsymbol{w}_t - \boldsymbol{w}_{t-1}\|^2,
$$

which further leads to

$$
\frac{1}{\sqrt{\beta_2\tilde{\boldsymbol{\nu}}_{t,i}}} \boldsymbol{G}_{t,i}^2
$$

$$
\leq \frac{1}{\sqrt{\beta_2\tilde{\boldsymbol{\nu}}_{t,i}}} \left( \boldsymbol{G}_{t-1,i}^2 + 2L|\boldsymbol{G}_{t,i}|\|\boldsymbol{w}_t - \boldsymbol{w}_{t-1}\| + 2L^2\|\boldsymbol{w}_t - \boldsymbol{w}_{t-1}\|^2 \right)
$$

$$
\overset{(\circ)}{\leq} \frac{1}{\sqrt{\beta_2\tilde{\boldsymbol{\nu}}_{t,i}}} \boldsymbol{G}_{t-1,i}^2 + \frac{(1-\frac{\beta_1}{\sqrt{\beta_2}})(1-\beta_1)\sqrt{\beta_2}}{16\sigma_1^2} \frac{|\boldsymbol{G}_{t,i}|^2}{\sqrt{\tilde{\boldsymbol{\nu}}_{t,i}}} + \frac{16L^2\sigma_1^2}{\beta_2^{\frac{3}{2}}(1-\frac{\beta_1}{\sqrt{\beta_2}})(1-\beta_1)\sqrt{\tilde{\boldsymbol{\nu}}_{t,i}}} \|\boldsymbol{w}_t - \boldsymbol{w}_{t-1}\|^2
$$

$$
+ \frac{2L^2}{\sqrt{\beta_2\tilde{\boldsymbol{\nu}}_{t,i}}} \|\boldsymbol{w}_t - \boldsymbol{w}_{t-1}\|^2
$$

$$
\leq \frac{1}{\sqrt{\beta_2\tilde{\boldsymbol{\nu}}_{t,i}}} \boldsymbol{G}_{t-1,i}^2 + \frac{(1-\frac{\beta_1}{\sqrt{\beta_2}})(1-\beta_1)\sqrt{\beta_2}}{16\sigma_1^2} \frac{|\boldsymbol{G}_{t,i}|^2}{\sqrt{\tilde{\boldsymbol{\nu}}_{t,i}}} + \frac{16L^2\sigma_1^2\eta^2}{\beta_2^{\frac{3}{2}}(1-\frac{\beta_1}{\sqrt{\beta_2}})(1-\beta_1)\sigma_0\sqrt{1-\beta_2}} \left\| \frac{1}{\sqrt{\boldsymbol{\nu}_{t-1}}} \odot \boldsymbol{m}_{t-1} \right\|^2
$$

$$
+ \frac{2L^2\eta^2}{\sigma_0\sqrt{\beta_2(1-\beta_2)}} \left\| \frac{1}{\sqrt{\boldsymbol{\nu}_{t-1}}} \odot \boldsymbol{m}_{t-1} \right\|^2
$$

$$
\leq \frac{1}{\sqrt{\beta_2\tilde{\boldsymbol{\nu}}_{t,i}}} \boldsymbol{G}_{t-1,i}^2 + \frac{(1-\frac{\beta_1}{\sqrt{\beta_2}})(1-\beta_1)\sqrt{\beta_2}}{16\sigma_1^2} \frac{|\boldsymbol{G}_{t,i}|^2}{\sqrt{\tilde{\boldsymbol{\nu}}_{t,i}}} + \frac{16(1+\sigma_1^2)L^2\eta^2}{\beta_2^{\frac{3}{2}}(1-\frac{\beta_1}{\sqrt{\beta_2}})(1-\beta_1)\sigma_0\sqrt{1-\beta_2}} \left\| \frac{1}{\sqrt{\boldsymbol{\nu}_{t-1}}} \odot \boldsymbol{m}_{t-1} \right\|^2.
$$

Applying the above inequality back to Eq. (11) leads to that

$$
\text{I.1.1} = \sum_{i=1}^{d} \frac{\eta}{1-\beta_1} \mathbb{E}^{|\mathcal{F}_t} \left[ |\boldsymbol{G}_{t,i}| \left( \frac{(1-\beta_2)\boldsymbol{g}_{t,i}^2}{\sqrt{\boldsymbol{\nu}_{t,i}}\sqrt{\tilde{\boldsymbol{\nu}}_{t,i}}(\sqrt{\boldsymbol{\nu}_{t,i}}+\sqrt{\tilde{\boldsymbol{\nu}}_{t,i}})} \right) |\boldsymbol{m}_{t,i}| \right]
$$

$$
\leq \frac{1}{2} \sum_{i=1}^{d} \eta \frac{1-\beta_1}{1-\frac{\beta_1}{\sqrt{\beta_2}}} \frac{|\boldsymbol{G}_{t,i}|^2}{\sqrt{\tilde{\boldsymbol{\nu}}_{t,i}}} + \frac{2\eta\sqrt{1-\beta_2}\sigma_0}{\left(1-\frac{\beta_1^2}{\beta_2}\right)^2} \sum_{i=1}^{d} \mathbb{E}^{|\mathcal{F}_t} \frac{\boldsymbol{g}_{t,i}^2}{\boldsymbol{\nu}_{t,i}}
$$

$$
+ \eta \frac{4(1-\beta_1)}{(1-\frac{\beta_1}{\sqrt{\beta_2}})^2\sqrt{\beta_2}} \sigma_1^2 \sum_{i=1}^{d} \mathbb{E}^{|\mathcal{F}_t} \left( \frac{\boldsymbol{G}_{t-1,i}^2}{\sqrt{\beta_2\tilde{\boldsymbol{\nu}}_{t,i}}} - \frac{\boldsymbol{G}_{t,i}^2}{\sqrt{\tilde{\boldsymbol{\nu}}_{t+1,i}}} \right)
$$

$$
+ \frac{64d(1+\sigma_1^2)\sigma_1^2 L^2\eta^3}{\beta_2^2(1-\frac{\beta_1}{\sqrt{\beta_2}})^3(1-\beta_1)\sigma_0\sqrt{1-\beta_2}} \left\| \frac{1}{\sqrt{\boldsymbol{\nu}_{t-1}}} \odot \boldsymbol{m}_{t-1} \right\|^2. \tag{12}
$$

As for I.1.2, we have

$$\text{I.1.2} = \sum_{i=1}^{d} \frac{\eta}{1 - \frac{\beta_1}{\sqrt{\beta_2}}} \mathbb{E}^{|\mathcal{F}_t} \left[ |\boldsymbol{G}_{t,i}| \left( \frac{(1 - \beta_2)\sigma_0^2}{\sqrt{\boldsymbol{\nu}_{t,i}}\sqrt{\tilde{\boldsymbol{\nu}}_{t,i}}(\sqrt{\boldsymbol{\nu}_{t,i}} + \sqrt{\tilde{\boldsymbol{\nu}}_{t,i}})} \right) |\boldsymbol{m}_{t,i}| \right]$$

$$\leq \sum_{i=1}^{d} \frac{\eta}{1 - \frac{\beta_1}{\sqrt{\beta_2}}} \mathbb{E}^{|\mathcal{F}_t} \left[ |\boldsymbol{G}_{t,i}| \left( \frac{\sqrt[4]{1 - \beta_2}\sqrt{\sigma_0}}{\sqrt[4]{\tilde{\boldsymbol{\nu}}_{t,i}}\sqrt{\boldsymbol{\nu}_{t,i}}} \right) |\boldsymbol{m}_{t,i}| \right]$$

$$\leq \frac{1 - \beta_1}{8(1 - \frac{\beta_1}{\sqrt{\beta_2}})} \sum_{i=1}^{d} \eta \frac{|\boldsymbol{G}_{t,i}|^2}{\sqrt{\tilde{\boldsymbol{\nu}}_{t,i}}} + \sum_{i=1}^{d} \frac{2\eta\sqrt{1 - \beta_2}\sigma_0}{(1 - \beta_1)(1 - \frac{\beta_1}{\sqrt{\beta_2}})} \mathbb{E}^{|\mathcal{F}_t} \left[ \left( \frac{|\boldsymbol{m}_{t,i}|^2}{\boldsymbol{\nu}_{t,i}} \right) \right]. \quad (13)$$

With Inequalities (12) and (13), we conclude that

$$\text{I.1} \leq \frac{5}{8} \sum_{i=1}^{d} \eta \frac{1 - \beta_1}{1 - \frac{\beta_1}{\sqrt{\beta_2}}} \mathbb{E} \frac{|\boldsymbol{G}_{t,i}|^2}{\sqrt{\tilde{\boldsymbol{\nu}}_{t,i}}} + \frac{2\eta\sqrt{1 - \beta_2}\sigma_0}{\left( 1 - \frac{\beta_1^2}{\beta_2} \right)^2} \sum_{i=1}^{d} \mathbb{E} \frac{\boldsymbol{g}_{t,i}^2}{\boldsymbol{\nu}_{t,i}}$$

$$+ \eta \frac{4(1 - \beta_1)}{(1 - \frac{\beta_1}{\sqrt{\beta_2}})^2 \sqrt{\beta_2}} \sigma_1^2 \sum_{i=1}^{d} \mathbb{E} \left( \frac{\boldsymbol{G}_{t-1,i}^2}{\sqrt{\beta_2 \tilde{\boldsymbol{\nu}}_{t,i}}} - \frac{\boldsymbol{G}_{t,i}^2}{\sqrt{\tilde{\boldsymbol{\nu}}_{t+1,i}}} \right) + \sum_{i=1}^{d} \frac{2\eta\sqrt{1 - \beta_2}\sigma_0}{(1 - \beta_1)(1 - \frac{\beta_1}{\sqrt{\beta_2}})} \mathbb{E} \left[ \left( \frac{|\boldsymbol{m}_{t,i}|^2}{\boldsymbol{\nu}_{t,i}} \right) \right]$$

$$+ \frac{64(1 + \sigma_1^2)\sigma_1^2 L^2 \eta^3 d}{\beta_2^2 (1 - \frac{\beta_1}{\sqrt{\beta_2}})^3 (1 - \beta_1)\sigma_0 \sqrt{1 - \beta_2}} \mathbb{E} \left\| \frac{1}{\sqrt{\boldsymbol{\nu}_{t-1}}} \odot \boldsymbol{m}_{t-1} \right\|^2.$$

$\square$

**Lemma 8** (Formal version of Lemma 2). *Let all conditions in Theorem 1 holds. Then,*

$$\mathbb{E} f(\boldsymbol{u}_{t+1})$$

$$\leq \mathbb{E} f(\boldsymbol{u}_t) - \frac{\eta}{4} \frac{1 - \beta_1}{1 - \frac{\beta_1}{\sqrt{\beta_2}}} \mathbb{E} \left[ \eta \left\langle \boldsymbol{G}_t, -\frac{1}{\sqrt{\tilde{\boldsymbol{\nu}}_t}} \odot \boldsymbol{G}_t \right\rangle \right] + \frac{2\eta\sqrt{1 - \beta_2}\sigma_0}{\left( 1 - \frac{\beta_1^2}{\beta_2} \right)^2} \sum_{i=1}^{d} \mathbb{E} \frac{\boldsymbol{g}_{t,i}^2}{\boldsymbol{\nu}_{t,i}}$$

$$+ \eta \frac{4}{(1 - \frac{\beta_1}{\sqrt{\beta_2}})^2 \sqrt{\beta_2}} \sigma_1^2 \sum_{i=1}^{d} \mathbb{E} \left( \frac{1}{\sqrt{\beta_2}} \xi_{t-1} - \xi_t \right) + \sum_{i=1}^{d} \frac{2\eta\sqrt{1 - \beta_2}\sigma_0}{(1 - \beta_1)(1 - \frac{\beta_1}{\sqrt{\beta_2}})} \mathbb{E} \left[ \left( \frac{|\boldsymbol{m}_{t,i}|^2}{\boldsymbol{\nu}_{t,i}} \right) \right]$$

$$+ \frac{64(1 + \sigma_1^2)\sigma_1^2 L^2 \eta^3 d}{\beta_2^2 (1 - \frac{\beta_1}{\sqrt{\beta_2}})^3 (1 - \beta_1)\sigma_0 \sqrt{1 - \beta_2}} \mathbb{E} \left\| \frac{1}{\sqrt{\boldsymbol{\nu}_{t-1}}} \odot \boldsymbol{m}_{t-1} \right\|^2 + \frac{2\eta\sqrt{1 - \beta_2}\beta_1^2 \sigma_0}{(1 - \beta_1)(1 - \frac{\beta_1}{\sqrt{\beta_2}})\beta_2} \sum_{i=1}^{d} \mathbb{E} \left[ \frac{|\boldsymbol{m}_{t-1,i}|^2}{\boldsymbol{\nu}_{t-1,i}} \right]$$

$$+ L\mathbb{E} \left[ 4 \left( \frac{\frac{\beta_1}{\sqrt{\beta_2}}}{1 - \frac{\beta_1}{\sqrt{\beta_2}}} \right)^2 \eta^2 \left\| \frac{1}{\sqrt{\boldsymbol{\nu}_{t-1}}} \odot \boldsymbol{m}_{t-1} \right\|^2 + 3 \left( \frac{1}{1 - \frac{\beta_1}{\sqrt{\beta_2}}} \right)^2 \eta^2 \left\| \frac{1}{\sqrt{\boldsymbol{\nu}_t}} \odot \boldsymbol{m}_t \right\|^2 \right].$$

*Proof.* According to the definition of $\boldsymbol{u}_t$, we have

$$
\boldsymbol{u}_{t+1} - \boldsymbol{u}_t = \frac{\boldsymbol{w}_{t+1} - \boldsymbol{w}_t}{1 - \frac{\beta_1}{\sqrt{\beta_2}}} - \frac{\beta_1}{\sqrt{\beta_2}} \frac{\boldsymbol{w}_t - \boldsymbol{w}_{t-1}}{1 - \frac{\beta_1}{\sqrt{\beta_2}}}
$$

$$
= -\frac{\eta}{1 - \frac{\beta_1}{\sqrt{\beta_2}}} \frac{1}{\sqrt{\boldsymbol{\nu}_t}} \odot \boldsymbol{m}_t + \beta_1 \frac{\eta}{1 - \frac{\beta_1}{\sqrt{\beta_2}}} \frac{1}{\sqrt{\beta_2 \boldsymbol{\nu}_{t-1}}} \odot \boldsymbol{m}_{t-1}
$$

$$
= -\frac{\eta}{1 - \frac{\beta_1}{\sqrt{\beta_2}}} \frac{1}{\sqrt{\widetilde{\boldsymbol{\nu}}_t}} \odot \boldsymbol{m}_t + \beta_1 \frac{\eta}{1 - \frac{\beta_1}{\sqrt{\beta_2}}} \frac{1}{\sqrt{\widetilde{\boldsymbol{\nu}}_t}} \odot \boldsymbol{m}_{t-1}
$$

$$
- \frac{\eta}{1 - \frac{\beta_1}{\sqrt{\beta_2}}} \left( \frac{1}{\sqrt{\boldsymbol{\nu}_t}} - \frac{1}{\sqrt{\widetilde{\boldsymbol{\nu}}_t}} \right) \odot \boldsymbol{m}_t + \beta_1 \frac{\eta}{1 - \frac{\beta_1}{\sqrt{\beta_2}}} \left( \frac{1}{\sqrt{\beta_2 \boldsymbol{\nu}_{t-1}}} - \frac{1}{\sqrt{\widetilde{\boldsymbol{\nu}}_t}} \right) \odot \boldsymbol{m}_{t-1}
$$

$$
\overset{(*)}{=} -\eta \frac{1 - \beta_1}{1 - \frac{\beta_1}{\sqrt{\beta_2}}} \frac{1}{\sqrt{\widetilde{\boldsymbol{\nu}}_t}} \odot \boldsymbol{g}_t
$$

$$
- \frac{\eta}{1 - \frac{\beta_1}{\sqrt{\beta_2}}} \left( \frac{1}{\sqrt{\boldsymbol{\nu}_t}} - \frac{1}{\sqrt{\widetilde{\boldsymbol{\nu}}_t}} \right) \odot \boldsymbol{m}_t + \beta_1 \frac{\eta}{1 - \frac{\beta_1}{\sqrt{\beta_2}}} \left( \frac{1}{\sqrt{\beta_2 \boldsymbol{\nu}_{t-1}}} - \frac{1}{\sqrt{\widetilde{\boldsymbol{\nu}}_t}} \right) \odot \boldsymbol{m}_{t-1},
$$

where Eq. $(*)$ is due to $\boldsymbol{m}_t = \beta_1 \boldsymbol{m}_{t-1} + (1 - \beta_1)\boldsymbol{g}_t$.

Applying the above equation to the "First Order" term, we find that it can be decomposed as

$$
\mathbb{E}\left[ \langle \nabla f(\boldsymbol{u}_t), \boldsymbol{u}_{t+1} - \boldsymbol{u}_t \rangle \right]
$$

$$
= \mathbb{E}\left[ \langle \boldsymbol{G}_t, \boldsymbol{u}_{t+1} - \boldsymbol{u}_t \rangle \right] + \mathbb{E}\left[ \langle \nabla f(\boldsymbol{u}_t) - G_t, \boldsymbol{u}_{t+1} - \boldsymbol{u}_t \rangle \right]
$$

$$
= \mathbb{E}\left[ \left\langle \boldsymbol{G}_t, -\eta \frac{1}{\sqrt{\widetilde{\boldsymbol{\nu}}_t}} \odot \boldsymbol{g}_t \right\rangle \right] + \mathbb{E}\left[ \left\langle \boldsymbol{G}_t, -\frac{\eta}{1 - \frac{\beta_1}{\sqrt{\beta_2}}} \left( \frac{1}{\sqrt{\boldsymbol{\nu}_t}} - \frac{1}{\sqrt{\widetilde{\boldsymbol{\nu}}_t}} \right) \odot \boldsymbol{m}_t \right\rangle \right]
$$

$$
+ \mathbb{E}\left[ \left\langle \boldsymbol{G}_t, \beta_1 \frac{\eta}{1 - \frac{\beta_1}{\sqrt{\beta_2}}} \left( \frac{1}{\sqrt{\beta_2 \boldsymbol{\nu}_{t-1}}} - \frac{1}{\sqrt{\widetilde{\boldsymbol{\nu}}_t}} \right) \odot \boldsymbol{m}_{t-1} \right\rangle \right] + \mathbb{E}\left[ \langle \nabla f(\boldsymbol{u}_t) - \boldsymbol{G}_t, \boldsymbol{u}_{t+1} - \boldsymbol{u}_t \rangle \right]
$$

$$
= -\eta \frac{1 - \beta_1}{1 - \frac{\beta_1}{\sqrt{\beta_2}}} \mathbb{E}\left\| \frac{1}{\sqrt[4]{\widetilde{\boldsymbol{\nu}}_t}} \odot \boldsymbol{G}_t \right\|^2 + \underbrace{\mathbb{E}\left[ \left\langle \boldsymbol{G}_t, -\frac{\eta}{1 - \frac{\beta_1}{\sqrt{\beta_2}}} \left( \frac{1}{\sqrt{\boldsymbol{\nu}_t}} - \frac{1}{\sqrt{\widetilde{\boldsymbol{\nu}}_t}} \right) \odot \boldsymbol{m}_t \right\rangle \right]}_{\text{I.1}}
$$

$$
+ \underbrace{\mathbb{E}\left[ \left\langle \boldsymbol{G}_t, \beta_1 \frac{\eta}{1 - \frac{\beta_1}{\sqrt{\beta_2}}} \left( \frac{1}{\sqrt{\beta_2 \boldsymbol{\nu}_{t-1}}} - \frac{1}{\sqrt{\widetilde{\boldsymbol{\nu}}_t}} \right) \odot \boldsymbol{m}_{t-1} \right\rangle \right]}_{\text{I.2}} + \underbrace{\mathbb{E}\left[ \langle \nabla f(\boldsymbol{u}_t) - \boldsymbol{G}_t, \boldsymbol{u}_{t+1} - \boldsymbol{u}_t \rangle \right]}_{\text{I.3}}.
$$

Here we apply Lemma 7 to bound I.1. We proceed by bounding I.2 and I.3 respectively.

As for I.2, we have

$$
\text{I.2} = \mathbb{E}\left[ \left\langle \boldsymbol{G}_t, \beta_1 \frac{\eta}{1 - \frac{\beta_1}{\sqrt{\beta_2}}} \left( \frac{1}{\sqrt{\beta_2 \boldsymbol{\nu}_{t-1}}} - \frac{1}{\sqrt{\widetilde{\boldsymbol{\nu}}_t}} \right) \odot \boldsymbol{m}_{t-1} \right\rangle \right]
$$

$$
\leq \frac{\eta \beta_1}{1 - \frac{\beta_1}{\sqrt{\beta_2}}} \sum_{i=1}^d \mathbb{E}\left[ |\boldsymbol{G}_{t,i}| \left| \frac{1}{\sqrt{\beta_2 \boldsymbol{\nu}_{t-1,i}}} - \frac{1}{\sqrt{\widetilde{\boldsymbol{\nu}}_{t,i}}} \right| |\boldsymbol{m}_{t-1,i}| \right]
$$

$$
= \frac{\eta \beta_1}{1 - \frac{\beta_1}{\sqrt{\beta_2}}} \sum_{i=1}^d \mathbb{E}\left[ |\boldsymbol{G}_{t,i}| \left| \frac{(1 - \beta_2)\sigma_0^2}{\sqrt{\beta_2 \boldsymbol{\nu}_{t-1,i}} \sqrt{\widetilde{\boldsymbol{\nu}}_{t,i}}(\sqrt{\widetilde{\boldsymbol{\nu}}_{t,i}} + \sqrt{\beta_2 \boldsymbol{\nu}_{t-1,i}})} \right| |\boldsymbol{m}_{t-1,i}| \right]
$$

$$
= \frac{\eta \beta_1}{1 - \frac{\beta_1}{\sqrt{\beta_2}}} \sum_{i=1}^d \mathbb{E}\left[ |\boldsymbol{G}_{t,i}| \left| \frac{\sqrt[4]{1 - \beta_2} \sqrt{\sigma_0}}{\sqrt{\beta_2 \boldsymbol{\nu}_{t-1,i}} \sqrt[4]{\widetilde{\boldsymbol{\nu}}_{t,i}}} \right| |\boldsymbol{m}_{t-1,i}| \right]
$$

$$
\leq \frac{1}{8} \frac{1 - \beta_1}{1 - \frac{\beta_1}{\sqrt{\beta_2}}} \sum_{i=1}^d \eta \mathbb{E} \frac{|\boldsymbol{G}_{t,i}|^2}{\sqrt{\widetilde{\boldsymbol{\nu}}_{t,i}}} + \frac{2\eta \sqrt{1 - \beta_2} \beta_1^2 \sigma_0}{(1 - \beta_1)(1 - \frac{\beta_1}{\sqrt{\beta_2}})\beta_2} \sum_{i=1}^d \mathbb{E}\left[ \frac{|\boldsymbol{m}_{t-1,i}|^2}{\boldsymbol{\nu}_{t-1,i}} \right].
$$

As for I.3, we directly apply Assumption 1 and obtain

$$
\begin{aligned}
\text{I.3} =& \mathbb{E}\left[\langle \nabla f(\boldsymbol{u}_t) - \boldsymbol{G}_t, \boldsymbol{u}_{t+1} - \boldsymbol{u}_t \rangle\right] \\
\leq& \mathbb{E}\left[\|\nabla f(\boldsymbol{u}_t) - \boldsymbol{G}_t\|\|\boldsymbol{u}_{t+1} - \boldsymbol{u}_t\|\right] \\
\leq& L\mathbb{E}\left[\|\boldsymbol{u}_t - \boldsymbol{w}_t\|\|\boldsymbol{u}_{t+1} - \boldsymbol{u}_t\|\right] \\
=& L\mathbb{E}\left[\frac{\frac{\beta_1}{\sqrt{\beta_2}}}{1 - \frac{\beta_1}{\sqrt{\beta_2}}}\|\boldsymbol{w}_t - \boldsymbol{w}_{t-1}\|\left(\frac{\frac{\beta_1}{\sqrt{\beta_2}}}{1 - \frac{\beta_1}{\sqrt{\beta_2}}}\|\boldsymbol{w}_{t+1} - \boldsymbol{w}_t\| + \frac{\frac{\beta_1}{\sqrt{\beta_2}}}{1 - \frac{\beta_1}{\sqrt{\beta_2}}}\|\boldsymbol{w}_t - \boldsymbol{w}_{t-1}\|\right)\right] \\
\leq& L\mathbb{E}\left[\frac{\frac{\beta_1}{\sqrt{\beta_2}}}{1 - \frac{\beta_1}{\sqrt{\beta_2}}}\|\boldsymbol{w}_t - \boldsymbol{w}_{t-1}\|\left(\frac{1}{1 - \frac{\beta_1}{\sqrt{\beta_2}}}\|\boldsymbol{w}_{t+1} - \boldsymbol{w}_t\| + \frac{\frac{\beta_1}{\sqrt{\beta_2}}}{1 - \frac{\beta_1}{\sqrt{\beta_2}}}\|\boldsymbol{w}_t - \boldsymbol{w}_{t-1}\|\right)\right] \\
\leq& L\mathbb{E}\left[2\left(\frac{\frac{\beta_1}{\sqrt{\beta_2}}}{1 - \frac{\beta_1}{\sqrt{\beta_2}}}\right)^2\|\boldsymbol{w}_t - \boldsymbol{w}_{t-1}\|^2 + \frac{1}{4}\left(\frac{1}{1 - \frac{\beta_1}{\sqrt{\beta_2}}}\right)^2\|\boldsymbol{w}_{t+1} - \boldsymbol{w}_t\|^2\right] \\
\leq& L\mathbb{E}\left[2\left(\frac{\frac{\beta_1}{\sqrt{\beta_2}}}{1 - \frac{\beta_1}{\sqrt{\beta_2}}}\right)^2\eta^2\left\|\frac{1}{\sqrt{\boldsymbol{\nu}_{t-1}}} \odot \boldsymbol{m}_{t-1}\right\|^2 + \frac{1}{4}\left(\frac{1}{1 - \frac{\beta_1}{\sqrt{\beta_2}}}\right)^2\eta^2\left\|\frac{1}{\sqrt{\boldsymbol{\nu}_t}} \odot \boldsymbol{m}_t\right\|^2\right].
\end{aligned}
$$

All in all, we summarize that the "First Order" term can be bounded by

$$
-\frac{\eta}{4}\frac{1 - \beta_1}{1 - \frac{\beta_1}{\sqrt{\beta_2}}}\mathbb{E}\left\|\frac{1}{\sqrt[4]{\widetilde{\boldsymbol{\nu}}_t}} \odot \boldsymbol{G}_t\right\|^2 + \frac{2\eta\sqrt{1 - \beta_2}\sigma_0}{\left(1 - \frac{\beta_1^2}{\beta_2}\right)^2}\sum_{i=1}^{d}\mathbb{E}\frac{\boldsymbol{g}_{t,i}^2}{\boldsymbol{\nu}_{t,i}}
$$

$$
+\eta\frac{4(1 - \beta_1)}{(1 - \frac{\beta_1}{\sqrt{\beta_2}})^2\sqrt{\beta_2}}\sigma_1^2\sum_{i=1}^{d}\mathbb{E}\left(\frac{\boldsymbol{G}_{t-1,i}^2}{\sqrt{\beta_2\widetilde{\boldsymbol{\nu}}_{t,i}}} - \frac{\boldsymbol{G}_{t,i}^2}{\sqrt{\widetilde{\boldsymbol{\nu}}_{t+1,i}}}\right) + \sum_{i=1}^{d}\frac{2\eta\sqrt{1 - \beta_2}\sigma_0}{(1 - \beta_1)(1 - \frac{\beta_1}{\sqrt{\beta_2}})}\mathbb{E}\left[\left(\frac{|\boldsymbol{m}_{t,i}|^2}{\boldsymbol{\nu}_{t,i}}\right)\right]
$$

$$
+\frac{64(1 + \sigma_1^2)\sigma_1^2 L^2\eta^3 d}{\beta_2^2(1 - \frac{\beta_1}{\sqrt{\beta_2}})^3(1 - \beta_1)\sigma_0\sqrt{1 - \beta_2}}\mathbb{E}\left\|\frac{1}{\sqrt{\boldsymbol{\nu}_{t-1}}} \odot \boldsymbol{m}_{t-1}\right\|^2 + \frac{2\eta\sqrt{1 - \beta_2}\beta_1^2\sigma_0}{(1 - \beta_1)(1 - \frac{\beta_1}{\sqrt{\beta_2}})\beta_2}\sum_{i=1}^{d}\mathbb{E}\left[\frac{|\boldsymbol{m}_{t-1,i}|^2}{\boldsymbol{\nu}_{t-1,i}}\right]
$$

$$
+ L\mathbb{E}\left[2\left(\frac{\frac{\beta_1}{\sqrt{\beta_2}}}{1 - \frac{\beta_1}{\sqrt{\beta_2}}}\right)^2\eta^2\left\|\frac{1}{\sqrt{\boldsymbol{\nu}_{t-1}}} \odot \boldsymbol{m}_{t-1}\right\|^2 + \frac{1}{4}\left(\frac{1}{1 - \frac{\beta_1}{\sqrt{\beta_2}}}\right)^2\eta^2\left\|\frac{1}{\sqrt{\boldsymbol{\nu}_t}} \odot \boldsymbol{m}_t\right\|^2\right].
$$

Furthermore, the "Second Order" term can be directly bounded by

$$
\begin{aligned}
\frac{L}{2}\mathbb{E}\|\boldsymbol{u}_{t+1} - \boldsymbol{u}_t\|^2 =& \frac{L}{2}\left\|\frac{\boldsymbol{w}_{t+1} - \boldsymbol{w}_t}{1 - \frac{\beta_1}{\sqrt{\beta_2}}} - \frac{\beta_1}{\sqrt{\beta_2}}\frac{\boldsymbol{w}_t - \boldsymbol{w}_{t-1}}{1 - \frac{\beta_1}{\sqrt{\beta_2}}}\right\|^2 \\
\leq& 2L\mathbb{E}\left\|\frac{\boldsymbol{w}_{t+1} - \boldsymbol{w}_t}{1 - \frac{\beta_1}{\sqrt{\beta_2}}}\right\|^2 + 2L\mathbb{E}\left\|\frac{\beta_1}{\sqrt{\beta_2}}\frac{\boldsymbol{w}_t - \boldsymbol{w}_{t-1}}{1 - \frac{\beta_1}{\sqrt{\beta_2}}}\right\|^2.
\end{aligned}
$$

Applying the estimations of the first-order term and the second-order term to the descent lemma then gives

$$
\mathbb{E}f(\boldsymbol{u}_{t+1})
$$

$$
\leq \mathbb{E}f(\boldsymbol{u}_t) - \frac{\eta}{4}\frac{1-\beta_1}{1-\frac{\beta_1}{\sqrt{\beta_2}}}\sum_{i=1}^{d}\mathbb{E}\frac{\boldsymbol{G}_{t,i}^2}{\sqrt{\widetilde{\boldsymbol{\nu}}_{t,i}}} + \frac{2\eta\sqrt{1-\beta_2}\sigma_0}{\left(1-\frac{\beta_1^2}{\beta_2}\right)^2}\sum_{i=1}^{d}\mathbb{E}\frac{\boldsymbol{g}_{t,i}^2}{\boldsymbol{\nu}_{t,i}}
$$

$$
+ \eta\frac{4}{(1-\frac{\beta_1}{\sqrt{\beta_2}})^2\sqrt{\beta_2}}\sigma_1^2\sum_{i=1}^{d}\mathbb{E}\left(\frac{\boldsymbol{G}_{t-1,i}^2}{\sqrt{\beta_2\widetilde{\boldsymbol{\nu}}_{t,i}}} - \frac{\boldsymbol{G}_{t,i}^2}{\sqrt{\widetilde{\boldsymbol{\nu}}_{t+1,i}}}\right) + \sum_{i=1}^{d}\frac{2\eta\sqrt{1-\beta_2}\sigma_0}{(1-\beta_1)(1-\frac{\beta_1}{\sqrt{\beta_2}})}\mathbb{E}\left[\left(\frac{|\boldsymbol{m}_{t,i}|^2}{\boldsymbol{\nu}_{t,i}}\right)\right]
$$

$$
+ \frac{64(1+\sigma_1^2)\sigma_1^2 L^2\eta^3 d}{\beta_2^2(1-\frac{\beta_1}{\sqrt{\beta_2}})^3(1-\beta_1)\sigma_0\sqrt{1-\beta_2}}\mathbb{E}\left\|\frac{1}{\sqrt{\boldsymbol{\nu}_{t-1}}}\odot\boldsymbol{m}_{t-1}\right\|^2 + \frac{2\eta\sqrt{1-\beta_2}\beta_1^2\sigma_0}{(1-\beta_1)(1-\frac{\beta_1}{\sqrt{\beta_2}})\beta_2}\sum_{i=1}^{d}\mathbb{E}\left[\frac{|\boldsymbol{m}_{t-1,i}|^2}{\boldsymbol{\nu}_{t-1,i}}\right]
$$

$$
+ L\mathbb{E}\left[4\left(\frac{\frac{\beta_1}{\sqrt{\beta_2}}}{1-\frac{\beta_1}{\sqrt{\beta_2}}}\right)^2\eta^2\left\|\frac{1}{\sqrt{\boldsymbol{\nu}_{t-1}}}\odot\boldsymbol{m}_{t-1}\right\|^2 + 3\left(\frac{1}{1-\frac{\beta_1}{\sqrt{\beta_2}}}\right)^2\eta^2\left\|\frac{1}{\sqrt{\boldsymbol{\nu}_t}}\odot\boldsymbol{m}_t\right\|^2\right].
$$

The proof is completed. $\qquad\square$

**Lemma 9** (Lemma 3, restated). *Let all conditions in Theorem 1 hold. Then,*

$$
\sum_{t=1}^{T+1}\sum_{i=1}^{d}\mathbb{E}\sqrt{\widetilde{\boldsymbol{\nu}}_{t,i}} \leq 2(T+1)\sum_{i=1}^{d}\sqrt{\boldsymbol{\nu}_{0,i}+\sigma_0^2} + 24d\frac{\sigma_1^2 C_1}{\sqrt{\beta_2}}\ln d\frac{\sigma_1^2 C_1}{\sqrt{\beta_2}} + \frac{12\sigma_1^2}{\sqrt{\beta_2}}C_2.
$$

*Proof of Lemma 3.* To begin with, we have that

$$
\sum_{t=1}^{T}\mathbb{E}\left[\frac{|\boldsymbol{G}_{t,i}|^2}{\sqrt{\widetilde{\boldsymbol{\nu}}_{t,i}}}\mathbf{1}_{|G_{t,i}|\geq\frac{\sigma_0}{\sigma_1}}\right] \leq \sum_{t=1}^{T}\mathbb{E}\left[\frac{|\boldsymbol{G}_{t,i}|^2}{\sqrt{\widetilde{\boldsymbol{\nu}}_{t,i}}}\right]. \tag{14}
$$

On the other hand, we have that

$$
\frac{|\boldsymbol{G}_{t,i}|^2}{\sqrt{\widetilde{\boldsymbol{\nu}}_{t,i}}}\mathbf{1}_{|G_{t,i}|\geq\frac{\sigma_0}{\sigma_1}} \geq \frac{\frac{2}{3}|\boldsymbol{G}_{t,i}|^2+\frac{1}{3}\frac{\sigma_0^2}{\sigma_1^2}}{\sqrt{\widetilde{\boldsymbol{\nu}}_{t,i}}}\mathbf{1}_{|G_{t,i}|\geq\frac{\sigma_0}{\sigma_1}} \geq \frac{\frac{\beta_2}{3\sigma_1^2}\mathbb{E}^{|\mathcal{F}_t}|\boldsymbol{g}_{t,i}|^2+\frac{1-\beta_2}{3}\frac{\sigma_0^2}{\sigma_1^2}}{\sqrt{\widetilde{\boldsymbol{\nu}}_{t,i}}}\mathbf{1}_{|G_{t,i}|\geq\frac{\sigma_0}{\sigma_1}}
$$

$$
= \mathbb{E}^{|\mathcal{F}_t}\frac{\frac{\beta_2}{3\sigma_1^2}|\boldsymbol{g}_{t,i}|^2+\frac{1-\beta_2}{3\sigma_1^2}\sigma_0^2}{\sqrt{\widetilde{\boldsymbol{\nu}}_{t,i}}}\mathbf{1}_{|G_{t,i}|\geq\frac{\sigma_0}{\sigma_1}} \geq \sqrt{\beta_2}\mathbb{E}^{|\mathcal{F}_t}\frac{\frac{\beta_2}{3\sigma_1^2}|\boldsymbol{g}_{t,i}|^2+\frac{1-\beta_2}{3\sigma_1^2}\sigma_0^2}{\sqrt{\widetilde{\boldsymbol{\nu}}_{t+1,i}}+\sqrt{\beta_2\widetilde{\boldsymbol{\nu}}_{t,i}}}\mathbf{1}_{|G_{t,i}|\geq\frac{\sigma_0}{\sigma_1}}.
$$

As a conclusion,

$$
\sum_{t=1}^{T}\mathbb{E}\left[\frac{|\boldsymbol{G}_{t,i}|^2}{\sqrt{\widetilde{\boldsymbol{\nu}}_{t,i}}}\mathbf{1}_{|G_{t,i}|\geq\frac{\sigma_0}{\sigma_1}}\right] \geq \sqrt{\beta_2}\sum_{t=1}^{T}\mathbb{E}\left[\frac{\frac{\beta_2}{3\sigma_1^2}|\boldsymbol{g}_{t,i}|^2+\frac{1-\beta_2}{3\sigma_1^2}\sigma_0^2}{\sqrt{\widetilde{\boldsymbol{\nu}}_{t+1,i}}+\sqrt{\beta_2\widetilde{\boldsymbol{\nu}}_{t,i}}}\mathbf{1}_{|G_{t,i}|\geq\frac{\sigma_0}{\sigma_1}}\right]
$$

$$
\geq \frac{\sqrt{\beta_2}}{3(1-\beta_2)\sigma_1^2}\sum_{t=1}^{T}\mathbb{E}\left[\left(\sqrt{\widetilde{\boldsymbol{\nu}}_{t+1,i}}-\sqrt{\beta_2\widetilde{\boldsymbol{\nu}}_{t,i}}\right)\mathbf{1}_{|G_{t,i}|\geq\frac{\sigma_0}{\sigma_1}}\right].
$$

On the other hand, as stated in Section 4.2, we define $\{\bar{\nu}_{t,i}\}_{t=0}^{\infty}$ as $\bar{\nu}_{0,i} = \nu_{0,i}$, $\bar{\nu}_{t,i} = \beta_2 \bar{\nu}_{t-1,i} + (1-\beta_2)|g_{t,i}|^2 \mathbf{1}_{|G_{t,i}|<\frac{\sigma_0^2}{\sigma_1^2}}$. One can easily observe that $\bar{\nu}_{t,i} \leq \nu_{t,i}$, and thus

$$\sum_{t=1}^{T} \mathbb{E}\left[\left(\sqrt{\widetilde{\nu}_{t+1,i}} - \sqrt{\beta_2 \widetilde{\nu}_{t,i}}\right) \mathbf{1}_{|G_{t,i}|<\frac{\sigma_0^2}{\sigma_1^2}}\right]$$

$$= \sum_{t=1}^{T} \mathbb{E}\left(\sqrt{\beta_2^2 \nu_{t-1,i} + \beta_2(1-\beta_2)|g_{t,i}|^2 + (1-\beta_2)\sigma_0^2} - \sqrt{\beta_2(\beta_2 \nu_{t-1,i} + (1-\beta_2)\sigma_0^2)}\right) \mathbf{1}_{|G_{t,i}|<\frac{\sigma_0^2}{\sigma_1^2}}$$

$$\leq \sum_{t=1}^{T} \mathbb{E}\left(\sqrt{\beta_2^2 \bar{\nu}_{t-1,i} + \beta_2(1-\beta_2)|g_{t,i}|^2 + (1-\beta_2)\sigma_0^2} - \sqrt{\beta_2(\beta_2 \bar{\nu}_{t-1,i} + (1-\beta_2)\sigma_0^2)}\right) \mathbf{1}_{|G_{t,i}|<\frac{\sigma_0^2}{\sigma_1^2}}$$

$$\leq \sum_{t=1}^{T} \mathbb{E}\left(\sqrt{\beta_2^2 \bar{\nu}_{t-1,i} + \beta_2(1-\beta_2)|g_{t,i}|^2 \mathbf{1}_{|G_{t,i}|<\frac{\sigma_0^2}{\sigma_1^2}} + (1-\beta_2)\sigma_0^2} - \sqrt{\beta_2(\beta_2 \bar{\nu}_{t-1,i} + (1-\beta_2)\sigma_0^2)}\right)$$

$$= \sum_{t=1}^{T} \mathbb{E}\left(\sqrt{\beta_2 \bar{\nu}_{t,i} + (1-\beta_2)\sigma_0^2} - \sqrt{\beta_2(\beta_2 \bar{\nu}_{t-1,i} + (1-\beta_2)\sigma_0^2)}\right)$$

$$= \mathbb{E}\sqrt{\beta_2 \bar{\nu}_{t,i} + (1-\beta_2)\sigma_0^2} + (1 - \sqrt{\beta_2}) \sum_{t=1}^{T-1} \mathbb{E}\sqrt{\beta_2 \bar{\nu}_{t,i} + (1-\beta_2)\sigma_0^2} - \mathbb{E}\sqrt{\beta_2(\beta_2 \bar{\nu}_{0,i} + (1-\beta_2)\sigma_0^2)}.$$

All in all, summing the above two inequalities together, we obtain that

$$\mathbb{E}\sqrt{\widetilde{\nu}_{t+1,i}} + (1 - \sqrt{\beta_2}) \sum_{t=2}^{T} \mathbb{E}\sqrt{\widetilde{\nu}_{t,i}} - \sqrt{\beta_2 \widetilde{\nu}_{1,i}}$$

$$= \sum_{t=1}^{T} \mathbb{E}\left(\sqrt{\widetilde{\nu}_{t,i}} - \sqrt{\beta_2 \widetilde{\nu}_{t-1,i}}\right)$$

$$\leq \sum_{t=1}^{T} \mathbb{E}\left(\sqrt{\widetilde{\nu}_{t,i}} - \sqrt{\beta_2 \widetilde{\nu}_{t-1,i}}\right) \mathbf{1}_{|G_{t,i}|\geq\frac{\sigma_0}{\sigma_1}} + \sum_{t=1}^{T} \mathbb{E}\left(\sqrt{\widetilde{\nu}_{t,i}} - \sqrt{\beta_2 \widetilde{\nu}_{t-1,i}}\right) \mathbf{1}_{|G_{t,i}|<\frac{\sigma_0^2}{\sigma_1^2}}$$

$$\leq \frac{3(1-\beta_2)\sigma_1^2}{\sqrt{\beta_2}} \sum_{t=1}^{T} \mathbb{E}\left[\frac{|G_{t,i}|^2}{\sqrt{\widetilde{\nu}_{t,i}}}\right] + \mathbb{E}\sqrt{\beta_2 \bar{\nu}_{t,i} + (1-\beta_2)\sigma_0^2} + (1 - \sqrt{\beta_2}) \sum_{t=1}^{T-1} \mathbb{E}\sqrt{\beta_2 \bar{\nu}_{t,i} + (1-\beta_2)\sigma_0^2} - \sqrt{\beta_2(\beta_2 \bar{\nu}_{0,i} + (1-\beta_2)\sigma_0^2)}.$$

Since $\forall t$,

$$\mathbb{E}\sqrt{\beta_2 \bar{\nu}_{t,i} + (1-\beta_2)\sigma_0^2} \leq \sqrt{\beta_2 \mathbb{E}\bar{\nu}_{t,i} + (1-\beta_2)\sigma_0^2} \leq \sqrt{\sigma_0^2 + \nu_{0,i}},$$

combining with $\sqrt{\beta_2 \widetilde{\nu}_{1,i}} = \sqrt{\beta_2(\beta_2 \bar{\nu}_{0,i} + (1-\beta_2)\sigma_0^2)}$ and $\mathbb{E}\sqrt{\widetilde{\nu}_{t+1,i}} = \mathbb{E}\sqrt{\beta_2 \nu_{t,i} + (1-\beta_2)\sigma_0^2} \geq \mathbb{E}\sqrt{\beta_2 \bar{\nu}_{t,i} + (1-\beta_2)\sigma_0^2}$, we obtain

$$(1 - \sqrt{\beta_2}) \sum_{t=2}^{T+1} \mathbb{E}\sqrt{\widetilde{\nu}_{t,i}} \leq \frac{3(1-\beta_2)\sigma_1^2}{\sqrt{\beta_2}} \sum_{t=2}^{T} \mathbb{E}\left[\frac{|G_{t,i}|^2}{\sqrt{\widetilde{\nu}_{t,i}}}\right] + +(1 - \sqrt{\beta_2}) \sum_{t=1}^{T} \mathbb{E}\sqrt{\beta_2 \bar{\nu}_{t,i} + (1-\beta_2)\sigma_0^2}$$

$$\leq \frac{3(1-\beta_2)\sigma_1^2}{\sqrt{\beta_2}} \sum_{t=2}^{T} \mathbb{E}\left[\frac{|G_{t,i}|^2}{\sqrt{\widetilde{\nu}_{t,i}}}\right] + (1 - \sqrt{\beta_2})T\sqrt{\sigma_0^2 + \nu_{0,i}}. \tag{15}$$

Leveraging Eq. (16), we then obtain that

$$
\sum_{t=1}^{T+1} \sum_{i=1}^{d} \mathbb{E}\sqrt{\widetilde{\boldsymbol{\nu}}_{t,i}}
$$

$$
\leq 3\frac{(1+\sqrt{\beta_2})\sigma_1^2}{\sqrt{\beta_2}} \sum_{t=1}^{T} \mathbb{E}\left[\frac{|\boldsymbol{G}_{t,i}|^2}{\sqrt{\widetilde{\boldsymbol{\nu}}_{t,i}}}\right] + (T+1) \sum_{i=1}^{d} \sqrt{\boldsymbol{\nu}_{0,i} + \sigma_0^2}
$$

$$
\leq \frac{6\sigma_1^2}{\sqrt{\beta_2}} \left( \frac{1-\frac{\beta_1}{\sqrt{\beta_2}}}{1-\beta_1} \frac{8}{\eta} f(\boldsymbol{u}_1) + \frac{32}{\beta_2\left(1-\frac{\beta_1}{\sqrt{\beta_2}}\right)^2} \sum_{i=1}^{d} \mathbb{E}\frac{\boldsymbol{G}_{1,i}^2}{\sqrt{\widetilde{\boldsymbol{\nu}}_{1,i}}} + C_1\mathbb{E}\sum_{i=1}^{d} \left( \ln\left(\frac{\boldsymbol{\nu}_{T,i}}{\boldsymbol{\nu}_{0,i}}\right) - T\ln\beta_2 \right) \right)
$$

$$
+ (T+1)\sum_{i=1}^{d} \sqrt{\boldsymbol{\nu}_{0,i} + \sigma_0^2}
$$

$$
\leq \frac{6\sigma_1^2}{\sqrt{\beta_2}} \left( \frac{1-\frac{\beta_1}{\sqrt{\beta_2}}}{1-\beta_1} \frac{8}{\eta} f(\boldsymbol{u}_1) + \frac{32}{\beta_2\left(1-\frac{\beta_1}{\sqrt{\beta_2}}\right)^2} \sum_{i=1}^{d} \mathbb{E}\frac{\boldsymbol{G}_{1,i}^2}{\sqrt{\widetilde{\boldsymbol{\nu}}_{1,i}}} + 2C_1\mathbb{E}\sum_{i=1}^{d} \left( \ln\left(\frac{\sum_{t=1}^{T+1}\sqrt{\widetilde{\boldsymbol{\nu}}_{t,i}}}{\sqrt{\beta_2}\boldsymbol{\nu}_{0,i}}\right) - T\ln\beta_2 \right) \right)
$$

$$
+ (T+1)\sum_{i=1}^{d} \sqrt{\boldsymbol{\nu}_{0,i} + \sigma_0^2}
$$

$$
\leq \frac{6\sigma_1^2}{\sqrt{\beta_2}} \left( \frac{1-\frac{\beta_1}{\sqrt{\beta_2}}}{1-\beta_1} \frac{8}{\eta} f(\boldsymbol{u}_1) + \frac{32}{\beta_2\left(1-\frac{\beta_1}{\sqrt{\beta_2}}\right)^2} \sum_{i=1}^{d} \mathbb{E}\frac{\boldsymbol{G}_{1,i}^2}{\sqrt{\widetilde{\boldsymbol{\nu}}_{1,i}}} + 2C_1 \sum_{i=1}^{d} \left( \ln\left(\frac{\mathbb{E}\sum_{t=1}^{T+1}\sum_{j=1}^{d}\sqrt{\widetilde{\boldsymbol{\nu}}_{t,j}}}{\sqrt{\beta_2}\boldsymbol{\nu}_{0,i}}\right) - T\ln\beta_2 \right) \right)
$$

$$
+ (T+1)\sum_{i=1}^{d} \sqrt{\boldsymbol{\nu}_{0,i} + \sigma_0^2},
$$

where in the last inequality we use the concavity of $h(x) = \ln x$. Solving the above inequality with respect to $\sum_{t=1}^{T+1} \sum_{i=1}^{d} \mathbb{E}\sqrt{\widetilde{\boldsymbol{\nu}}_{t,i}}$ then gives

$$
\sum_{t=1}^{T+1} \sum_{i=1}^{d} \mathbb{E}\sqrt{\widetilde{\boldsymbol{\nu}}_{t,i}} \leq 2(T+1)\sum_{i=1}^{d} \sqrt{\boldsymbol{\nu}_{0,i} + \sigma_0^2} + 24d\frac{\sigma_1^2 C_1}{\sqrt{\beta_2}} \ln d\frac{\sigma_1^2 C_1}{\sqrt{\beta_2}}
$$

$$
+ \frac{12\sigma_1^2}{\sqrt{\beta_2}} \left( \frac{1-\frac{\beta_1}{\sqrt{\beta_2}}}{1-\beta_1} \frac{8}{\eta} f(\boldsymbol{u}_1) + \frac{32}{\beta_2\left(1-\frac{\beta_1}{\sqrt{\beta_2}}\right)^2} \sum_{i=1}^{d} \mathbb{E}\frac{\boldsymbol{G}_{1,i}^2}{\sqrt{\widetilde{\boldsymbol{\nu}}_{1,i}}} + 2C_1 \sum_{i=1}^{d} \left( \ln\left(\frac{1}{\sqrt{\beta_2}\boldsymbol{\nu}_{0,i}}\right) - T\ln\beta_2 \right) \right).
$$

The proof is then completed by applying the definition of $C_2$.

$\square$

## C.2  Proof of Theorem 1

*Proof of Theorem 1.*  Summing the inequality in Lemma 8 over $t$ from $1$ to $T$ and collecting the terms, we obtain

$$\mathbb{E}f(\boldsymbol{u}_{T+1})$$

$$\leq f(\boldsymbol{u}_1) - \frac{\eta}{4}\frac{1-\beta_1}{1-\frac{\beta_1}{\sqrt{\beta_2}}}\sum_{t=1}^{T}\sum_{i=1}^{d}\mathbb{E}\frac{\boldsymbol{G}_{t,i}^2}{\sqrt{\widetilde{\boldsymbol{\nu}}_{t,i}}} + \eta\frac{4(1-\beta_1)}{(1-\frac{\beta_1}{\sqrt{\beta_2}})^2\sqrt{\beta_2}}\sigma_1^2\sum_{t=1}^{T}\sum_{i=1}^{d}\mathbb{E}\left(\frac{\boldsymbol{G}_{t-1,i}^2}{\sqrt{\beta_2\widetilde{\boldsymbol{\nu}}_{t,i}}} - \frac{\boldsymbol{G}_{t,i}^2}{\sqrt{\widetilde{\boldsymbol{\nu}}_{t+1,i}}}\right)$$

$$+ \tilde{C}\sum_{t=1}^{T}\mathbb{E}\left\|\frac{1}{\sqrt{\boldsymbol{\nu}_t}}\odot\boldsymbol{m}_t\right\|^2$$

$$\leq f(\boldsymbol{u}_1) - \frac{\eta}{4}\frac{1-\beta_1}{1-\frac{\beta_1}{\sqrt{\beta_2}}}\sum_{t=1}^{T}\sum_{i=1}^{d}\mathbb{E}\frac{\boldsymbol{G}_{t,i}^2}{\sqrt{\widetilde{\boldsymbol{\nu}}_{t,i}}} + \eta\frac{4(1-\beta_1)}{(1-\frac{\beta_1}{\sqrt{\beta_2}})^2\sqrt{\beta_2}}\sigma_1^2\sum_{i=1}^{d}\mathbb{E}\left(\frac{\boldsymbol{G}_{1,i}^2}{\sqrt{\beta_2\widetilde{\boldsymbol{\nu}}_{1,i}}}\right)$$

$$+ \left(\frac{1}{\beta_2}-1\right)\sum_{t=1}^{T-1}\mathbb{E}\frac{\boldsymbol{G}_{t,i}^2}{\sqrt{\beta_2\widetilde{\boldsymbol{\nu}}_{t+1,i}}} + \tilde{C}\sum_{t=1}^{T}\mathbb{E}\left\|\frac{1}{\sqrt{\boldsymbol{\nu}_t}}\odot\boldsymbol{m}_t\right\|^2$$

$$\overset{(*)}{\leq} f(\boldsymbol{u}_1) - \frac{\eta}{4}\frac{1-\beta_1}{1-\frac{\beta_1}{\sqrt{\beta_2}}}\sum_{t=1}^{T}\sum_{i=1}^{d}\mathbb{E}\frac{\boldsymbol{G}_{t,i}^2}{\sqrt{\widetilde{\boldsymbol{\nu}}_{t,i}}} + \eta\frac{4(1-\beta_1)}{(1-\frac{\beta_1}{\sqrt{\beta_2}})^2\sqrt{\beta_2}}\sigma_1^2\sum_{i=1}^{d}\mathbb{E}\frac{\boldsymbol{G}_{1,i}^2}{\sqrt{\beta_2\widetilde{\boldsymbol{\nu}}_{1,i}}}$$

$$+ \frac{\eta}{8}\frac{1-\beta_1}{1-\frac{\beta_1}{\sqrt{\beta_2}}}\sum_{t=1}^{T-1}\mathbb{E}\frac{\boldsymbol{G}_{t,i}^2}{\sqrt{\widetilde{\boldsymbol{\nu}}_{t,i}}} + \tilde{C}\sum_{t=1}^{T}\mathbb{E}\left\|\frac{1}{\sqrt{\boldsymbol{\nu}_t}}\odot\boldsymbol{m}_t\right\|^2$$

$$\overset{(\circ)}{\leq} f(\boldsymbol{u}_1) - \frac{\eta}{8}\frac{1-\beta_1}{1-\frac{\beta_1}{\sqrt{\beta_2}}}\sum_{t=1}^{T}\sum_{i=1}^{d}\mathbb{E}\frac{\boldsymbol{G}_{t,i}^2}{\sqrt{\widetilde{\boldsymbol{\nu}}_{t,i}}} + \eta\frac{4(1-\beta_1)}{(1-\frac{\beta_1}{\sqrt{\beta_2}})^2\sqrt{\beta_2}}\sigma_1^2\sum_{i=1}^{d}\mathbb{E}\frac{\boldsymbol{G}_{1,i}^2}{\sqrt{\beta_2\widetilde{\boldsymbol{\nu}}_{1,i}}}$$

$$+ \tilde{C}\frac{(1-\beta_1)^2}{(1-\frac{\beta_1}{\sqrt{\beta_2}})^2(1-\beta_2)}\sum_{i=1}^{d}\left(\ln\left(\frac{\boldsymbol{\nu}_{T,i}}{\boldsymbol{\nu}_{0,i}}\right) - T\ln\beta_2\right),$$

where we define

$$\tilde{C} \triangleq 4L\eta^2\left(\frac{1+\frac{\beta_1}{\sqrt{\beta_2}}}{1-\frac{\beta_1}{\sqrt{\beta_2}}}\right)^2 + \frac{2\eta\sqrt{1-\beta_2}\beta_1^2\sigma_0}{(1-\beta_1)(1-\frac{\beta_1}{\sqrt{\beta_2}})\beta_2} + \frac{64(1+\sigma_1^2)\sigma_1^2L^2\eta^3 d}{\beta_2^2(1-\frac{\beta_1}{\sqrt{\beta_2}})^3(1-\beta_1)\sigma_0\sqrt{1-\beta_2}}.$$

to simplify the notations, inequality $(*)$ is due to that $\tilde{\boldsymbol{\nu}}_{t+1,i} \geq \beta_2\tilde{\boldsymbol{\nu}}_{t+1,i}$ and $\beta_1 \leq \sqrt{\beta_2} - 8\sigma_1^2(1-\beta_2)\beta_2^{-2}$, and inequality $(\circ)$ is due to Lemma 5. Simple rearrangement of the above inequality then gives

$$\sum_{t=1}^{T}\mathbb{E}\left[\left\|\frac{1}{\sqrt[4]{\widetilde{\boldsymbol{\nu}}_t}}\odot\boldsymbol{G}_t\right\|^2\right] \leq \frac{1-\frac{\beta_1}{\sqrt{\beta_2}}}{1-\beta_1}\frac{8}{\eta}f(\boldsymbol{u}_1) + \frac{32}{\beta_2\left(1-\frac{\beta_1}{\sqrt{\beta_2}}\right)^2}\sum_{i=1}^{d}\mathbb{E}\frac{\boldsymbol{G}_{1,i}^2}{\sqrt{\widetilde{\boldsymbol{\nu}}_{1,i}}}$$

$$+ C_1\sum_{i=1}^{d}\mathbb{E}\left(\ln\left(\frac{\boldsymbol{\nu}_{T,i}}{\boldsymbol{\nu}_{0,i}}\right) - T\ln\beta_2\right). \tag{16}$$

Then, according to Cauchy's inequality, we have

$$\left(\mathbb{E}\sum_{t=1}^{T}\|\boldsymbol{G}_t\|_1\right)^2 \leq \left(\sum_{t=1}^{T}\mathbb{E}\left[\left\|\frac{1}{\sqrt[4]{\widetilde{\boldsymbol{\nu}}_t^1}}\odot\boldsymbol{G}_t\right\|^2\right]\right)\left(\sum_{t=1}^{T}\mathbb{E}\left[\left\|\sqrt[4]{\widetilde{\boldsymbol{\nu}}_t^1}\right\|^2\right]\right). \tag{17}$$

Meanwhile, by Lemma 3, we have

$$\sum_{t=1}^{T}\sum_{i=1}^{d}\mathbb{E}\sqrt{\widetilde{\boldsymbol{\nu}}_{t,i}} \leq 2(T+1)\sum_{i=1}^{d}\sqrt{\boldsymbol{\nu}_{0,i}+\sigma_0^2} + 24d\frac{\sigma_1^2C_1}{\sqrt{\beta_2}}\ln d\frac{\sigma_1^2C_1}{\sqrt{\beta_2}} + \frac{12\sigma_1^2}{\sqrt{\beta_2}}C_2.$$

Combining the above inequality and Eq. (17) gives

$$\left(\mathbb{E}\sum_{t=1}^{T}\|\boldsymbol{G}_t\|_1\right)^2$$

$$\leq\left(\frac{1-\frac{\beta_1}{\sqrt{\beta_2}}}{1-\beta_1}\frac{8}{\eta}f(\boldsymbol{u}_1)+\frac{32}{\beta_2\left(1-\frac{\beta_1}{\sqrt{\beta_2}}\right)^2}\sum_{i=1}^{d}\mathbb{E}\frac{\boldsymbol{G}_{1,i}^2}{\sqrt{\widetilde{\boldsymbol{\nu}}_{1,i}}}+C_1\sum_{i=1}^{d}\left(\ln\left(\frac{\boldsymbol{\nu}_{T,i}}{\boldsymbol{\nu}_{0,i}}\right)-T\ln\beta_2\right)\right)$$

$$\times\left(2(T+1)\sum_{i=1}^{d}\sqrt{\boldsymbol{\nu}_{0,i}+\sigma_0^2}+24d\frac{\sigma_1^2C_1}{\sqrt{\beta_2}}\ln d\frac{\sigma_1^2C_1}{\sqrt{\beta_2}}+\frac{12\sigma_1^2}{\sqrt{\beta_2}}C_2\right)$$

$$\leq\left(C_2+2C_1\sum_{i=1}^{d}\left(\ln\left(\sum_{t=1}^{T}\sum_{i=1}^{d}\mathbb{E}\sqrt{\widetilde{\boldsymbol{\nu}}_{t,i}}\right)\right)\right)\times\left(2(T+1)\sum_{i=1}^{d}\sqrt{\boldsymbol{\nu}_{0,i}+\sigma_0^2}+24d\frac{\sigma_1^2C_1}{\sqrt{\beta_2}}\ln d\frac{\sigma_1^2C_1}{\sqrt{\beta_2}}+\frac{12\sigma_1^2}{\sqrt{\beta_2}}C_2\right)$$

$$\leq\left(C_2+2C_1\sum_{i=1}^{d}\left(\ln\left(2(T+1)\sum_{i=1}^{d}\sqrt{\boldsymbol{\nu}_{0,i}+\sigma_0^2}+24d\frac{\sigma_1^2C_1}{\sqrt{\beta_2}}\ln d\frac{\sigma_1^2C_1}{\sqrt{\beta_2}}+\frac{12\sigma_1^2}{\sqrt{\beta_2}}C_2\right)\right)\right)$$

$$\times\left(2(T+1)\sum_{i=1}^{d}\sqrt{\boldsymbol{\nu}_{0,i}+\sigma_0^2}+24d\frac{\sigma_1^2C_1}{\sqrt{\beta_2}}\ln d\frac{\sigma_1^2C_1}{\sqrt{\beta_2}}+\frac{12\sigma_1^2}{\sqrt{\beta_2}}C_2\right).$$

The proof is then completed.

$\square$

# D   Proof of Theorem 2

*Proof.* To start with, we have that

$$\frac{|\boldsymbol{G}_{t,i}|^2}{\sqrt{\widetilde{\boldsymbol{\nu}}_{t,i}}}\mathbf{1}_{|G_{t,i}|\geq\frac{\sigma_0}{\sigma_1}}\geq\frac{\frac{1}{2\sigma_1^2}\mathbb{E}^{|\mathcal{F}_t}|\boldsymbol{g}_{t,i}|^2}{\sqrt{\widetilde{\boldsymbol{\nu}}_{t,i}}}\mathbf{1}_{|G_{t,i}|\geq\frac{\sigma_0}{\sigma_1}}$$

$$=\frac{\frac{1}{2\sigma_1^2}\mathbb{E}^{|\mathcal{F}_t}|\boldsymbol{g}_{t,i}|^2}{\sqrt{\beta_2\boldsymbol{\nu}_{t-1,i}+(1-\beta_2)\sigma_0^2}}\mathbf{1}_{|G_{t,i}|\geq\frac{\sigma_0}{\sigma_1}}$$

$$\geq\frac{1}{2\sigma_1^2\sqrt{1-\beta_2}}\mathbb{E}^{|\mathcal{F}_t}\frac{|\boldsymbol{g}_{t,i}|^2}{\sqrt{\frac{\boldsymbol{\nu}_{0,i}}{1-\beta_2}+\sum_{s=1}^{T}|g_{s,i}|^2+\sigma_0^2}}\mathbf{1}_{|G_{t,i}|\geq\frac{\sigma_0}{\sigma_1}},$$

where the last inequality is due to that

$$\beta_2\boldsymbol{\nu}_{t-1,i}=(1-\beta_2)\sum_{s=1}^{t-1}\beta_2^{t-s}|\boldsymbol{g}_{s,i}|^2+\beta_2^t\boldsymbol{\nu}_{0,i}\leq(1-\beta_2)\sum_{s=1}^{T}|\boldsymbol{g}_{s,i}|^2+\boldsymbol{\nu}_{0,i}.\qquad(18)$$

Furthermore, we have

$$\frac{\sigma_0^2+\frac{\boldsymbol{\nu}_{0,i}}{1-\beta_2}}{\sqrt{\frac{\boldsymbol{\nu}_{0,i}}{1-\beta_2}+\sum_{s=1}^{T}|g_{s,i}|^2+\sigma_0^2}}+\sum_{t=1}^{T}\mathbb{E}\frac{|\boldsymbol{g}_{t,i}|^2}{\sqrt{\frac{\boldsymbol{\nu}_{0,i}}{1-\beta_2}+\sum_{s=1}^{T}|g_{s,i}|^2+\sigma_0^2}}\mathbf{1}_{|G_{t,i}|<\frac{\sigma_0}{\sigma_1}}$$

$$\leq\frac{\sigma_0^2+\frac{\boldsymbol{\nu}_{0,i}}{1-\beta_2}}{\sqrt{\frac{\boldsymbol{\nu}_{0,i}}{1-\beta_2}+\sum_{s=1}^{T}|g_{s,i}|^2\mathbf{1}_{|\boldsymbol{G}_{s,i}|<\frac{\sigma_0}{\sigma_1}}+\sigma_0^2}}+\sum_{t=1}^{T}\mathbb{E}\frac{|\boldsymbol{g}_{t,i}|^2}{\sqrt{\frac{\boldsymbol{\nu}_{0,i}}{1-\beta_2}+\sum_{s=1}^{T}|g_{s,i}|^2\mathbf{1}_{|\boldsymbol{G}_{s,i}|<\frac{\sigma_0}{\sigma_1}}+\sigma_0^2}}\mathbf{1}_{|G_{t,i}|<\frac{\sigma_0}{\sigma_1}}$$

$$=\mathbb{E}\sqrt{\frac{\boldsymbol{\nu}_{0,i}}{1-\beta_2}+\sum_{s=1}^{T}|g_{s,i}|^2\mathbf{1}_{|\boldsymbol{G}_{s,i}|<\frac{\sigma_0}{\sigma_1}}+\sigma_0^2}\leq\sqrt{\frac{\boldsymbol{\nu}_{0,i}}{1-\beta_2}+\mathbb{E}\sum_{s=1}^{T}|g_{s,i}|^2\mathbf{1}_{|\boldsymbol{G}_{s,i}|<\frac{\sigma_0}{\sigma_1}}+\sigma_0^2}$$

$$\leq\sqrt{\frac{\boldsymbol{\nu}_{0,i}}{1-\beta_2}+2\sigma_0^2T+\sigma_0^2}.\qquad(19)$$

Conclusively, we obtain

$$\mathbb{E}\sqrt{\frac{\boldsymbol{\nu}_{0,i}}{1-\beta_2} + \sum_{s=1}^{T}|g_{s,i}|^2 + \sigma_0^2}$$

$$=\mathbb{E}\frac{\sigma_0^2 + \frac{\boldsymbol{\nu}_{0,i}}{1-\beta_2}}{\sqrt{\frac{\boldsymbol{\nu}_{0,i}}{1-\beta_2} + \sum_{s=1}^{T}|g_{s,i}|^2 + \sigma_0^2}} + \sum_{t=1}^{T}\mathbb{E}\frac{|\boldsymbol{g}_{t,i}|^2}{\sqrt{\frac{\boldsymbol{\nu}_{0,i}}{1-\beta_2} + \sum_{s=1}^{T}|g_{s,i}|^2 + \sigma_0^2}}\mathbf{1}_{|\boldsymbol{G}_{t,i}|<\frac{\sigma_0}{\sigma_1}}$$

$$+\sum_{t=1}^{T}\mathbb{E}\frac{|\boldsymbol{g}_{t,i}|^2}{\sqrt{\frac{\boldsymbol{\nu}_{0,i}}{1-\beta_2} + \sum_{s=1}^{T}|g_{s,i}|^2 + \sigma_0^2}}\mathbf{1}_{|\boldsymbol{G}_{t,i}|\geq\frac{\sigma_0}{\sigma_1}}$$

$$\leq\sqrt{\frac{\boldsymbol{\nu}_{0,i}}{1-\beta_2} + 2\sigma_0^2 T + \sigma_0^2} + 2\sqrt{1-\beta_2}\sigma_1^2\mathbb{E}\sum_{t=1}^{T}\frac{|\boldsymbol{G}_{t,i}|^2}{\sqrt{\widetilde{\boldsymbol{\nu}}_{t,i}}}\mathbf{1}_{|\boldsymbol{G}_{t,i}|\geq\frac{\sigma_0}{\sigma_1}}$$

$$\leq\sqrt{\frac{\boldsymbol{\nu}_{0,i}}{1-\beta_2} + 2\sigma_0^2 T + \sigma_0^2} + 2\sqrt{1-\beta_2}\sigma_1^2\mathbb{E}\sum_{t=1}^{T}\frac{|\boldsymbol{G}_{t,i}|^2}{\sqrt{\widetilde{\boldsymbol{\nu}}_{t,i}}}.$$

Secondly, as $\beta_2 \to 1$ as $T \to \infty$, $\beta_1 \leq \sqrt{\beta_2} - 8\sigma_1^2(1-\beta_2)\beta_2^{-2}$ holds for large enough $T$, and thus Theorem 1 holds. Applying the value of $\beta_1$, $\beta_2$, and $\eta$ to Eq. (16), we obtain that

$$\sum_{t=1}^{T}\mathbb{E}\left[\left\|\frac{1}{\sqrt[4]{\widetilde{\boldsymbol{\nu}}_t}}\odot\boldsymbol{G}_t\right\|^2\right] \leq D_2\sqrt{T} + D_1\sqrt{T}\sum_{i=1}^{d}\mathbb{E}\ln\boldsymbol{\nu}_{T,i} + \frac{64}{(1-c)^2}\sum_{i=1}^{d}\mathbb{E}\frac{\boldsymbol{G}_{1,i}^2}{\sqrt{\widetilde{\boldsymbol{\nu}}_{1,i}}}. \qquad (20)$$

Summing Eq. (19) with respect to $i$ then gives

$$\sum_{i=1}^{d}\mathbb{E}\sqrt{\frac{\boldsymbol{\nu}_{0,i}}{1-\beta_2} + \sum_{s=1}^{T}|g_{s,i}|^2 + \sigma_0^2}$$

$$\leq\sum_{i=1}^{d}\sqrt{\frac{\boldsymbol{\nu}_{0,i}}{1-\beta_2} + 2\sigma_0^2 T + \sigma_0^2} + 2\sqrt{1-\beta_2}\sigma_1^2\sum_{i=1}^{d}\sum_{t=1}^{T}\frac{|\boldsymbol{G}_{t,i}|^2}{\sqrt{\widetilde{\boldsymbol{\nu}}_{t,i}}}$$

$$\leq\sum_{i=1}^{d}\sqrt{\frac{\boldsymbol{\nu}_{0,i}}{1-\beta_2} + 2\sigma_0^2 T + \sigma_0^2} + 2D_2\sigma_1^2\sqrt{b} + 2D_1\sigma_1^2\sqrt{b}\sum_{i=1}^{d}\mathbb{E}\ln\boldsymbol{\nu}_{T,i} + \frac{128\sigma_1^2\sqrt{b}}{(1-c)^2\sqrt{T}}\sum_{i=1}^{d}\mathbb{E}\frac{\boldsymbol{G}_{1,i}^2}{\sqrt{\widetilde{\boldsymbol{\nu}}_{1,i}}}$$

$$=\sum_{i=1}^{d}\sqrt{\frac{\boldsymbol{\nu}_{0,i}}{1-\beta_2} + 2\sigma_0^2 T + \sigma_0^2} + 2D_2\sigma_1^2\sqrt{b} + 4D_1\sigma_1^2\sqrt{b}\sum_{i=1}^{d}\mathbb{E}\ln\sqrt{\boldsymbol{\nu}_{T,i}} + \frac{128\sigma_1^2\sqrt{b}}{(1-c)^2\sqrt{T}}\sum_{i=1}^{d}\mathbb{E}\frac{\boldsymbol{G}_{1,i}^2}{\sqrt{\widetilde{\boldsymbol{\nu}}_{1,i}}}$$

$$\leq\sum_{i=1}^{d}\sqrt{\frac{\boldsymbol{\nu}_{0,i}}{1-\beta_2} + 2\sigma_0^2 T + \sigma_0^2} + 2D_2\sigma_1^2\sqrt{b} + 4D_1\sigma_1^2\sqrt{b}\sum_{i=1}^{d}\mathbb{E}\ln\left(\sum_{i=1}^{d}\sqrt{1-\beta_2}\sqrt{\frac{\boldsymbol{\nu}_{0,i}}{1-\beta_2} + \sum_{s=1}^{T}|g_{s,i}|^2 + \sigma_0^2}\right)$$

$$+ \frac{128\sigma_1^2\sqrt{b}}{(1-c)^2\sqrt{T}}\sum_{i=1}^{d}\mathbb{E}\frac{\boldsymbol{G}_{1,i}^2}{\sqrt{\widetilde{\boldsymbol{\nu}}_{1,i}}}$$

$$\leq\sum_{i=1}^{d}\sqrt{\frac{\boldsymbol{\nu}_{0,i}}{1-\beta_2} + 2\sigma_0^2 T + \sigma_0^2} + 2D_2\sigma_1^2\sqrt{b} + 4D_1\sigma_1^2\sqrt{b}\sum_{i=1}^{d}\ln\mathbb{E}\left(\sum_{i=1}^{d}\sqrt{1-\beta_2}\sqrt{\frac{\boldsymbol{\nu}_{0,i}}{1-\beta_2} + \sum_{s=1}^{T}|g_{s,i}|^2 + \sigma_0^2}\right)$$

$$+ \frac{128\sigma_1^2\sqrt{b}}{(1-c)^2\sqrt{T}}\sum_{i=1}^{d}\mathbb{E}\frac{\boldsymbol{G}_{1,i}^2}{\sqrt{\widetilde{\boldsymbol{\nu}}_{1,i}}},$$

where the second inequality is due to Eq. (20), the second-to-last inequality is due to Eq. (18), and the last inequality is due to Jensen's inequality. Solving the above ineqaulity with respect to

$\sqrt{1-\beta_2}\sum_{i=1}^d \mathbb{E}\sqrt{\frac{\boldsymbol{\nu}_{0,i}}{1-\beta_2}+\sum_{s=1}^T |g_{s,i}|^2+\sigma_0^2}$ then gives

$$\sqrt{1-\beta_2}\sum_{i=1}^d \mathbb{E}\sqrt{\frac{\boldsymbol{\nu}_{0,i}}{1-\beta_2}+\sum_{s=1}^T |g_{s,i}|^2+\sigma_0^2}$$

$$\leq 2\sum_{i=1}^d \sqrt{\boldsymbol{\nu}_{0,i}+3b\sigma_0^2}+\frac{4D_2\sigma_1^2 b}{\sqrt{T}}+\frac{256\sigma_1^2 b}{(1-c)^2 T}\sum_{i=1}^d \mathbb{E}\frac{\boldsymbol{G}_{1,i}^2}{\sqrt{\widetilde{\boldsymbol{\nu}}_{1,i}}}+\frac{16D_1\sigma_1^2 b}{\sqrt{T}}\ln\left(e+\frac{4\tilde{D}\sigma_1^2 b}{\sqrt{T}}\right).$$
(21)

Therefore, by Cauchy's inequality, we have

$$\mathbb{E}\left[\sum_{t=1}^T \|\boldsymbol{G}_t\|_1\right]^2 \leq \left(\sum_{t=1}^T \mathbb{E}\left[\left\|\frac{1}{\sqrt[4]{\widetilde{\boldsymbol{\nu}}_t^1}}\odot\boldsymbol{G}_t\right\|^2\right]\right)\left(\sum_{t=1}^T\sum_{i=1}^d \mathbb{E}\sqrt{\widetilde{\boldsymbol{\nu}}_{t,i}}\right).$$

Since

$$\sum_{t=1}^T\sum_{i=1}^d \sqrt{\widetilde{\boldsymbol{\nu}}_{t,i}}\leq \sum_{t=1}^T\sum_{i=1}^d \sqrt{\beta_2\boldsymbol{\nu}_{t-1,i}+(1-\beta_2)\sigma_0^2}\leq T\sum_{i=1}^d\sqrt{1-\beta_2}\sqrt{\frac{\boldsymbol{\nu}_{0,i}}{1-\beta_2}+\sum_{s=1}^T |g_{s,i}|^2+\sigma_0^2},$$

we have

$$\mathbb{E}\left[\sum_{t=1}^T \|\boldsymbol{G}_t\|_1\right]^2$$

$$\leq\left(2T\sum_{i=1}^d\sqrt{\boldsymbol{\nu}_{0,i}+3b\sigma_0^2}+4D_2\sigma_1^2 b\sqrt{T}+\frac{256\sigma_1^2 b}{(1-c)^2}\sum_{i=1}^d \mathbb{E}\frac{\boldsymbol{G}_{1,i}^2}{\sqrt{\widetilde{\boldsymbol{\nu}}_{1,i}}}+16D_1\sigma_1^2 b\sqrt{T}\ln\left(e+\frac{4\tilde{D}\sigma_1^2 b}{\sqrt{T}}\right)\right)$$

$$\times\left(D_2\sqrt{T}+D_1\sqrt{T}\sum_{i=1}^d \mathbb{E}\ln\boldsymbol{\nu}_{T,i}+\frac{64}{(1-c)^2}\sum_{i=1}^d \mathbb{E}\frac{\boldsymbol{G}_{1,i}^2}{\sqrt{\widetilde{\boldsymbol{\nu}}_{1,i}}}\right)$$

$$\leq\left(2T\sum_{i=1}^d\sqrt{\boldsymbol{\nu}_{0,i}+3b\sigma_0^2}+4D_2\sigma_1^2 b\sqrt{T}+\frac{256\sigma_1^2 b}{(1-c)^2}\sum_{i=1}^d \mathbb{E}\frac{\boldsymbol{G}_{1,i}^2}{\sqrt{\widetilde{\boldsymbol{\nu}}_{1,i}}}+16D_1\sigma_1^2 b\sqrt{T}\ln\left(e+\frac{4\tilde{D}\sigma_1^2 b}{\sqrt{T}}\right)\right)$$

$$\times\left(2D_1\sqrt{T}\sum_{i=1}^d\ln\left(2\sum_{i=1}^d\sqrt{\boldsymbol{\nu}_{0,i}+3b\sigma_0^2}+\frac{4D_2\sigma_1^2 b}{\sqrt{T}}+\frac{256\sigma_1^2 b}{(1-c)^2 T}\sum_{i=1}^d \mathbb{E}\frac{\boldsymbol{G}_{1,i}^2}{\sqrt{\widetilde{\boldsymbol{\nu}}_{1,i}}}+\frac{16D_1\sigma_1^2 b}{\sqrt{T}}\ln\left(e+\frac{4\tilde{D}\sigma_1^2 b}{\sqrt{T}}\right)\right)\right.$$

$$\left.+\frac{64}{(1-c)^2}\sum_{i=1}^d \mathbb{E}\frac{\boldsymbol{G}_{1,i}^2}{\sqrt{\widetilde{\boldsymbol{\nu}}_{1,i}}}+D_2\sqrt{T}\right).$$

The proof is completed. $\qquad\square$

## E   Experiments

Table 1: Exploring effect of $\beta_1$ of Adam. We explore the dataset of Cifar10 using VGG13[28] and ResNet18[17] and WiKiText2[24] using Transforemer[29]. We show the training loss after 50 epochs

| $\beta_1$ | 0 | 0.1 | 0.2 | 0.3 | 0.4 | 0.5 | 0.6 | 0.7 | 0.8 | 0.9 | 0.99 | 0.999 | 0.9999 |
|---|---|---|---|---|---|---|---|---|---|---|---|---|---|
| Cifar10 ResNet18 | 0.2268 | 0.2197 | 0.2158 | 0.2197 | 0.2182 | 0.2198 | 0.2217 | 0.2204 | 0.2218 | 0.2222 | 0.2351 | 0.3620 | 0.6187 |
| Cifar10 VGG13 | 0.1416 | 0.1605 | 0.1428 | 0.1453 | 0.1391 | 0.1421 | 0.1387 | 0.1457 | 0.1417 | 0.1419 | 0.1551 | 0.3497 | 0.6645 |
| WikiText2 | 3.3600 | 3.3589 | 3.3586 | 3.3573 | 3.3565 | 3.3599 | 3.3627 | 3.3634 | 3.3659 | 3.3749 | 3.4314 | 6.3274 | 7.5384 |

As mentioned in Section 6, one of the limitations of our theory is that it can not provide better results when momentum is present. To complement such a limitation, we initialize a empirical study of the effect of momentum in Adam as follows.

**Experimental setting.**   We use Adam training on Cifar 10 with ResNet18 [28] and VGG13 [17] and wikitext2 with two-layer Transformer [29] for 50 epoch and record its training loss at 50 epoch

as a measure for the optimization speed. Smaller loss indicates better optimization. The batch size is set 1024 for Cifar10 dataset and 100 for WikiText2 Dataset.

The results are given in Table 1. Our discoveries are:

- Momentum can benefit the optimization when the $\beta$ is not too large.
- For all datasets, larger $\beta_1$ (Setting $\beta_1$ close to 1) will worse the optimization.

**Connection with Theorem 1.** One can easily observe that in Theorem 1 both $C_1$ and $C_2$ polynomially depend on $\frac{1}{1-\beta_1}$ and thus so does $\mathbb{E}\sum_{t=1}^{T}\|\nabla f(\boldsymbol{w}_t)\| = \tilde{\mathcal{O}}(\frac{1}{1-\beta_1})$. Therefore, **Theorem 1 is aligned with the experimental results in the sense** that both our theory and the experimental results indicate that the Adam will become worse when the $\beta_1$ is close to 1. **However, Theorem 1 cannot explain the benefit of using momentum (by setting $\beta_1$ larger than 0).** This may be due to that our Theorem 1 is a worst-case analysis. We conjecture that theoretically deriving the benefit of momentum requires restricting the underlying objective function to a more specific range, which we leave as a future work.

