$$+ \sqrt{C_2 + \frac{2}{(1-\beta_1)\eta}C_1 d\ln\left(12C_2 + 2T\sum_{l=1}^{d}\sqrt{\boldsymbol{\nu}_{0,l} + (3-\beta_2)\sigma^2} + 4dC_1\ln dC_1\right)}$$

$$\times \sqrt{12C_2 + 2T\sum_{l=1}^{d}\sqrt{\boldsymbol{\nu}_{0,l} + (3-\beta_2)\sigma^2} + 4dC_1\ln dC_1}. \tag{2}$$

*where $\boldsymbol{\nu}_{0,l}$ is the l-th coordinate of $\boldsymbol{\nu}_0$,*

$$C_1 = \left(\frac{L}{2}\eta^2 + 2\frac{\sqrt{1-\beta_2}}{(1-\beta_1)^2}\eta\sigma + \frac{\eta^2\beta_1}{\sqrt{\beta_2}(1-\frac{\beta_1}{\sqrt{\beta_2}})} + L^2\frac{\beta_1\eta^3(1-\beta_1)}{\beta_2(1-\beta_2)^{\frac{1}{2}}(1-\frac{\beta_1^2}{\beta_2})(1-\frac{\beta_1}{\beta_2})^2}\frac{d}{\sigma}\frac{(1-\beta_1)^2}{(1-\frac{\beta_1}{\sqrt{\beta_2}})^2}\right)\frac{1}{1-\beta_2}.$$

*and*

$$C_2 = \frac{2}{(1-\beta_1)\eta}\left(f(\boldsymbol{w}_1) + \sum_{l=1}^{d}2C_1\left(\mathbb{E}\ln\left(\frac{1}{\boldsymbol{\nu}_{0,l}}\right) - T\ln\beta_2\right)\right).$$

A proof sketch is given in Section 4.2 and the full proof is deferred to Appendix.

The right-hand side in Eq. (2) looks messy at the first glance. We next explain Theorem 1 in detail and make the upper bound's dependence over hyperparameters crystally clear.

## 4.1 Discussion on Theorem 1

**Required assumptions and conditions.** As mentioned previously, Theorem 1 only requires Assumption 1 and 2, which aligns with the setting of the lower bound (Proposition 1). To our best knowledge, this is the first analysis of Adam without additional assumptions. Also, Theorem 1 holds for general choices of hyperparameters since the only condition posed on hyperparameters is $\beta_1 < \beta_2$. Such condition covers a wide range of hyperparameters, e.g., the default setting $\beta_1 = 0.9$ and $\beta_2 = 0.999$ in PyTorch [19].

**Dependence over $\beta_2$, $\eta$, and $T$.** Here we consider the influence of $\beta_2$, $\eta$, and $T$ while fixing $\beta_1$ constant (we will discuss the effect of $\beta_1$ in Section 6). With logarithmic factors ignored and coefficients hidden, $C_1$, $C_2$ and the right-hand-side of Eq. (2) can be rewritten with asymptotic notations as

$$C_1 = \tilde{\mathcal{O}}\left(\frac{\eta}{\sqrt{1-\beta_2}} + \frac{\eta^3}{(1-\beta_2)^{\frac{3}{2}}}\right),$$

$$C_2 = \tilde{\mathcal{O}}\left(\frac{1}{\sqrt{1-\beta_2}} + \frac{\eta^2}{(1-\beta_2)^{\frac{3}{2}}} + \frac{1}{\eta} + T\sqrt{1-\beta_2} + \frac{\eta^2 T}{(1-\beta_2)^{\frac{1}{2}}}\right),$$

$$\mathbb{E}\sum_{t=1}^{T}\|\nabla f(\boldsymbol{w}_t)\| = \tilde{\mathcal{O}}\left(\sqrt{1-\beta_2}C_2 + \frac{\sqrt{1-\beta_2}}{\eta}C_1 + \sqrt{C_2 + \frac{C_1}{\eta}}\sqrt{C_2 + T + C_1}\right),$$

where $\tilde{\mathcal{O}}$ denotes $\mathcal{O}$ with logarithmic terms ignored. Consequently, the dependence of Eq. (2) over $\beta_2, \eta$ and $T$ becomes

$$\mathbb{E}\sum_{t=1}^{T}\|\nabla f(\boldsymbol{w}_t)\| = \tilde{\mathcal{O}}\left(\frac{1}{\sqrt{1-\beta_2}} + \frac{\eta^2}{(1-\beta_2)^{\frac{3}{2}}} + \frac{1}{\eta} + \frac{\eta^2 T}{(1-\beta_2)^{\frac{1}{2}}}\right)$$

$$+ \tilde{\mathcal{O}}\left(\frac{\sqrt{T}}{\sqrt[4]{1-\beta_2}} + \frac{\eta\sqrt{T}}{(1-\beta_2)^{\frac{3}{4}}} + \frac{\sqrt{T}}{\sqrt{\eta}} + T\sqrt[4]{1-\beta_2} + \frac{\eta T}{(1-\beta_2)^{\frac{1}{4}}}\right).$$

Therefore, in order to ensure convergence, $\min_{t\in[T]}\mathbb{E}\|\boldsymbol{G}_t\|_1 \to 0$ as $T \to \infty$, a sufficient condition is that the right-hand-side of the above equation is $\boldsymbol{o}(T)$. Specifically, by choosing $\eta = \Theta(T^{-a})$ and $1 - \beta_2 = \Theta(T^{-b})$, we obtain that

$$\frac{1}{T}\mathbb{E}\sum_{t=1}^{T}\|\nabla f(\boldsymbol{w}_t)\| = \tilde{\mathcal{O}}\left(T^{\frac{b}{2}-1} + T^{-2a+\frac{3}{2}b-1} + T^{a-1} + T^{-2a+\frac{1}{2}b} + T^{\frac{b}{4}-\frac{1}{2}} + T^{-a+\frac{3}{4}b-\frac{1}{2}} + T^{\frac{1}{2}a-\frac{1}{2}} + T^{-a+\frac{1}{4}b}\right).$$

By simple calculation, we obtain that the right-hand side of the above inequality is $\boldsymbol{o}(1)$ as $T \to \infty$ if and only if $0 < \frac{b}{4} < a < 1$ and $3b - 4a < 2$. Moreover, the minimum of the right-hand side of the above inequality is $\tilde{\mathcal{O}}(\frac{1}{T^{\frac{1}{4}}})$, which is achieved at $a = \frac{1}{2}$ and $b = 1$. Such a minimum implies an upper bound of the iteration complexity which at most differs from the lower bound by logarithmic factors as solving $\tilde{\mathcal{O}}(\frac{1}{T^{\frac{1}{4}}}) = \varepsilon$ gives $T = \tilde{\mathcal{O}}(\frac{1}{\varepsilon^4})$. In Theorem 2, we will further remove the logarithmic factor by giving a refined proof when $a = \frac{1}{2}$ and $b = 1$ and close the gap between the upper and lower bounds.

## 4.2 Proof Sketch of Theorem 1

In this section, we demonstrate the proof idea of Theorem 1. Concretely, we sketch the proof by identifying two key challenges in the proof and provide our solutions respectively.

**Challenge I: Disentangle the stochasticity in momentum and adaptive learning rate.** According to the standard descent lemma, we have that

$$
\mathbb{E}f(\boldsymbol{w}_{t+1}) = f(\boldsymbol{w}_t) + \mathbb{E}\left[\langle \boldsymbol{G}_t, \boldsymbol{w}_{t+1} - \boldsymbol{w}_t \rangle + \frac{L}{2}\|\boldsymbol{w}_{t+1} - \boldsymbol{w}_t\|^2\right]
$$
$$
\leq \mathbb{E}f(\boldsymbol{w}_t) + \underbrace{\mathbb{E}\left[\left\langle \boldsymbol{G}_t, -\eta\frac{1}{\sqrt{\boldsymbol{\nu}_t}} \odot \boldsymbol{m}_t \right\rangle\right]}_{\text{First Order}} + \underbrace{\frac{L}{2}\eta^2\mathbb{E}\left\|\frac{1}{\sqrt{\boldsymbol{\nu}_t}} \odot \boldsymbol{m}_t\right\|^2}_{\text{Second Order}} \tag{3}
$$

The first challenge arises from bounding the "First Order" term above. To faciliate the understanding of the difficulty, we compare the "First Order" term of Adam to the corresponding "First Order" term of SGD, i.e., $-\eta\mathbb{E}\langle \boldsymbol{G}_t, \boldsymbol{g}_t \rangle$. By directly applying $\mathbb{E}^{|\mathcal{F}_t}\boldsymbol{g}_t = \boldsymbol{G}_t$, we obtain that the "First-Order" term of SGD equals to $-\eta\mathbb{E}\|\boldsymbol{G}_t\|^2\rangle$. However, as for Adam, there are two folds of trouble: firstly, we do not know what $\mathbb{E}^{|\mathcal{F}_t}\frac{1}{\sqrt{\boldsymbol{\nu}_t}} \odot \boldsymbol{m}_t$ is, as the stochasticity in $\boldsymbol{m}_t$ and $\boldsymbol{\nu}_t$ entangles. Secondly, even without $\boldsymbol{\nu}_t$, it is unclear how $\mathbb{E}^{\mathcal{F}_t}\boldsymbol{m}_t$ aligns with $\boldsymbol{G}_t$ given the existence of $\boldsymbol{g}_{t-1}, \cdots, \boldsymbol{g}_1$ in $\boldsymbol{m}_t$.

**Solution to Challenge I.** For $i \in [1, t]$, we define a set of surrogate conditioner $\widetilde{\boldsymbol{\nu}}_t^i \triangleq \beta_2^i\boldsymbol{\nu}_{t-i} + \sum_{j=0}^{i-1}\beta_2^j(1 - \beta_2)\boldsymbol{G}_{t-i+1}^{\odot 2} + (1 - \beta_2)\sigma^2$, and $\widetilde{\boldsymbol{\nu}}_t^0 \triangleq \boldsymbol{\nu}_t$. Note that $\widetilde{\boldsymbol{\nu}}_t^i$ is measurable with respect to $\mathcal{F}_{t-i+1}$. The key idea of our solution is the following *peeling-off strategy*: starting from $\mathbb{E}[\langle \boldsymbol{G}_t, \frac{1}{\sqrt{\boldsymbol{\nu}_t}} \odot \boldsymbol{m}_t \rangle]$, we replace $\boldsymbol{\nu}_t = \widetilde{\boldsymbol{\nu}}_t^0$ by $\widetilde{\boldsymbol{\nu}}_t^1$ (of course, such a replacement will bring a error term, which we temporily ignore and will consider it in the formal proof) and obtain $\mathbb{E}[\langle \boldsymbol{G}_t, \frac{1}{\sqrt{\widetilde{\boldsymbol{\nu}}_t^1}} \odot \boldsymbol{m}_t \rangle]$. As $\boldsymbol{m}_t = \beta_1\boldsymbol{m}_{t-1} + (1 - \beta_1)\boldsymbol{g}_t$, we further have $\mathbb{E}[\langle \boldsymbol{G}_t, \frac{1}{\sqrt{\widetilde{\boldsymbol{\nu}}_t^1}} \odot \boldsymbol{m}_t \rangle] = \mathbb{E}[\langle \boldsymbol{G}_t, \frac{1}{\sqrt{\widetilde{\boldsymbol{\nu}}_t^1}} \odot (1 - \beta_1)\boldsymbol{g}_t \rangle] + \mathbb{E}[\langle \boldsymbol{G}_t - \boldsymbol{G}_{t-1}, \frac{1}{\sqrt{\widetilde{\boldsymbol{\nu}}_t^1}} \odot \beta_1\boldsymbol{m}_{t-1} \rangle] + \mathbb{E}[\langle \boldsymbol{G}_{t-1}, \frac{1}{\sqrt{\widetilde{\boldsymbol{\nu}}_t^1}} \odot \beta_1\boldsymbol{m}_{t-1} \rangle]$. As $\widetilde{\boldsymbol{\nu}}_t^1$ is measurable w.r.t. $\mathcal{F}_t$, we can then disentangle the stochasticity in $g_t$ and $\boldsymbol{\nu}_t$, and the term $\mathbb{E}[\langle \boldsymbol{G}_t, \frac{1}{\sqrt{\widetilde{\boldsymbol{\nu}}_t^1}} \odot (1 - \beta_1)\boldsymbol{g}_t \rangle]$ equals to $\mathbb{E}[\langle \boldsymbol{G}_t, \frac{1}{\sqrt{\widetilde{\boldsymbol{\nu}}_t^1}} \odot (1 - \beta_1)\boldsymbol{G}_t \rangle]$, which is desired. The term $\mathbb{E}[\langle \boldsymbol{G}_t - \boldsymbol{G}_{t-2}, \frac{1}{\sqrt{\widetilde{\boldsymbol{\nu}}_t^1}} \odot \beta_1\boldsymbol{m}_{t-1} \rangle]$ is small due to $L$-smooth condition. The term $\mathbb{E}[\langle \boldsymbol{G}_{t-1}, \frac{1}{\sqrt{\widetilde{\boldsymbol{\nu}}_t^1}} \odot \beta_1\boldsymbol{m}_{t-1} \rangle]$ resembles $\mathbb{E}[\langle \boldsymbol{G}_t, \frac{1}{\sqrt{\boldsymbol{\nu}_t}} \odot \boldsymbol{m}_t \rangle]$, and we can apply the methodology recursively to get $\mathbb{E}[\langle \boldsymbol{G}_{t-2}, \frac{1}{\sqrt{\widetilde{\boldsymbol{\nu}}_t^2}} \odot \beta_1^2\boldsymbol{m}_{t-2} \rangle]$, $\mathbb{E}[\langle \boldsymbol{G}_{t-3}, \frac{1}{\sqrt{\widetilde{\boldsymbol{\nu}}_t^3}} \odot \beta_1^3\boldsymbol{m}_{t-3} \rangle]$, and so on. All in all, the above methodology can be summarized as the following lemma.

**Lemma 1.** *Let all conditions in Theorem 1 hold. Denote $F_t^i \triangleq \mathbb{E}\langle \boldsymbol{G}_{t-i}, \frac{1}{\sqrt{\widetilde{\boldsymbol{\nu}}_t^i}} \odot \boldsymbol{m}_{t-i} \rangle$. Set $\boldsymbol{G}_0 \triangleq \boldsymbol{G}_1$ Then, $\forall t \geq 1$ and $i \in [0, t-1]$,*

$$
F_t^i \geq \beta_1 F_t^{i+1} + \frac{(1 - \beta_1)}{2}\mathbb{E}\left[\left\|\frac{1}{\sqrt[4]{\widetilde{\boldsymbol{\nu}}_t^{i+1}}} \odot \boldsymbol{G}_{t-i}\right\|^2\right] - \beta_1 L\mathbb{E}\left[\|\boldsymbol{w}_{t-i} - \boldsymbol{w}_{t-i-1}\|\left\|\frac{1}{\sqrt{\widetilde{\boldsymbol{\nu}}_t^{i+1}}} \odot \boldsymbol{m}_{t-i-1}\right\|\right]
$$
$$
- \left(2\frac{\sqrt{1 - \beta_2}}{1 - \beta_1}\sigma + L^2\frac{\eta^2(1 - \beta_1)}{(1 - \beta_2)^{\frac{1}{2}}(1 - \frac{\beta_1^2}{\beta_2})\beta_2^i}\frac{i}{\sigma}d\right)\mathbb{E}\left\|\frac{1}{\sqrt{\boldsymbol{\nu}_{t-i}}} \odot \boldsymbol{m}_{t-i}\right\|^2.
$$

The proof is deferred to Appendix C.1. We highlight here that despite the simple methodology above, the proof itself is highly non-trivial and technical. The core difficulty lies in handling the error introduced by approximating $\widetilde{\nu}_t^i$ with $\widetilde{\nu}_t^{i+1}$, where we need to bound the gap both between $g_{t-i}$ and $G_{t-i}$ and between $G_{t-i}$ and $G_{t-i+1}$.

**Remark 1.** *Our surrogate conditioners $\widetilde{\nu}_t^i$ are novel. Previously, there are other surrogate condition-ers in Défossez et al. [10], Zou et al. [27] which help to disentangle the stochasticity in $g_t$ and $\nu_t$. However, none of them can be applied in our setting because the bounded gradient assumption is required to use them, which is missed in our setting. Therefore, our surrogate conditioners may also shed light on the other analysis of Adam where no bounded gradient is assumed.*

Based on Lemma 1, we can estimate the "First-Order" term recursively. Combining the estimation of the "First-Order" term back to the descent lemma (Eq. (3)) and summing the descent lemma over $t$ from 1 to $T$, we obtain

$$\sum_{t=1}^{T}\frac{(1-\beta_1)\eta}{2}\mathbb{E}\left[\left\|\frac{1}{\sqrt[4]{\widetilde{\nu}_t^1}}\odot G_t\right\|^2\right] \leq f(w_1) - \mathbb{E}f(w_{T+1}) + \sum_{l=1}^{d}C_1\left(\mathbb{E}\ln\left(\frac{\nu_{T,l}}{\nu_{0,l}}\right) - T\ln\beta_2\right). \quad (4)$$

We then encounter the second challenge.

**Challenge II: Convert Eq. (4) to a bound of gradient norm.** Although we have bounded the sum of $\mathbb{E}[\|\frac{1}{\sqrt[4]{\widetilde{\nu}_t^1}}\odot G_t\|^2]$, we need to convert it into a bound of $\mathbb{E}[\|G_t\|^2]$. In existing works [27, 10, 14] which assumes bounded gradient, such a conversion is straightforward because (their version of) $\widetilde{\nu}_t^1$ is upper bounded. However, we do not assume bounded gradient and $\widetilde{\nu}_t^1$ can be aribitrarily large, making $\mathbb{E}[\|\frac{1}{\sqrt[4]{\widetilde{\nu}_t^1}}\odot G_t\|^2]$ arbitrarily small than $\mathbb{E}[\|G_t\|^2]$.

**Solution to Challenge II.** As this part involves coordinate-wise analysis, we define $g_{t,l}$, $G_{t,l}$, $\nu_{t,l}$, and $\widetilde{\nu}_{t,l}^1$ respectively as the $l$-th coordinate of $g_t$, $G_t$, $\nu_t$, and $\widetilde{\nu}_t^1$. To begin with, note that due to Cauchy's inequality and Hölder's inequality,

$$\left(\mathbb{E}\sum_{t=1}^{T}\|G_t\|\right)^2 \leq \left(\sum_{t=1}^{T}\mathbb{E}\left[\left\|\frac{1}{\sqrt[4]{\widetilde{\nu}_t^1}}\odot G_t\right\|^2\right]\right)\left(\sum_{t=1}^{T}\mathbb{E}\left[\left\|\sqrt[4]{\widetilde{\nu}_t^1}\right\|^2\right]\right). \quad (5)$$

Therefore, we only need to derive an upper bound of $\sum_{t=1}^{T}\mathbb{E}[\|\sqrt[4]{\widetilde{\nu}_t^1}\|^2]$, which is achieved by the following divide-and-conque methodology. Firstly, when $|G_{t,l}| \geq \sigma$, we can show $2\mathbb{E}^{|\mathcal{F}_t}|g_{t,l}|^2 \geq 2|G_{t,l}|^2 \geq \mathbb{E}^{|\mathcal{F}_t}|g_{t,l}|^2$. Then, by the concavity of $f(x) = \frac{x}{\sqrt{a+x}}(a > 0)$ and through a massive calculation, we obtain that

$$\mathbb{E}\left[\frac{|G_{t,l}|^2}{\sqrt{\widetilde{\nu}_{t,l}^1}}\mathbf{1}_{|G_{t,l}|\geq\sigma}\right] \geq \frac{1}{3(1-\beta_2)}\mathbb{E}(\sqrt{\nu_{t,l} + (1-\beta_2)\sigma^2} - \sqrt{\beta_2(\nu_{t-1,l} + (1-\beta_2)\sigma^2)})\mathbf{1}_{|G_{t,l}|\geq\sigma},$$

and thus

$$\sum_{t=1}^{T}\mathbb{E}\left[\frac{|G_{t,l}|^2}{\sqrt{\widetilde{\nu}_{t,l}^1}}\right] \geq \sum_{t=1}^{T}\frac{1}{3(1-\beta_2)}\mathbb{E}(\sqrt{\nu_{t,l} + (1-\beta_2)\sigma^2} - \sqrt{\beta_2(\nu_{t-1,l} + (1-\beta_2)\sigma^2)})\mathbf{1}_{|G_{t,l}|\geq\sigma}.$$

Secondly, when $|G_{t,l}| < \sigma$, define $\{\bar{\nu}_{t,l}\}_{t=0}^{\infty}$ as $\bar{\nu}_{0,l} = \nu_{0,l}$, $\bar{\nu}_{t,l} = \bar{\nu}_{t-1,l} + |g_{t,l}|^2\mathbf{1}_{|G_{t,l}|<\sigma}$. One can easily observe that $\bar{\nu}_{t,l} \leq \nu_{t,l}$, and thus

$$\sum_{t=1}^{T}\mathbb{E}\left(\sqrt{\nu_{t,l} + (1-\beta_2)\sigma^2} - \sqrt{\beta_2(\nu_{t-1,l} + (1-\beta_2)\sigma^2)}\right)\mathbf{1}_{|G_{t,l}|\geq\sigma}$$

$$\leq \sum_{t=1}^{T}\mathbb{E}\left(\sqrt{\bar{\nu}_{t,l} + (1-\beta_2)\sigma^2} - \sqrt{\beta_2(\bar{\nu}_{t-1,l} + (1-\beta_2)\sigma^2)}\right)$$

$$= \mathbb{E}\sqrt{\bar{\nu}_{T,l} + (1-\beta_2)\sigma^2} + (1 - \sqrt{\beta_2})\sum_{t=1}^{T-1}\mathbb{E}\sqrt{\bar{\nu}_{t,l} + (1-\beta_2)\sigma^2} - \mathbb{E}\sqrt{\beta_2(\bar{\nu}_{0,l} + (1-\beta_2)\sigma^2)}.$$

Putting the above two estimations together, we derive that

$$\sum_{t=1}^{T} \sum_{l=1}^{d} \mathbb{E}\sqrt{\boldsymbol{\nu}_{t,l} + (1-\beta_2)\sigma^2} \leq 3(1+\sqrt{\beta_2}) \sum_{t=1}^{T} \sum_{l=1}^{d} \mathbb{E}\left[\frac{|\boldsymbol{G}_{t,l}|^2}{\sqrt{\widetilde{\boldsymbol{\nu}}_{t,l}^1}}\right] + T\sum_{l=1}^{d} \sqrt{\boldsymbol{\nu}_{0,l} + (3-\beta_2)\sigma^2}.$$

The above methodology can be summarized as the following lemma.

**Lemma 2.** *Let all conditions in Theorem 1 hold. Then,*

$$\sum_{t=1}^{T} \sum_{l=1}^{d} \mathbb{E}\sqrt{\boldsymbol{\nu}_{t,l} + (1-\beta_2)\sigma^2} \leq 2T\sum_{l=1}^{d} \sqrt{\boldsymbol{\nu}_{0,l} + (3-\beta_2)\sigma^2} + 4dC_1 \ln dC_1 + 12C_2.$$

Based on Lemma 2, we can derive the estimation of $\sum_{t=1}^{T} \mathbb{E}[\|\sqrt[4]{\widetilde{\boldsymbol{\nu}}_t^1}\|^2]$ since $\widetilde{\boldsymbol{\nu}}_t^1$ is close to $\boldsymbol{\nu}_t$. The proof is then completed by combining the estimation of $\sum_{t=1}^{T} \mathbb{E}[\|\sqrt[4]{\widetilde{\boldsymbol{\nu}}_t^1}\|^2]$ and Eq. (5).

# 5 Gap-closing upper bound on the iteration complexity of Adam

In this section, based on a refined proof of Stage II of Theorem 1 (see Appendix C) under the specific case $\eta = \Theta(1/\sqrt{T})$ and $\beta_2 = 1 - \Theta(1/T)$, we show that the logarithmic factor in Theorem 1 can be removed and the lower bound can be achieved. Specifically, we have the following theorem.

**Theorem 2.** *Let Assumption 1 and Assumption 2 hold. Then, select the hyperparameters of Adam as $\eta = \frac{a}{\sqrt{T}}$, $\beta_2 = 1 - \frac{b}{T}$ and $\beta_1 = c\beta_2$, where $a, b > 0$ and $0 \leq c < 1$ are independent of $T$. Then, let $\boldsymbol{w}_\tau$ be the output of Adam in Algorithm 1, and we have*

$$\mathbb{E}\|\nabla f(\boldsymbol{w}_\tau)\| \leq \frac{1}{\sqrt[4]{T}} \sqrt{\frac{2}{\sqrt{b}}\left(D_1 + 2D_2 \ln\left(\frac{2\sqrt{b}}{\sqrt{T}}D_1 + \frac{4b}{T}D_2^2 + \sum_{l=1}^{d}\sqrt{\boldsymbol{\nu}_{0,l} + 3b\sigma^2}\right)\right)}$$

$$\times \sqrt{\frac{2\sqrt{b}}{\sqrt{T}}D_1 + \frac{4b}{T}D_2^2 + \sum_{l=1}^{d}\sqrt{\boldsymbol{\nu}_{0,l} + 3b\sigma^2} + \frac{1}{T}\left(D_1 + 2D_2 \ln\left(\frac{2\sqrt{b}}{\sqrt{T}}D_1 + \frac{4b}{T}D_2^2 + \sum_{l=1}^{d}\sqrt{\boldsymbol{\nu}_{0,l} + 3b\sigma^2}\right)\right)},$$

*where*

$$D_1 \triangleq \frac{4\sqrt{b}}{a(1-c)}f(\boldsymbol{w}_1) + \sum_{l=1}^{d}\frac{2}{ab\sqrt{b}}\left(La^2 + 4\frac{a\sqrt{b}\sigma}{(1-c)^2} + 2\frac{a^2c}{1-c} + 2\frac{L^2ca^3d}{\sqrt{b}(1-c)^5\sigma}\right)\left(-\ln\left(\boldsymbol{\nu}_{0,l}\right) + b\right),$$

$$D_2 \triangleq d\frac{2}{ab\sqrt{b}}\left(La^2 + 4\frac{a\sqrt{b}\sigma}{(1-c)^2} + 2\frac{a^2c}{1-c} + 4\frac{L^2ca^3d}{\sqrt{b}(1-c)^5\sigma}\

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

+1,l} - \boldsymbol{w}_{t,l}| = \eta\left|\frac{\boldsymbol{m}_{t,l}}{\sqrt{\boldsymbol{\nu}_{t,l}}}\right| \leq \eta\frac{\sum_{i=0}^{t-1}(1 - \beta_1)\beta_1^i|\boldsymbol{g}_{t-i,l}|}{\sqrt{\sum_{i=0}^{t-1}(1 - \beta_2)\beta_2^i|\boldsymbol{g}_{t-i,l}|^2 + \beta_2^t\boldsymbol{\nu}_{0,l}}}$$

$$\leq \eta\frac{1 - \beta_1}{\sqrt{1 - \beta_2}}\frac{\sqrt{\sum_{i=0}^{t-1}\beta_2^i|\boldsymbol{g}_{t-i,l}|^2}\sqrt{\sum_{i=0}^{t-1}\frac{\beta_1^{2i}}{\beta_2^i}}}{\sqrt{\sum_{i=0}^{t-1}\beta_2^i|\boldsymbol{g}_{t-i,l}|^2}} \leq \eta\frac{1 - \beta_1}{\sqrt{1 - \beta_2}\sqrt{1 - \frac{\beta_1^2}{\beta_2}}}.$$

Here the second inequality is due to Cauchy's inequality. The proof is completed. □

# C   Proof of Theorem 1

## C.1   Proof of Lemma 1 and Lemma 2

*Proof of Lemma 1.* $\forall i \in [0, t-1]$, we have the following decomposition:

$$F_t^i = \underbrace{\mathbb{E}\left[\left\langle \boldsymbol{G}_{t-i}, \frac{1}{\sqrt{\widetilde{\boldsymbol{\nu}}_t^{i+1}}} \odot \boldsymbol{m}_{t-i} \right\rangle\right]}_{(i)_t^i} + \underbrace{\mathbb{E}\left[\left\langle \boldsymbol{G}_{t-i}, \left(\frac{1}{\sqrt{\widetilde{\boldsymbol{\nu}}_t^i}} - \frac{1}{\sqrt{\widetilde{\boldsymbol{\nu}}_t^{i+1}}}\right) \odot \boldsymbol{m}_{t-i} \right\rangle\right]}_{(ii)_t^i}.$$

As for $(i)_t^i$, according to the definition of $\boldsymbol{m}_{t-i}$, it can be lower bounded as

$$\mathbb{E}\left[\left\langle \boldsymbol{G}_{t-i}, \frac{1}{\sqrt{\widetilde{\boldsymbol{\nu}}_t^{i+1}}} \odot \boldsymbol{m}_{t-i} \right\rangle\right] = \mathbb{E}\left[\left\langle \boldsymbol{G}_{t-i}, (1-\beta_1)\frac{1}{\sqrt{\widetilde{\boldsymbol{\nu}}_t^{i+1}}} \odot \boldsymbol{g}_{t-i} \right\rangle\right] + \mathbb{E}\left[\left\langle \boldsymbol{G}_{t-i}, \beta_1 \frac{1}{\sqrt{\widetilde{\boldsymbol{\nu}}_t^{i+1}}} \odot \boldsymbol{m}_{t-i-1} \right\rangle\right]$$

$$= \mathbb{E}\left[(1-\beta_1)\left\|\frac{1}{\sqrt[4]{\widetilde{\boldsymbol{\nu}}_t^{i+1}}} \odot \boldsymbol{G}_{t-i}\right\|^2\right] + \mathbb{E}\left[\left\langle \boldsymbol{G}_{t-i-1}, \beta_1 \frac{1}{\sqrt{\widetilde{\boldsymbol{\nu}}_t^{i+1}}} \odot \boldsymbol{m}_{t-i-1} \right\rangle\right] + \mathbb{E}\left[\left\langle \boldsymbol{G}_{t-i} - \boldsymbol{G}_{t-i-1}, \beta_1 \frac{1}{\sqrt{\widetilde{\boldsymbol{\nu}}_t^{i+1}}} \odot \boldsymbol{m}_{t-i-1} \right\rangle\right]$$

$$\geq \mathbb{E}\left[(1-\beta_1)\left\|\frac{1}{\sqrt[4]{\widetilde{\boldsymbol{\nu}}_t^{i+1}}} \odot \boldsymbol{G}_{t-i}\right\|^2\right] + \mathbb{E}\left[\left\langle \boldsymbol{G}_{t-i-1}, \beta_1 \frac{1}{\sqrt{\widetilde{\boldsymbol{\nu}}_t^{i+1}}} \odot \boldsymbol{m}_{t-i-1} \right\rangle\right] - \beta_1 L\mathbb{E}\left[\|\boldsymbol{w}_{t-i} - \boldsymbol{w}_{t-i-1}\|\left\|\frac{1}{\sqrt{\widetilde{\boldsymbol{\nu}}_t^{i+1}}} \odot \boldsymbol{m}_{t-i-1}\right\|\right],$$

where the last inequality is due to Assumption 1. As for $(ii)_t^i$, if $i = 0$, we have

$$\left|\mathbb{E}^{|\mathcal{F}_t}\left[\left\langle \boldsymbol{G}_t, \left(\frac{1}{\sqrt{\widetilde{\boldsymbol{\nu}}_t^0}} - \frac{1}{\sqrt{\widetilde{\boldsymbol{\nu}}_t^1}}\right) \odot \boldsymbol{m}_t \right\rangle\right]\right| \leq \sum_{l=1}^d |\boldsymbol{G}_{t,l}|\mathbb{E}^{|\mathcal{F}_t}\left[|\boldsymbol{m}_{t,l}|\left|\frac{1}{\sqrt{\boldsymbol{\nu}_{t,l}}} - \frac{1}{\sqrt{\widetilde{\boldsymbol{\nu}}_{t,l}^1}}\right|\right]$$

$$= \sum_{l=1}^d |\boldsymbol{G}_{t,l}|\|\mathbb{E}^{|\mathcal{F}_t}\left[|\boldsymbol{m}_{t,l}|\frac{(1-\beta_2)\left||\boldsymbol{G}_{t,l}|^2 - |\boldsymbol{g}_{t,l}|^2\right| + (1-\beta_2)\sigma^2}{\sqrt{\boldsymbol{\nu}_{t,l}\widetilde{\boldsymbol{\nu}}_{t,l}^1}(\sqrt{\boldsymbol{\nu}_{t,l}} + \sqrt{\widetilde{\boldsymbol{\nu}}_{t,l}^1})}\right]$$

$$\overset{(*)}{\leq} \sum_{l=1}^d |\boldsymbol{G}_{t,l}|\mathbb{E}^{|\mathcal{F}_t}\left[\||\boldsymbol{m}_{t,l}|\frac{(1-\beta_2)|\boldsymbol{G}_{t,l} - \boldsymbol{g}_{t,l}|(|\boldsymbol{G}_{t,l}| + |\boldsymbol{g}_{t,l}|) + (1-\beta_2)\sigma^2}{\sqrt{\boldsymbol{\nu}_{t,l}\widetilde{\boldsymbol{\nu}}_{t,l}^i}(\sqrt{\boldsymbol{\nu}_{t,l}} + \sqrt{\widetilde{\boldsymbol{\nu}}_{t,l}^1})}\right]$$

$$\leq \sum_{l=1}^d |\boldsymbol{G}_{t,l}|\mathbb{E}^{|\mathcal{F}_t}\left[|\boldsymbol{m}_{t,l}|\frac{\sqrt{1-\beta_2}|\boldsymbol{G}_{t,l} - \boldsymbol{g}_{t,l}| + \sqrt{1-\beta_2}\sigma}{\sqrt{\boldsymbol{\nu}_{t,l}\widetilde{\boldsymbol{\nu}}_{t,l}^1}}\right]$$

$$\overset{(\star)}{\leq} \sum_{l=1}^d \frac{\sqrt{1-\beta_2}(1-\beta_1)|\boldsymbol{G}_{t,l}|^2}{4\sigma\widetilde{\boldsymbol{\nu}}_{t,l}^1}\left(\mathbb{E}^{|\mathcal{F}_t}|\boldsymbol{G}_{t,l} - \boldsymbol{g}_{t,l}|^2 + \sigma^2\right) + \sum_{l=1}^d 2\frac{\sqrt{1-\beta_2}}{1-\beta_1}\mathbb{E}^{|\mathcal{F}_t}\sigma\frac{|\boldsymbol{m}_{t,l}|^2}{\boldsymbol{\nu}_{t,l}}$$

$$\leq \sum_{l=1}^d \frac{(1-\beta_1)|\boldsymbol{G}_{t,l}|^2}{2\sqrt{\widetilde{\boldsymbol{\nu}}_{t,l}^1}} + \sum_{l=1}^d 2\frac{\sqrt{1-\beta_2}}{1-\beta_1}\mathbb{E}^{|\mathcal{F}_t}\sigma\frac{|\boldsymbol{m}_{t,l}|^2}{\boldsymbol{\nu}_{t,l}},$$

where inequality $(*)$ is due to the triangle inequality, and inequality $(\star)$ is due to the mean-value

inequality, and the last inequality is due to Assumption 2.

If $i > 0$, then

$$\left| \mathbb{E}^{|\mathcal{F}_{t-i}} \left[ \left\langle \boldsymbol{G}_{t-i}, \left( \frac{1}{\sqrt{\widetilde{\boldsymbol{\nu}}_t^i}} - \frac{1}{\sqrt{\widetilde{\boldsymbol{\nu}}_t^{i+1}}} \right) \odot \boldsymbol{m}_{t-i} \right\rangle \right] \right|$$

$$\leq \sum_{l=1}^d |\boldsymbol{G}_{t-i,l}| \mathbb{E}^{|\mathcal{F}_{t-i}} \left[ |\boldsymbol{m}_{t-i,l}| \left| \frac{1}{\sqrt{\widetilde{\boldsymbol{\nu}}_{t,l}^i}} - \frac{1}{\sqrt{\widetilde{\boldsymbol{\nu}}_{t,l}^{i+1}}} \right| \right]$$

$$\leq \sum_{l=1}^d |\boldsymbol{G}_{t-i,l}| \mathbb{E}^{|\mathcal{F}_{t-i}} \left[ |\boldsymbol{m}_{t-i,l}| \frac{(1-\beta_2)\beta_2^i \left| |\boldsymbol{G}_{t-i,l}|^2 - |\boldsymbol{g}_{t-i,l}|^2 \right| + \sum_{j=0}^{i-1} \beta_2^j (1-\beta_2) \left| |\boldsymbol{G}_{t-i,l}|^2 - |\boldsymbol{G}_{t-i+1,l}|^2 \right|}{\sqrt{\widetilde{\boldsymbol{\nu}}_{t,l}^i \widetilde{\boldsymbol{\nu}}_{t,l}^{i+1}} \left( \sqrt{\widetilde{\boldsymbol{\nu}}_{t,l}^i} + \sqrt{\widetilde{\boldsymbol{\nu}}_{t,l}^{i+1}} \right)} \right]$$

$$\leq \sum_{l=1}^d |\boldsymbol{G}_{t-i,l}| \mathbb{E}^{|\mathcal{F}_{t-i}} \left[ |\boldsymbol{m}_{t-i,l}| \frac{(1-\beta_2)\beta_2^i |\boldsymbol{G}_{t-i,l} - \boldsymbol{g}_{t-i,l}|(|\boldsymbol{G}_{t-i,l}| + |\boldsymbol{g}_{t-i,l}|)}{\sqrt{\widetilde{\boldsymbol{\nu}}_{t,l}^i \widetilde{\boldsymbol{\nu}}_{t,l}^{i+1}} \left( \sqrt{\widetilde{\boldsymbol{\nu}}_{t,l}^i} + \sqrt{\widetilde{\boldsymbol{\nu}}_{t,l}^1} \right)} \right]$$

$$+ \sum_{l=1}^d |\boldsymbol{G}_{t-i,l}| \mathbb{E}^{|\mathcal{F}_{t-i}} \left[ |\boldsymbol{m}_{t-i,l}| \frac{\sum_{j=0}^{i-1} \beta_2^j (1-\beta_2) ||\boldsymbol{G}_{t-i,l}| - |\boldsymbol{G}_{t-i+1,l}||(|\boldsymbol{G}_{t-i,l}| + |\boldsymbol{G}_{t-i+1,l}|)}{\sqrt{\widetilde{\boldsymbol{\nu}}_{t,l}^i \widetilde{\boldsymbol{\nu}}_{t,l}^{i+1}} \left( \sqrt{\widetilde{\boldsymbol{\nu}}_{t,l}^i} + \sqrt{\widetilde{\boldsymbol{\nu}}_{t,l}^1} \right)} \right]$$

Applying Cauchy's inequality, we obtain the RHS of the above inequality is smaller than

$$\sum_{l=1}^d |\boldsymbol{G}_{t-i,l}| \mathbb{E}^{|\mathcal{F}_{t-i}} \left[ |\boldsymbol{m}_{t-i,l}| \frac{\sqrt{1-\beta_2}\sqrt{\beta_2^i}|\boldsymbol{G}_{t-i,l} - \boldsymbol{g}_{t-i,l}|}{\sqrt{\widetilde{\boldsymbol{\nu}}_{t,l}^i \widetilde{\boldsymbol{\nu}}_{t,l}^{i+1}}} \right]$$

$$+ \sum_{l=1}^d |\boldsymbol{G}_{t-i,l}| \mathbb{E}^{|\mathcal{F}_{t-i}} \left[ |\boldsymbol{m}_{t-i,l}| \frac{\sqrt{\sum_{j=0}^{i-1} \beta_2^j (1-\beta_2)|\boldsymbol{G}_{t-i,l} - \boldsymbol{G}_{t-i+1,l}|^2}}{\sqrt{\widetilde{\boldsymbol{\nu}}_{t,l}^i \widetilde{\boldsymbol{\nu}}_{t,l}^{i+1}}} \right]$$

$$\overset{(\star)}{\leq} \sum_{l=1}^d \frac{\sqrt{1-\beta_2}(1-\beta_1)|\boldsymbol{G}_{t-i,l}|^2}{4\sigma \widetilde{\boldsymbol{\nu}}_{t,l}^{i+1}} \left( \mathbb{E}^{|\mathcal{F}_t} |\boldsymbol{G}_{t,l} - \boldsymbol{g}_{t,l}|^2 \right) + \sum_{l=1}^d \frac{\sqrt{1-\beta_2}}{(1-\beta_1)} \beta_2^i \mathbb{E}^{|\mathcal{F}_{t-i}} \sigma \frac{|\boldsymbol{m}_{t-i,l}|^2}{\widetilde{\boldsymbol{\nu}}_{t,l}^i}$$

$$+ \sum_{l=1}^d \frac{\sqrt{1-\beta_2}\sigma(1-\beta_1)|\boldsymbol{G}_{t-i,l}|^2}{4\widetilde{\boldsymbol{\nu}}_{t,l}^{i+1}} + \sum_{l=1}^d \frac{1}{(1-\beta_1)} \mathbb{E}^{|\mathcal{F}_{t-i}} \frac{1}{\sigma} \frac{|\boldsymbol{m}_{t-i,l}|^2}{\widetilde{\boldsymbol{\nu}}_{t,l}^i} \left( \sum_{j=0}^{i-1} \beta_2^j \sqrt{1-\beta_2}|\boldsymbol{G}_{t-i,l} - \boldsymbol{G}_{t-i+1,l}|^2 \right)$$

$$\leq \sum_{l=1}^d \frac{(1-\beta_1)|\boldsymbol{G}_{t-i,l}|^2}{2\sqrt{\widetilde{\boldsymbol{\nu}}_{t,l}^{i+1}}} + \sum_{l=1}^d \frac{\sqrt{1-\beta_2}}{1-\beta_1} \mathbb{E}^{|\mathcal{F}_{t-i}} \sigma \beta_2^i \frac{|\boldsymbol{m}_{t-i,l}|^2}{\boldsymbol{\nu}_{t,l}} + L^2 \frac{\eta^2 \sqrt{1-\beta_2}}{(1-\beta_1)\beta_2^i} \frac{i}{\sigma} \mathbb{E}^{|\mathcal{F}_{t-i}} \left( \sum_{l=1}^d \frac{|\boldsymbol{m}_{t-i,l}|^2}{\boldsymbol{\nu}_{t-i,l}} \right)^2$$

$$\overset{(\circ)}{\leq} \sum_{l=1}^d \frac{(1-\beta_1)|\boldsymbol{G}_{t-i,l}|^2}{2\sqrt{\widetilde{\boldsymbol{\nu}}_{t,l}^{i+1}}} + \sum_{l=1}^d \frac{\sqrt{1-\beta_2}}{(1-\beta_1)} \mathbb{E}^{|\mathcal{F}_{t-i}} \sigma \frac{|\boldsymbol{m}_{t-i,l}|^2}{\boldsymbol{\nu}_{t-i,l}} + L^2 \frac{\eta^2(1-\beta_1)}{(1-\beta_2)^{\frac{1}{2}}(1-\frac{\beta_1^2}{\beta_2})\beta_2^i} \frac{i}{\sigma} d \mathbb{E}^{|\mathcal{F}_{t-i}} \left( \sum_{l=1}^d \frac{|\boldsymbol{m}_{t-i,l}|^2}{\boldsymbol{\nu}_{t-i,l}} \right).$$

Here inequality $(\star)$ is due to the mean-value inequality, and inequality $(\circ)$ is due to Lemma 5. Putting the estimation of $(i)_t^i$ and $(ii)_t^i$ together completes the proof. $\qquad\square$

*Proof of Lemma 2.* To begin with, we have that

$$\sum_{t=1}^T \mathbb{E} \left[ \frac{|\boldsymbol{G}_{t,l}|^2}{\sqrt{\widetilde{\boldsymbol{\nu}}_{t,l}^1}} \mathbf{1}_{|G_{t,l}| \geq \sigma} \right] \leq \sum_{t=1}^T \mathbb{E} \left[ \frac{|\boldsymbol{G}_{t,l}|^2}{\sqrt{\widetilde{\boldsymbol{\nu}}_{t,l}^1}} \right]. \tag{8}$$

On the other hand, we have that

$$\frac{|\boldsymbol{G}_{t,l}|^2}{\sqrt{\widetilde{\boldsymbol{\nu}}_{t,l}^1}} \mathbf{1}_{|G_{t,l}| \geq \sigma} \geq \frac{\frac{2}{3}|\boldsymbol{G}_{t,l}|^2 + \frac{1}{3}\sigma^2}{\sqrt{\widetilde{\boldsymbol{\nu}}_{t,l}^1}} \mathbf{1}_{|G_{t,l}| \geq \sigma} \geq \frac{\frac{1}{3}\mathbb{E}^{|\mathcal{F}_t}|\boldsymbol{g}_{t,l}|^2 + \frac{1-\beta_2}{3}\sigma^2}{\sqrt{\widetilde{\boldsymbol{\nu}}_{t,l}^1}} \mathbf{1}_{|G_{t,l}| \geq \sigma}$$

$$\geq \frac{\frac{1}{3}\mathbb{E}^{|\mathcal{F}_t}|\boldsymbol{g}_{t,l}|^2 + \frac{1-\beta_2}{3}\sigma^2}{\sqrt{\beta_2 \boldsymbol{\nu}_{t-1,l} + (1-\beta_2)\mathbb{E}^{|\mathcal{F}_t}|\boldsymbol{g}_{t,l}|^2 + (1-\beta_2)\sigma^2}} \mathbf{1}_{|G_{t,l}| \geq \sigma}$$

$$\geq \mathbb{E}^{|\mathcal{F}_t} \frac{\frac{1}{3}|\boldsymbol{g}_{t,l}|^2 + \frac{1-\beta_2}{3}\sigma^2}{\sqrt{\beta_2 \boldsymbol{\nu}_{t-1,l} + (1-\beta_2)|\boldsymbol{g}_{t,l}|^2 + (1-\beta_2)\sigma^2}} \mathbf{1}_{|G_{t,l}| \geq \sigma}.$$

Here the last inequality is due to the concavity of $\frac{x}{\sqrt{x+a}}$ with respect to $x$. As a conclusion,

$$\sum_{t=1}^{T}\mathbb{E}\left[\frac{|\boldsymbol{G}_{t,l}|^2}{\sqrt{\widetilde{\boldsymbol{\nu}}_{t,l}^1}}\mathbf{1}_{|\boldsymbol{G}_{t,l}|\geq\sigma}\right] \geq \sum_{t=1}^{T}\mathbb{E}\left[\frac{\left(\frac{1}{3}|\boldsymbol{g}_{t,l}|^2 + \frac{1-\beta_2}{3}\sigma^2\right)}{\sqrt{\beta_2\boldsymbol{\nu}_{t-1,l} + (1-\beta_2)|\boldsymbol{g}_{t,l}|^2 + (1-\beta_2)\sigma^2}}\mathbf{1}_{|\boldsymbol{G}_{t,l}|\geq\sigma}\right]$$

$$\geq \frac{1}{3(1-\beta_2)}\sum_{t=1}^{T}\mathbb{E}\left(\sqrt{\boldsymbol{\nu}_{t,l} + (1-\beta_2)\sigma^2} - \sqrt{\beta_2(\boldsymbol{\nu}_{t-1,l} + (1-\beta_2)\sigma^2)}\right)\mathbf{1}_{|\boldsymbol{G}_{t,l}|\geq\sigma}.$$

On the other hand, as stated in Section 4.2, we define $\{\bar{\boldsymbol{\nu}}_{t,l}\}_{t=0}^{\infty}$ as $\bar{\boldsymbol{\nu}}_{0,l} = \boldsymbol{\nu}_{0,l}$, $\bar{\boldsymbol{\nu}}_{t,l} = \bar{\boldsymbol{\nu}}_{t-1,l} + |g_{t,l}|^2\mathbf{1}_{|\boldsymbol{G}_{t,l}|<\sigma}$. One can easily observe that $\bar{\boldsymbol{\nu}}_{t,l} \leq \boldsymbol{\nu}_{t,l}$, and thus

$$\sum_{t=1}^{T}\mathbb{E}\left(\sqrt{\boldsymbol{\nu}_{t,l} + (1-\beta_2)\sigma^2} - \sqrt{\beta_2(\boldsymbol{\nu}_{t-1,l} + (1-\beta_2)\sigma^2)}\right)\mathbf{1}_{|\boldsymbol{G}_{t,l}|<\sigma}$$

$$= \sum_{t=1}^{T}\mathbb{E}\left(\sqrt{\beta_2\boldsymbol{\nu}_{t-1,l} + (1-\beta_2)|g_{t,l}|^2 + (1-\beta_2)\sigma^2} - \sqrt{\beta_2(\boldsymbol{\nu}_{t-1,l} + (1-\beta_2)\sigma^2)}\right)\mathbf{1}_{|\boldsymbol{G}_{t,l}|<\sigma}$$

$$\leq \sum_{t=1}^{T}\mathbb{E}\left(\sqrt{\beta_2\bar{\boldsymbol{\nu}}_{t-1,l} + (1-\beta_2)|g_{t,l}|^2 + (1-\beta_2)\sigma^2} - \sqrt{\beta_2(\bar{\boldsymbol{\nu}}_{t-1,l} + (1-\beta_2)\sigma^2)}\right)\mathbf{1}_{|\boldsymbol{G}_{t,l}|<\sigma}$$

$$\leq \sum_{t=1}^{T}\mathbb{E}(\sqrt{\beta_2\bar{\boldsymbol{\nu}}_{t-1,l} + (1-\beta_2)|g_{t,l}|^2\mathbf{1}_{|\boldsymbol{G}_{t,l}|<\sigma} + (1-\beta_2)\sigma^2} - \sqrt{\beta_2(\bar{\boldsymbol{\nu}}_{t-1,l} + (1-\beta_2)\sigma^2)})$$

$$= \sum_{t=1}^{T}\mathbb{E}(\sqrt{\bar{\boldsymbol{\nu}}_{t,l} + (1-\beta_2)\sigma^2} - \sqrt{\beta_2(\bar{\boldsymbol{\nu}}_{t-1,l} + (1-\beta_2)\sigma^2)})$$

$$= \mathbb{E}\sqrt{\bar{\boldsymbol{\nu}}_{T,l} + (1-\beta_2)\sigma^2} + (1-\sqrt{\beta_2})\sum_{t=1}^{T-1}\mathbb{E}\sqrt{\bar{\boldsymbol{\nu}}_{t,l} + (1-\beta_2)\sigma^2} - \mathbb{E}\sqrt{\beta_2(\bar{\boldsymbol{\nu}}_{0,l} + (1-\beta_2)\sigma^2)}.$$

All in all, summing the above two inequalities together, we obtain that

$$\mathbb{E}\sqrt{\boldsymbol{\nu}_{T,l} + (1-\beta_2)\sigma^2} + (1-\sqrt{\beta_2})\sum_{t=1}^{T-1}\mathbb{E}\sqrt{\boldsymbol{\nu}_{t,l} + (1-\beta_2)\sigma^2} - \mathbb{E}\sqrt{\beta_2(\boldsymbol{\nu}_{0,l} + (1-\beta_2)\sigma^2)}$$

$$= \sum_{t=1}^{T}\mathbb{E}(\sqrt{\boldsymbol{\nu}_{t,l} + (1-\beta_2)\sigma^2} - \sqrt{\beta_2(\boldsymbol{\nu}_{t-1,l} + (1-\beta_2)\sigma^2)})$$

$$\leq \sum_{t=1}^{T}\mathbb{E}(\sqrt{\boldsymbol{\nu}_{t,l} + (1-\beta_2)\sigma^2} - \sqrt{\beta_2(\boldsymbol{\nu}_{t-1,l} + (1-\beta_2)\sigma^2)})\mathbf{1}_{|\boldsymbol{G}_{t,l}|\geq\sigma}$$

$$+ \sum_{t=1}^{T}\mathbb{E}(\sqrt{\boldsymbol{\nu}_{t,l} + (1-\beta_2)\sigma^2} - \sqrt{\beta_2(\boldsymbol{\nu}_{t-1,l} + (1-\beta_2)\sigma^2)})\mathbf{1}_{|\boldsymbol{G}_{t,l}|<\sigma}$$

$$= 3(1-\beta_2)\sum_{t=1}^{T}\mathbb{E}\left[\frac{|\boldsymbol{G}_{t,l}|^2}{\sqrt{\widetilde{\boldsymbol{\nu}}_{t,l}^1}}\right] + \mathbb{E}\sqrt{\bar{\boldsymbol{\nu}}_{T,l} + (1-\beta_2)\sigma^2} + (1-\sqrt{\beta_2})\sum_{t=1}^{T-1}\mathbb{E}\sqrt{\bar{\boldsymbol{\nu}}_{t,l} + (1-\beta_2)\sigma^2} - \mathbb{E}\sqrt{\beta_2(\bar{\boldsymbol{\nu}}_{0,l} + (1-\beta_2)\sigma^2)}.$$

As $\mathbb{E}\sqrt{\boldsymbol{\nu}_{T,l} + (1-\beta_2)\sigma^2} \geq \mathbb{E}\sqrt{\bar{\boldsymbol{\nu}}_{T,l} + (1-\beta_2)\sigma^2}$ and $\mathbb{E}\sqrt{\boldsymbol{\nu}_{0,l} + (1-\beta_2)\sigma^2} = \mathbb{E}\sqrt{\bar{\boldsymbol{\nu}}_{0,l} + (1-\beta_2)\sigma^2}$, we obtain that

$$(1-\sqrt{\beta_2})\sum_{t=1}^{T}\mathbb{E}\sqrt{\boldsymbol{\nu}_{t,l} + (1-\beta_2)\sigma^2} \leq 3(1-\beta_2)\sum_{t=1}^{T}\mathbb{E}\left[\frac{|\boldsymbol{G}_{t,l}|^2}{\sqrt{\widetilde{\boldsymbol{\nu}}_{t,l}^1}}\right] + (1-\sqrt{\beta_2})\sum_{t=1}^{T}\mathbb{E}\sqrt{\bar{\boldsymbol{\nu}}_{t,l} + (1-\beta_2)\sigma^2}$$

$$\leq 3(1-\beta_2)\sum_{t=1}^{T}\mathbb{E}\left[\frac{|\boldsymbol{G}_{t,l}|^2}{\sqrt{\widetilde{\boldsymbol{\nu}}_{t,l}^1}}\right] + (1-\sqrt{\beta_2})\sum_{t=1}^{T}\sqrt{\mathbb{E}\bar{\boldsymbol{\nu}}_{t,l} + (1-\beta_2)\sigma^2}$$

$$\leq 3(1-\beta_2)\sum_{t=1}^{T}\mathbb{E}\left[\frac{|\boldsymbol{G}_{t,l}|^2}{\sqrt{\widetilde{\boldsymbol{\nu}}_{t,l}^1}}\right] + (1-\sqrt{\beta_2})\sum_{t=1}^{T}\sqrt{\bar{\boldsymbol{\nu}}_{0,l} + (3-\beta_2)\sigma^2}.$$

$$(9)$$

Leveraging Eq. (4), we then obtain that

$$\sum_{t=1}^{T}\sum_{l=1}^{d}\mathbb{E}\sqrt{\boldsymbol{\nu}_{t,l}+(1-\beta_2)\sigma^2}$$

$$\leq 3(1+\sqrt{\beta_2})\sum_{t=1}^{T}\mathbb{E}\left[\frac{|\boldsymbol{G}_{t,l}|^2}{\sqrt{\widetilde{\boldsymbol{\nu}}_{t,l}^1}}\right]+\sum_{t=1}^{T}\sum_{l=1}^{d}\sqrt{\boldsymbol{\nu}_{0,l}+(3-\beta_2)\sigma^2}$$

$$\leq \frac{12}{(1-\beta_1)\eta}\left(f(\boldsymbol{w}_1)+2\sum_{l=1}^{d}C_1\left(\mathbb{E}\ln\left(\frac{\sqrt{\boldsymbol{\nu}_{T,l}+(1-\beta_2)\sigma^2}}{\boldsymbol{\nu}_{0,l}}\right)-T\ln\beta_2\right)\right)+T\sum_{l=1}^{d}\sqrt{\boldsymbol{\nu}_{0,l}+(3-\beta_2)\sigma^2}$$

$$\leq \frac{12}{(1-\beta_1)\eta}\left(f(\boldsymbol{w}_1)+2\sum_{l=1}^{d}C_1\left(\mathbb{E}\ln\left(\frac{\sum_{t=1}^{T}\sqrt{\boldsymbol{\nu}_{t,l}+(1-\beta_2)\sigma^2}}{\boldsymbol{\nu}_{0,l}}\right)-T\ln\beta_2\right)\right)+T\sum_{l=1}^{d}\sqrt{\boldsymbol{\nu}_{0,l}+(3-\beta_2)\sigma^2}$$

$$\leq \frac{12}{(1-\beta_1)\eta}\left(f(\boldsymbol{w}_1)+2\sum_{l=1}^{d}C_1\left(\ln\left(\frac{\mathbb{E}\sum_{t=1}^{T}\sum_{m=1}^{d}\sqrt{\boldsymbol{\nu}_{t,m}+(1-\beta_2)\sigma^2}}{\boldsymbol{\nu}_{0,l}}\right)-T\ln\beta_2\right)\right)+T\sum_{l=1}^{d}\sqrt{\boldsymbol{\nu}_{0,l}+(3-\beta_2)\sigma^2},$$

where in the last inequality we use the concavity of $h(x)=\ln x$. Solving the above inequality with respect to $\sum_{t=1}^{T}\sum_{l=1}^{d}\mathbb{E}\sqrt{\boldsymbol{\nu}_{t,l}+(1-\beta_2)\sigma^2}$ then gives

$$\sum_{t=1}^{T}\sum_{l=1}^{d}\mathbb{E}\sqrt{\boldsymbol{\nu}_{t,l}+(1-\beta_2)\sigma^2}\leq 2T\sum_{l=1}^{d}\sqrt{\boldsymbol{\nu}_{0,l}+(3-\beta_2)\sigma^2}+4dC_1\ln dC_1$$

$$+\frac{24}{(1-\beta_1)\eta}\left(f(\boldsymbol{w}_1)+2\sum_{l=1}^{d}C_1\left(\ln\left(\frac{1}{\boldsymbol{\nu}_{0,l}}\right)-T\ln\beta_2\right)\right).$$

The proof is then completed.

$\square$

## C.2 Proof of Theorem 1

*Proof of Theorem 1.* As stated in Section 4.2, the proof involves solving two key challenges. We respectively divide the proof into two stages according to the challenges.

**Stage I.** Based on Lemma 1, we can estimate $\mathbb{E}\langle\boldsymbol{G}_t,\frac{1}{\sqrt{\widetilde{\boldsymbol{\nu}}_t}}\odot\boldsymbol{m}_t\rangle=F_t^0$ recursively. Specifically, we have

$$F_t^0\geq\sum_{i=0}^{t-1}\beta_1^i\left(\frac{(1-\beta_1)}{2}\mathbb{E}\left[\left\|\frac{1}{\sqrt[4]{\widetilde{\boldsymbol{\nu}}_t^{i+1}}}\odot\boldsymbol{G}_{t-i}\right\|^2\right]-\beta_1\mathbb{E}\left[\|\boldsymbol{w}_{t-i}-\boldsymbol{w}_{t-i-1}\|\left\|\frac{1}{\sqrt{\widetilde{\boldsymbol{\nu}}_t^{i+1}}}\odot\boldsymbol{m}_{t-i-1}\right\|\right]\right.$$

$$\left.-\left(2\frac{\sqrt{1-\beta_2}}{1-\beta_1}\sigma+L^2\frac{\eta^2(1-\beta_1)}{(1-\beta_2)^{\frac{1}{2}}(1-\frac{\beta_1^2}{\beta_2})\beta_2^i}\frac{i}{\sigma}d\right)\mathbb{E}\left\|\frac{1}{\sqrt{\boldsymbol{\nu}_{t-i}}}\odot\boldsymbol{m}_{t-i}\right\|^2\right)$$

$$\geq\frac{(1-\beta_1)}{2}\mathbb{E}\left[\left\|\frac{1}{\sqrt[4]{\widetilde{\boldsymbol{\nu}}_t^1}}\odot\boldsymbol{G}_t\right\|^2\right]-\sum_{i=0}^{t-1}\beta_1^i\left(\beta_1\mathbb{E}\left[\|\boldsymbol{w}_{t-i}-\boldsymbol{w}_{t-i-1}\|\left\|\frac{1}{\sqrt{\widetilde{\boldsymbol{\nu}}_t^{i+1}}}\odot\boldsymbol{m}_{t-i-1}\right\|\right]\right.$$

$$\left.+\left(2\frac{\sqrt{1-\beta_2}}{1-\beta_1}\sigma+L^2\frac{\eta^2(1-\beta_1)}{(1-\beta_2)^{\frac{1}{2}}(1-\frac{\beta_1^2}{\beta_2})\beta_2^i}\frac{i}{\sigma}d\right)\mathbb{E}\left\|\frac{1}{\sqrt{\boldsymbol{\nu}_{t-i}}}\odot\boldsymbol{m}_{t-i}\right\|^2\right)$$

413 Applying the above inequality back to Eq. (3) then gives

$$\mathbb{E}f(\boldsymbol{w}_{t+1})$$

$$\leq \mathbb{E}f(\boldsymbol{w}_t) - \frac{(1-\beta_1)\eta}{2}\mathbb{E}\left[\left\|\frac{1}{\sqrt[4]{\widetilde{\boldsymbol{\nu}}_t^1}}\odot\boldsymbol{G}_t\right\|^2\right] + \frac{L}{2}\eta^2\mathbb{E}\left\|\frac{1}{\sqrt{\boldsymbol{\nu}_t}}\odot\boldsymbol{m}_t\right\|^2 + \eta\sum_{i=0}^{t-1}\beta_1^i\left(\beta_1\mathbb{E}\left[\|\boldsymbol{w}_{t-i}-\boldsymbol{w}_{t-i-1}\|\right.\right.$$

$$\times\left.\left\|\frac{1}{\sqrt{\widetilde{\boldsymbol{\nu}}_t^{i+1}}}\odot\boldsymbol{m}_{t-i-1}\right\|\right] + \left(2\frac{\sqrt{1-\beta_2}}{1-\beta_1}\sigma + L^2\frac{\eta^2(1-\beta_1)}{(1-\beta_2)^{\frac{1}{2}}(1-\frac{\beta_1^2}{\beta_2})\beta_2^i}\frac{i}{\sigma}d\right)\mathbb{E}\left\|\frac{1}{\sqrt{\boldsymbol{\nu}_{t-i}}}\odot\boldsymbol{m}_{t-i}\right\|^2\right).$$

414 Summing the above inequality with respect to $t$ then gives

$$\mathbb{E}f(\boldsymbol{w}_{T+1})$$

$$\leq f(\boldsymbol{w}_1) - \sum_{t=1}^{T}\frac{(1-\beta_1)\eta}{2}\mathbb{E}\left[\left\|\frac{1}{\sqrt{\widetilde{\boldsymbol{\nu}}_t^1}}\odot\boldsymbol{G}_t\right\|^2\right] + \left(\frac{L}{2}\eta^2 + 2\frac{\sqrt{1-\beta_2}}{(1-\beta_1)^2}\eta\sigma + \frac{\eta^2\beta_1}{\sqrt{\beta_2}(1-\frac{\beta_1}{\sqrt{\beta_2}})}\right.$$

$$\left. + L^2\frac{\beta_1\eta^3(1-\beta_1)}{\beta_2(1-\beta_2)^{\frac{1}{2}}(1-\frac{\beta_1^2}{\beta_2})(1-\frac{\beta_1}{\beta_2})^2}\frac{d}{\sigma}\right)\sum_{t=1}^{T}\mathbb{E}\left\|\frac{1}{\sqrt{\boldsymbol{\nu}_t}}\odot\boldsymbol{m}_t\right\|^2.$$

415 Here the inequality is due to

$$2\frac{\sqrt{1-\beta_2}}{1-\beta_1}\eta\sum_{t=1}^{T}\sum_{i=0}^{t-1}\beta_1^i\mathbb{E}\sigma\left\|\frac{1}{\sqrt{\boldsymbol{\nu}_{t-i}}}\odot\boldsymbol{m}_{t-i}\right\|^2 = 2\frac{\sqrt{1-\beta_2}}{1-\beta_1}\eta\sigma\sum_{i=1}^{T}\sum_{t=i}^{T}\beta_1^{t-i}\mathbb{E}\left\|\frac{1}{\sqrt{\boldsymbol{\nu}_t}}\odot\boldsymbol{m}_t\right\|^2$$

$$\leq 2\frac{\sqrt{1-\beta_2}}{(1-\beta_1)^2}\eta\sigma\sum_{i=1}^{T}\mathbb{E}\left\|\frac{1}{\sqrt{\boldsymbol{\nu}_i}}\odot\boldsymbol{m}_i\right\|^2,$$

416

$$\eta\sum_{t=1}^{T}\sum_{i=0}^{t-1}\beta_1^{i+1}\mathbb{E}\left[\|\boldsymbol{w}_{t-i}-\boldsymbol{w}_{t-i-1}\|\left\|\frac{1}{\sqrt{\widetilde{\boldsymbol{\nu}}_t^{i+1}}}\odot\boldsymbol{m}_{t-i-1}\right\|\right]$$

$$\leq\eta\sum_{t=1}^{T}\sum_{i=0}^{t-1}\frac{\beta_1^{i+1}}{\sqrt{\beta_2^{i+1}}}\mathbb{E}\left[\|\boldsymbol{w}_{t-i}-\boldsymbol{w}_{t-i-1}\|\left\|\frac{1}{\sqrt{\boldsymbol{\nu}_{t-i-1}}}\odot\boldsymbol{m}_{t-i-1}\right\|\right]$$

$$=\eta^2\sum_{t=1}^{T}\sum_{i=0}^{t-1}\frac{\beta_1^{i+1}}{\sqrt{\beta_2^{i+1}}}\mathbb{E}\left[\left\|\frac{1}{\sqrt{\boldsymbol{\nu}_{t-i-1}}}\odot\boldsymbol{m}_{t-i-1}\right\|^2\right] = \eta^2\sum_{i=0}^{T-1}\sum_{t=i+1}^{T}\frac{\beta_1^{t-i}}{\sqrt{\beta_2^{t-i}}}\mathbb{E}\left[\left\|\frac{1}{\sqrt{\boldsymbol{\nu}_i}}\odot\boldsymbol{m}_i\right\|^2\right]$$

$$\leq\frac{\eta^2\beta_1}{\sqrt{\beta_2}(1-\frac{\beta_1}{\sqrt{\beta_2}})}\sum_{i=0}^{T-1}\mathbb{E}\left[\left\|\frac{1}{\sqrt{\boldsymbol{\nu}_i}}\odot\boldsymbol{m}_i\right\|^2\right] = \frac{\eta^2\beta_1}{\sqrt{\beta_2}(1-\frac{\beta_1}{\sqrt{\beta_2}})}\sum_{i=1}^{T-1}\mathbb{E}\left[\left\|\frac{1}{\sqrt{\boldsymbol{\nu}_i}}\odot\boldsymbol{m}_i\right\|^2\right],$$

417 and

$$L^2\frac{\eta^3(1-\beta_1)}{(1-\beta_2)^{\frac{1}{2}}(1-\frac{\beta_1^2}{\beta_2})}\frac{d}{\sigma}\sum_{t=1}^{T}\sum_{i=0}^{t-1}\frac{\beta_1^i}{\beta_2^i}i\mathbb{E}\left\|\frac{1}{\sqrt{\boldsymbol{\nu}_{t-i}}}\odot\boldsymbol{m}_{t-i}\right\|^2$$

$$=L^2\frac{\eta^3(1-\beta_1)}{(1-\beta_2)^{\frac{1}{2}}(1-\frac{\beta_1^2}{\beta_2})}\frac{d}{\sigma}\sum_{i=1}^{T}\sum_{t=i}^{T}\frac{\beta_1^{t-i}}{\beta_2^{t-i}}(t-i)\mathbb{E}\left\|\frac{1}{\sqrt{\boldsymbol{\nu}_i}}\odot\boldsymbol{m}_i\right\|^2 \leq L^2\frac{\beta_1\eta^3(1-\beta_1)}{\beta_2(1-\beta_2)^{\frac{1}{2}}(1-\frac{\beta_1^2}{\beta_2})(1-\frac{\beta_1}{\beta_2})^2}\frac{d}{\sigma}\sum_{i=1}^{T}\mathbb{E}\left\|\frac{1}{\sqrt{\boldsymbol{\nu}_i}}\odot\boldsymbol{m}_i\right\|^2.$$

418 Applying Lemma 4, we obtain that

$$\mathbb{E}f(\boldsymbol{w}_{T+1})$$

$$\leq f(\boldsymbol{w}_1) + \sum_{l=1}^{d}\left(\frac{L}{2}\eta^2 + 2\frac{\sqrt{1-\beta_2}}{(1-\beta_1)^2}\eta\sigma + \frac{\eta^2\beta_1}{\sqrt{\beta_2}(1-\frac{\beta_1}{\sqrt{\beta_2}})} + L^2\frac{\beta_1\eta^3(1-\beta_1)}{\beta_2(1-\beta_2)^{\frac{1}{2}}(1-\frac{\beta_1^2}{\beta_2})(1-\frac{\beta_1}{\beta_2})^2}\frac{d}{\sigma}\frac{(1-\beta_1)^2}{(1-\frac{\beta_1}{\sqrt{\beta_2}})^2}\right)\frac{1}{1-\beta_2}$$

$$\times\left(\mathbb{E}\ln\left(\frac{\boldsymbol{\nu}_{T,l}}{\boldsymbol{\nu}_{0,l}}\right) - T\ln\beta_2\right) - \sum_{t=1}^{T}\frac{(1-\beta_1)\eta}{2}\mathbb{E}\left[\left\|\frac{1}{\sqrt[4]{\widetilde{\boldsymbol{\nu}}_t^1}}\odot\boldsymbol{G}_t\right\|^2\right].$$

419    The proof of Stage I is completed.

420    **Stage II.** According to Cauchy's inequality, we have

$$\left( \mathbb{E} \sum_{t=1}^{T} \|\boldsymbol{G}_t\|_1 \right)^2 \leq \left( \sum_{t=1}^{T} \mathbb{E} \left[ \left\| \frac{1}{\sqrt[4]{\widetilde{\boldsymbol{\nu}}_t^1}} \odot \boldsymbol{G}_t \right\|^2 \right] \right) \left( \sum_{t=1}^{T} \mathbb{E} \left[ \left\| \sqrt[4]{\widetilde{\boldsymbol{\nu}}_t^1} \right\|^2 \right] \right). \tag{10}$$

421    Meanwhile, by Lemma 2, we have

$$\sum_{t=1}^{T} \mathbb{E} \left[ \left\| \sqrt[4]{\widetilde{\boldsymbol{\nu}}_t^1} \right\|^2 \right] = \mathbb{E} \left[ \sum_{t=1}^{T} \sum_{l=1}^{d} \sqrt{\beta_2 \boldsymbol{\nu}_{t-1,l} + (1-\beta_2)|\boldsymbol{G}_{t,l}|^2 + (1-\beta_2)\sigma^2} \right]$$

$$\leq \mathbb{E} \left[ \sum_{t=1}^{T} \sum_{l=1}^{d} \left( \sqrt{\beta_2 \boldsymbol{\nu}_{t-1,l} + (1-\beta_2)\sigma^2} + \sqrt{1-\beta_2}|\boldsymbol{G}_{t,l}| \right) \right]$$

$$= \mathbb{E} \left[ \sum_{t=1}^{T} \sum_{l=1}^{d} \sqrt{\beta_2 \boldsymbol{\nu}_{t-1,l} + (1-\beta_2)\sigma^2} + \sum_{t=1}^{T} \sqrt{1-\beta_2}\|\boldsymbol{G}_t\|_1 \right]$$

$$\leq \mathbb{E} \left[ \sum_{t=1}^{T} \sqrt{1-\beta_2}\|\boldsymbol{G}_t\|_1 \right] + 2T\sqrt{\boldsymbol{\nu}_{0,l} + (3-\beta_2)\sigma^2} + 4dC_1 \ln dC_1$$

$$+ \frac{24}{(1-\beta_1)\eta} \left( f(\boldsymbol{w}_1) + 2\sum_{l=1}^{d} C_1 \left( \ln\left(\frac{1}{\boldsymbol{\nu}_{0,l}}\right) - T\ln\beta_2 \right) \right).$$

422    Combining the above inequality and Eq. (10) gives

$$\left( \mathbb{E} \sum_{t=1}^{T} \|\boldsymbol{G}_t\|_1 \right)^2 \leq \frac{2}{(1-\beta_1)\eta} \left( f(\boldsymbol{w}_1) + \sum_{l=1}^{d} C_1 \left( \mathbb{E} \ln\left(\frac{\boldsymbol{\nu}_{T,l}}{\boldsymbol{\nu}_{0,l}}\right) - T\ln\beta_2 \right) \right)$$

$$\times \left( \mathbb{E} \left[ \sum_{t=1}^{T} \sqrt{1-\beta_2}\|\boldsymbol{G}_t\|_1 \right] + 2T\sqrt{\boldsymbol{\nu}_{0,l} + (3-\beta_2)\sigma^2} + 4dC_1 \ln dC_1 \right.$$

$$+ \frac{24}{(1-\beta_1)\eta} \left( f(\boldsymbol{w}_1) + 2\sum_{l=1}^{d} C_1 \left( \ln\left(\frac{1}{\boldsymbol{\nu}_{0,l}}\right) - T\ln\beta_2 \right) \right) \Bigg).$$

423    Solving the above quadratic inequality with respect to $\mathbb{E} \sum_{t=1}^{T} \|\boldsymbol{G}_t\|_1$ then completes the proof.

424    $\square$

# D    Proof of Theorem 2

426    *Proof.* According to Stage I in the proof of Theorem 1, we obtain

$$\mathbb{E} f(\boldsymbol{w}_{T+1})$$

$$\leq f(\boldsymbol{w}_1) + \sum_{l=1}^{d} \left( \frac{L}{2}\eta^2 + 2\frac{\sqrt{1-\beta_2}}{(1-\beta_1)^2}\eta\sigma + \frac{\eta^2\beta_1}{\sqrt{\beta_2}(1-\frac{\beta_1}{\sqrt{\beta_2}})} + L^2 \frac{\beta_1\eta^3(1-\beta_1)}{\beta_2(1-\beta_2)^{\frac{1}{2}}(1-\frac{\beta_1^2}{\beta_2})(1-\frac{\beta_1}{\beta_2})^2} \frac{d}{\sigma} \frac{(1-\beta_1)^2}{(1-\frac{\beta_1}{\sqrt{\beta_2}})^2} \right) \frac{1}{1-\beta_2}$$

$$\times \mathbb{E} \left( \ln\left(\frac{\boldsymbol{\nu}_{T,l}}{\boldsymbol{\nu}_{0,l}}\right) - T\ln\beta_2 \right) - \sum_{t=1}^{T} \frac{(1-\beta_1)\eta}{2} \mathbb{E} \left[ \left\| \frac{1}{\sqrt[4]{\widetilde{\boldsymbol{\nu}}_t^1}} \odot \boldsymbol{G}_t \right\|^2 \right].$$

427    Applying the definition of $\eta$, $\beta_1$, and $\beta_2$, we obtain that

$$\sum_{t=1}^{T} \mathbb{E} \left[ \left\| \frac{1}{\sqrt[4]{\widetilde{\boldsymbol{\nu}}_t^1}} \odot \boldsymbol{G}_t \right\|^2 \right] \leq \frac{2\sqrt{T}}{\sqrt{b}} \left( D_1 + \frac{D_2}{d} \sum_{l=1}^{d} \mathbb{E} \ln \boldsymbol{\nu}_{T,l} \right). \tag{11}$$

Meanshile, we have that

$$
\frac{|\boldsymbol{G}_{t,l}|^2}{\sqrt{\widetilde{\boldsymbol{\nu}}_{t,l}^1}}\mathbf{1}_{|\boldsymbol{G}_{t,l}|\geq\sigma} \geq \frac{\frac{1}{2}\mathbb{E}^{|\mathcal{F}_t}|\boldsymbol{g}_{t,l}|^2}{\sqrt{\widetilde{\boldsymbol{\nu}}_{t,l}^1}}\mathbf{1}_{|\boldsymbol{G}_{t,l}|\geq\sigma}
$$

$$
=\frac{\frac{1}{2}\mathbb{E}^{|\mathcal{F}_t}|\boldsymbol{g}_{t,l}|^2}{\sqrt{\beta_2\boldsymbol{\nu}_{t-1,l}+(1-\beta_2)\mathbb{E}^{|\mathcal{F}_t}|\boldsymbol{g}_{t,l}|^2+(1-\beta_2)\sigma^2}}\mathbf{1}_{|\boldsymbol{G}_{t,l}|\geq\sigma}
$$

$$
\geq\frac{1}{2}\mathbb{E}^{|\mathcal{F}_t}\frac{|\boldsymbol{g}_{t,l}|^2}{\sqrt{\beta_2\boldsymbol{\nu}_{t-1,l}+(1-\beta_2)|\boldsymbol{g}_{t,l}|^2+(1-\beta_2)\sigma^2}}\mathbf{1}_{|\boldsymbol{G}_{t,l}|\geq\sigma}
$$

$$
\geq\frac{1}{2\sqrt{1-\beta_2}}\mathbb{E}^{|\mathcal{F}_t}\frac{|\boldsymbol{g}_{t,l}|^2}{\sqrt{\frac{\boldsymbol{\nu}_{0,l}}{1-\beta_2}+\sum_{s=1}^{T}|g_{s,l}|^2+\sigma^2}}\mathbf{1}_{|\boldsymbol{G}_{t,l}|\geq\sigma},
$$

where the last inequality is due to that

$$
\beta_2\boldsymbol{\nu}_{t-1,l}+(1-\beta_2)|\boldsymbol{g}_{t,l}|^2 = (1-\beta_2)\sum_{s=1}^{t}\beta_2^{t-s}|\boldsymbol{g}_{s,l}|^2 + \beta_2^t\boldsymbol{\nu}_{0,l}
$$

$$
\leq(1-\beta_2)\sum_{s=1}^{T}|\boldsymbol{g}_{s,l}|^2 + \boldsymbol{\nu}_{0,l}. \tag{12}
$$

Furthermore, we have

$$
\frac{\sigma^2+\frac{\boldsymbol{\nu}_{0,l}}{1-\beta_2}}{\sqrt{\frac{\boldsymbol{\nu}_{0,l}}{1-\beta_2}+\sum_{s=1}^{T}|g_{s,l}|^2+\sigma^2}}+\sum_{t=1}^{T}\mathbb{E}\frac{|\boldsymbol{g}_{t,l}|^2}{\sqrt{\frac{\boldsymbol{\nu}_{0,l}}{1-\beta_2}+\sum_{s=1}^{T}|g_{s,l}|^2+\sigma^2}}\mathbf{1}_{|\boldsymbol{G}_{t,l}|<\sigma}
$$

$$
\leq\frac{\sigma^2+\frac{\boldsymbol{\nu}_{0,l}}{1-\beta_2}}{\sqrt{\frac{\boldsymbol{\nu}_{0,l}}{1-\beta_2}+\sum_{s=1}^{T}|g_{s,l}|^2+\sigma^2}}+\sum_{t=1}^{T}\mathbb{E}\frac{|\boldsymbol{g}_{t,l}|^2}{\sqrt{\frac{\boldsymbol{\nu}_{0,l}}{1-\beta_2}+\sum_{s=1}^{T}|g_{s,l}|^2\mathbf{1}_{|\boldsymbol{G}_{s,l}|<\sigma}+\sigma^2}}\mathbf{1}_{|\boldsymbol{G}_{t,l}|<\sigma}
$$

$$
=\mathbb{E}\sqrt{\frac{\boldsymbol{\nu}_{0,l}}{1-\beta_2}+\sum_{s=1}^{T}|g_{s,l}|^2\mathbf{1}_{|\boldsymbol{G}_{s,l}|<\sigma}+\sigma^2} \leq \sqrt{\frac{\boldsymbol{\nu}_{0,l}}{1-\beta_2}+\mathbb{E}\sum_{s=1}^{T}|g_{s,l}|^2\mathbf{1}_{|\boldsymbol{G}_{s,l}|<\sigma}+\sigma^2}
$$

$$
\leq\sqrt{\frac{\boldsymbol{\nu}_{0,l}}{1-\beta_2}+2\sigma^2 T+\sigma^2}.
$$

Conclusively, we obtain

$$
\mathbb{E}\sqrt{\frac{\boldsymbol{\nu}_{0,l}}{1-\beta_2}+\sum_{s=1}^{T}|g_{s,l}|^2+\sigma^2}
$$

$$
=\frac{\sigma^2+\frac{\boldsymbol{\nu}_{0,l}}{1-\beta_2}}{\sqrt{\frac{\boldsymbol{\nu}_{0,l}}{1-\beta_2}+\sum_{s=1}^{T}|g_{s,l}|^2+\sigma^2}}+\sum_{t=1}^{T}\mathbb{E}\frac{|\boldsymbol{g}_{t,l}|^2}{\sqrt{\frac{\boldsymbol{\nu}_{0,l}}{1-\beta_2}+\sum_{s=1}^{T}|g_{s,l}|^2+\sigma^2}}\mathbf{1}_{|\boldsymbol{G}_{t,l}|<\sigma}
$$

$$
+\sum_{t=1}^{T}\mathbb{E}\frac{|\boldsymbol{g}_{t,l}|^2}{\sqrt{\frac{\boldsymbol{\nu}_{0,l}}{1-\beta_2}+\sum_{s=1}^{T}|g_{s,l}|^2+\sigma^2}}\mathbf{1}_{|\boldsymbol{G}_{t,l}|\geq\sigma}
$$

$$
\leq\sqrt{\frac{\boldsymbol{\nu}_{0,l}}{1-\beta_2}+2\sigma^2 T+\sigma^2}+2\sqrt{1-\beta_2}\sum_{t=1}^{T}\frac{|\boldsymbol{G}_{t,l}|^2}{\sqrt{\widetilde{\boldsymbol{\nu}}_{t,l}^1}}\mathbf{1}_{|\boldsymbol{G}_{t,l}|\geq\sigma}.
$$

Summing the above inequality with respect to $l$ then gives

$$\sum_{l=1}^{d} \mathbb{E} \sqrt{\frac{\boldsymbol{\nu}_{0,l}}{1-\beta_2} + \sum_{s=1}^{T} |g_{s,l}|^2 + \sigma^2}$$

$$\leq \sum_{l=1}^{d} \sqrt{\frac{\boldsymbol{\nu}_{0,l}}{1-\beta_2} + 2\sigma^2 T + \sigma^2} + 2\sqrt{1-\beta_2} \sum_{l=1}^{d} \sum_{t=1}^{T} \frac{|\boldsymbol{G}_{t,l}|^2}{\sqrt{\widetilde{\boldsymbol{\nu}}_{t,l}^1}} \mathbf{1}_{|\boldsymbol{G}_{t,l}|\geq\sigma}$$

$$\leq \sum_{l=1}^{d} \sqrt{\frac{\boldsymbol{\nu}_{0,l}}{1-\beta_2} + 2\sigma^2 T + \sigma^2}$$

$$+ \frac{4\sqrt{b}}{a(1-c)} f(\boldsymbol{w}_1) + \sum_{l=1}^{d} \frac{2}{ab\sqrt{b}} \left( La^2 + 4\frac{a\sqrt{b}\sigma}{(1-c)^2} + 2\frac{a^2 c}{1-c} + 2\frac{L^2 c a^3 d}{\sqrt{b}(1-c)^5\sigma} \right) \left( \mathbb{E}\ln\left(\frac{\boldsymbol{\nu}_{T,l}}{\boldsymbol{\nu}_{0,l}}\right) + b \right)$$

$$= \sum_{l=1}^{d} \sqrt{\frac{\boldsymbol{\nu}_{0,l}}{1-\beta_2} + 2\sigma^2 T + \sigma^2} + \sum_{l=1}^{d} \frac{2}{ab\sqrt{b}} \left( La^2 + 4\frac{a\sqrt{b}\sigma}{(1-c)^2} + 2\frac{a^2 c}{1-c} + 4\frac{L^2 c a^3 d}{\sqrt{b}(1-c)^5\sigma} \right) \mathbb{E}\ln\left(\sqrt{\boldsymbol{\nu}_{T,l}}\right)$$

$$+ \frac{4\sqrt{b}}{a(1-c)} f(\boldsymbol{w}_1) + \sum_{l=1}^{d} \frac{2}{ab\sqrt{b}} \left( La^2 + 4\frac{a\sqrt{b}\sigma}{(1-c)^2} + 2\frac{a^2 c}{1-c} + 2\frac{L^2 c a^3 d}{\sqrt{b}(1-c)^5\sigma} \right) \left( -\ln\left(\boldsymbol{\nu}_{0,l}\right) + b \right)$$

$$\leq \sum_{l=1}^{d} \sqrt{\frac{\boldsymbol{\nu}_{0,l}}{1-\beta_2} + 2\sigma^2 T + \sigma^2}$$

$$+ d\frac{2}{ab\sqrt{b}} \left( La^2 + 4\frac{a\sqrt{b}\sigma}{(1-c)^2} + 2\frac{a^2 c}{1-c} + 4\frac{L^2 c a^3 d}{\sqrt{b}(1-c)^5\sigma} \right) \mathbb{E}\ln\left( \sum_{l=1}^{d} \sqrt{1-\beta_2} \sqrt{\frac{\boldsymbol{\nu}_{0,l}}{1-\beta_2} + \sum_{s=1}^{T} |g_{s,l}|^2 + \sigma^2} \right)$$

$$+ \frac{4\sqrt{b}}{a(1-c)} f(\boldsymbol{w}_1) + \sum_{l=1}^{d} \frac{2}{ab\sqrt{b}} \left( La^2 + 4\frac{a\sqrt{b}\sigma}{(1-c)^2} + 2\frac{a^2 c}{1-c} + 2\frac{L^2 c a^3 d}{\sqrt{b}(1-c)^5\sigma} \right) \left( -\ln\left(\boldsymbol{\nu}_{0,l}\right) + b \right)$$

$$\leq \sum_{l=1}^{d} \sqrt{\frac{\boldsymbol{\nu}_{0,l}}{1-\beta_2} + 3\sigma^2 T} + D_1 + D_2 \ln\left( \mathbb{E} \sum_{l=1}^{d} \sqrt{1-\beta_2} \sqrt{\frac{\boldsymbol{\nu}_{0,l}}{1-\beta_2} + \sum_{s=1}^{T} |g_{s,l}|^2 + \sigma^2} \right),$$

where the second inequality is due to Eq. (11), the second-to-last inequality is due to Eq. (12), and the last inequality is due to Jensen's inequality. Solving the above ineqaulity with respect to $\sqrt{1-\beta_2} \sum_{l=1}^{d} \mathbb{E}\sqrt{\frac{\boldsymbol{\nu}_{0,l}}{1-\beta_2} + \sum_{s=1}^{T} |g_{s,l}|^2 + \sigma^2}$ then gives

$$\sqrt{1-\beta_2} \sum_{l=1}^{d} \mathbb{E} \sqrt{\frac{\boldsymbol{\nu}_{0,l}}{1-\beta_2} + \sum_{s=1}^{T} |g_{s,l}|^2 + \sigma^2} \leq 2\sqrt{1-\beta_2} D_1 + 4\sqrt{1-\beta_2} D_2 \ln(1 + \sqrt{1-\beta_2} D_2)$$

$$+ \sum_{l=1}^{d} \sqrt{\boldsymbol{\nu}_{0,l} + 3b\sigma^2}.$$

Therefore, by Cauchy's inequality, we have

$$\mathbb{E}\left[ \sum_{t=1}^{T} \|\boldsymbol{G}_t\|_1 \right]^2 \leq \left( \sum_{t=1}^{T} \mathbb{E}\left[ \left\| \frac{1}{\sqrt[4]{\widetilde{\boldsymbol{\nu}}_t}} \odot \boldsymbol{G}_t \right\|^2 \right] \right) \left( \sum_{t=1}^{T} \sum_{l=1}^{d} \mathbb{E}\sqrt{\widetilde{\boldsymbol{\nu}}_{t,l}^1} \right).$$

Since

$$\sum_{t=1}^{T}\sum_{l=1}^{d}\sqrt{\widetilde{\nu}_{t,l}^{1}} \leq \sum_{t=1}^{T}\sum_{l=1}^{d}\left(\sqrt{\beta_{2}\nu_{t-1,l}+(1-\beta_{2})\sigma^{2}}+\sqrt{(1-\beta_{2})}|G_{t,l}|\right)$$

$$\leq T\sum_{l=1}^{d}\sqrt{1-\beta_{2}}\sqrt{\frac{\nu_{0,l}}{1-\beta_{2}}+\sum_{s=1}^{T}|g_{s,l}|^{2}+\sigma^{2}}+\sum_{t=1}^{T}\sum_{l=1}^{d}\sqrt{(1-\beta_{2})}|G_{t,l}|$$

$$\leq T\left(2\sqrt{1-\beta_{2}}D_{1}+4\sqrt{1-\beta_{2}}D_{2}\ln(1+\sqrt{1-\beta_{2}}D_{2})+\sum_{l=1}^{d}\sqrt{\nu_{0,l}+3b\sigma^{2}}\right)+\sum_{t=1}^{T}\sqrt{(1-\beta_{2})}\|G_{t}\|_{1},$$

we have

$$\mathbb{E}\left[\sum_{t=1}^{T}\|G_{t}\|_{1}\right]^{2}$$

$$\leq\left(T\left(2\sqrt{1-\beta_{2}}D_{1}+4\sqrt{1-\beta_{2}}D_{2}\ln(1+\sqrt{1-\beta_{2}}D_{2})+\sum_{l=1}^{d}\sqrt{\nu_{0,l}+3b\sigma^{2}}\right)+\sum_{t=1}^{T}\sqrt{(1-\beta_{2})}\mathbb{E}\|G_{t}\|_{1}\right)$$

$$\times\frac{2\sqrt{T}}{\sqrt{b}}\left(D_{1}+\frac{D_{2}}{d}\sum_{l=1}^{d}\mathbb{E}\ln\nu_{T,l}\right).$$

Solving the above inequality with respect to $\sum_{t=1}^{T}\mathbb{E}\|G_{t}\|_{1}$ completes the proof. $\qquad\square$