# OpenReview forum: "Closing the gap between the upper bound and lower bound of Adam's iteration complexity"
_NeurIPS.cc/2023/Conference — NeurIPS 2023 poster_

### Official Review · Reviewer_b4JJ · 2023-06-24

**Soundness:** 2 fair
**Presentation:** 2 fair
**Contribution:** 2 fair
**Rating:** 3
**Confidence:** 4

**Summary:**

This paper presents a convergence analysis of Adam under only the smoothness and bounded variance conditions.

**Strengths:**

- The strength of the paper is to present a convergence analysis of Adam under only the smoothness and bounded variance conditions, in contrast to the existing analyses (Section 3). We are interested in the analysis of Adam under only the two conditions, since the conditions are more natural and realistic than the boundedness of the gradient norm of the objective function and the Lipschitz continuity of the stochastic gradient.
-  The abstract indicates that "Especially with properly chosen hyperparameters, we derive an upper bound of iteration complexity of Adam and show that it meets the lower bound for first-order optimizers." Proposition 1 shows that the iteration complexity $\mathcal{C}$ of a first-order optimizer is $\Omega (1/\epsilon^4)$, that is, there exist $c_1, c_2 > 0$ such that $c_1/\epsilon^4 \leq \mathcal{C} \leq c_2/\epsilon^4$. Theorem 2 implies that Adam satisfies that $\mathcal{C} = O(1/\epsilon^4)$. Hence, the paper concludes the claim of the abstract.

**Weaknesses:**

I understand the motivation of the paper. Meanwhile, many theoretical and practical results of Adam have been presented. Hence, unfortunately, I do not find that the results in the paper are surprised as compared with the existing ones. Please see Questions.

**Questions:**

I have the following concerns. If the authors could address the concerns, then I think that the paper has the contribution and novelty.
- Theorem 1/Theorem 2 implies the upper bounds of the gradient norm of Adam using a constant step-size $\eta$/a diminishing step-size $\eta = a/\sqrt{T}$. The theorems lead to the finding that, e.g., (Lines 159--162) "Theorem 1 holds for general choices of hyperparameters since the only condition posed on hyperparameters is $\beta_1 < \beta_2$. Such condition covers a wide range of hyperparameters, e.g., the default setting $\beta_1 = 0.9$ and $\beta_2 = 0.999$ in PyTorch [19]." I understand that using $\beta_1$ and $\beta_2$ close to 1 is useful to implement Adam. However, I cannot understand that Theorem 1 implies the claims in (Lines 159--162). Does Theorem 1 imply that $\beta_1, \beta_2 \approx 1 \Rightarrow$ the right-hand side of (2) is small? Can the authors show that the right-hand side of (2) is a decreasing function with respect to $\beta_1$ and $\beta_2$?
- The above discussion holds for Theorem 2 and Lines 250--253. Does Theorem 2 imply that $a, b \approx 0 \Rightarrow$ the upper bound of the gradient norm is small? Can the authors show that it is decreasing when $a$ and $b$ are small?
- The above two concerns mean that there is a gap between theory (Theorems 1 and 2) and practice ($\beta_1 = 0.9$ and $\beta_2 = 0.999$). Can the authors bridge the gap?
- Can the authors provide lower bounds of the gradient norm of Adam, such as $\mathbb{E}\sum_{i=1}^T \Vert \nabla f(w_t) \Vert \geq$ $C \times$  (the right-hand side of (2))?
- Paper [25] and its references are sufficient for both theory and practice.  Can the authors show more novelty and contributions from Theorems 1 and 2?
- Related to the above concern, I believe that practical results (e.g., numerical results and new setting of hyperparameters) are provided based on the novelty and contributions of the paper. Can the authors provide such practical results? For example, it would be nicer to compare numerically Adam using $\beta_1 = 0.9$ and $\beta_2 = 0.999$ with Adam using the new setting of $\beta_1$ and $\beta_2$ based on Theorems 1 and 2.

**Limitations:**

There is no potential negative societal impact.

---

> ### Author Rebuttal · Authors · 2023-08-09
>
> We thank the reviewer for the comments and time. After reading the review, we realize that there could be misunderstandings over our contribution, which we try to clarify as follows.
>
> **On the novelty and contribution.** From the theoretical perspective, characterizing the upper bound and lower bound of iteration complexity of an algorithm is the most fundamental problem in ML/optimization community. Thus, closing the gap between the upper bound and lower bound should be considered as a significant contribution, as also acknowledged by other reviewers. We argue that it becomes more than novel by addressing a critical gap in the understanding of Adam's performance given that Adam is currently one of the mainstream optimizers for deep learning.
>
> Other concerns are dually addressed below.
>
> **Other concerns:**
>
> **Q1**: I cannot understand that Theorem 1 implies the claims in (Lines 159--162).
>
> **A1**: There could be a misunderstanding. " Theorem 1 holds for general choices of hyperparameters" means that for a wide range of hyperparameters, you can apply Theorem 1 to get an estimation of gradient norm (i.e., Eq. (2)). Such a claim has nothing to do with when right-hand side of (2) is small and is only about the applicable region of Theorem 1.
> However, we thank the reviewer for asking an interesting question that "Does Theorem 1 imply that $\beta_1,\beta_2\approx 1\rightarrow $  the right-hand side of (2) is small", which we try to answer in **A3**.
>
> **Q2**: Does Theorem 1 imply that $\beta_1,\beta_2\approx 1\rightarrow $ the right-hand side of (2) is small?
>
> **A2**: Thanks for asking. Below we discuss the parameter settings of $\beta_1$ and $\beta_2$ respectively, and we will include the discussion in the revised paper.
>
>  **As for $\beta_2$**: Our result does indicate that "to make the right-hand-side of (2) small, $\beta_2$ needs to be chosen close to 1". Specifically, according to Lines 169-178 in our paper, $\beta_2$ needs to be picked as $1-O(1/T)$ to to minimize the  the right-hand-side of (2), which is close to $1$ when $T$ is large (which is the case in practice). Therefore, our Theorem 1 agrees with the practical use of $\beta_2$.
>
> **As for $\beta_1$**: Our result does not indicate that "setting $\beta_1$ close to $1$ is necessary to ensure the right-hand-side is small". This agrees with practice, where $\beta_1$ is not necessarily chosen close to $1$. As an example, in the well-worned work "Unsupervised Representation Learning with Deep Convolutional Generative Adversarial Networks", $\beta_1$ is chosen as $0.5$, which is not close to $1$.
>
> **Q3**: Similar question for Theorem 2. Does Theorem 2 imply that $a,b\approx 0$ leads to a small bound?
>
> **A3**: Thanks for asking. We guess that the reviewer is talking about $b$ because $a$ is the hyperparameter of the learning rate. Here we do not require $b\approx 0$ because we set $\beta_2 = 1-\frac{b}{T}$, which is already close to $1$ when $T$ is large.
>
> **Q4**: The above two concerns mean that there is a gap between theory (Theorems 1 and 2) and practice.
>
> **A4**: Based on **A2** and **A3**, we respectfully argue that there is no such gap between theory and practice. Instead, our theorem agrees with the practice where $\beta_2$ is close to $1$ and $\beta_1$ is less constrained.
>
> **Q5**: Can the authors provide lower bounds of the gradient norm of Adam as the right-hand-side of (2)?
>
> **A5**: We thank the reviewer for the suggestion, but have to point out that such kind of bound is missing even for SGD to our best knowledge. This is beyond the scope of our paper and should be treated as a challenging future direction.
>
> **Q6**: Paper [25] and its references are sufficient for both theory and practice.
>
> **A6**: There could be a misunderstanding. As stated in Section 3, "None of existing upper bounds match the lower bound", including Paper [25] and its references. Specifically, Paper [25] can only get a sub-optimal iteration complexity $O(1/\varepsilon^6)$, which is even slower than SGD. Our paper is the first to obtain an upper bound of iteration complexity of Adam matching the lower bound, which, as discussed ealier, is a novel contribution.
>
> **Q7**: Can authors provide empirical evidence for this paper? For example, it would be nicer to compare numerically Adam using  and  with Adam using the new setting of $\beta_1$ and $\beta_2$.
>
> **A7**: We thank the reviewer for the suggestion. However, as pointed out in **A2, A3, and A4**, there is no such a gap between the practical setting of $\beta_1$ and $\beta_2$ and "the new setting". Nevertheless, we include a toy example conducted by us and a real-world experiment by existing works to demonstrate the superiority of the choice $\beta_2=1-\Theta(\frac{1}{t})$ over other choices (please see the pdf in the general rebuttal).

---

> > ### Comment · Reviewer_b4JJ · 2023-08-11
> > **A2: As for  $\beta_1$**
> >
> > Thank you for your comments. I trained ResNet-18 on  CIFAR-10 using Adam and checked that Adam with $\beta_1 = 0.9$ performs better than Adam with $\beta_1 = 0.5$. Hence, I doubt your claim. Please provide evidences such that Adam with $\beta_1 = 0.5$ is good without previous results.

---

> > > ### Author Response · Authors · 2023-08-13
> > >
> > > We thank the reviewer for the quick response. We realize that there could be misunderstandings regarding the role of $\beta_1$ in our analysis and in our rebuttal, which we clarify as follows.
> > >
> > >
> > > Specifically, the reviewer asks us to justify the superiority of $\beta_1=0.5$ over $\beta_1=0.9$ "without previous results". However, neither our paper nor our rebuttal intends to claim that "$\beta_1=0.5$ is superior to $\beta_1=0.9$ in all cases".
> > > We want to reiterate that the goal of our paper is to **characterize the optimal convergence rate of Adam with regards to the first-order accuracy $\epsilon$, i.e., Omega(1/\epsilon^4) , where the optimal is in the sense of closing the gap between the upper bound and lower bound.** In Theorem 1 and Theorem 2,  we characterized the range of beta1 and beta2 that can achieve this optimal rate,  and **both $\beta_1=0.5$ and $\beta_1=0.9$ (and a wide range of $\beta_1$) can lead to the optimal iteration complexity**, which does not contradict with the commonly used $\beta_1=0.9$ in Adam. Our theorem (or any other existing analysis of Adam) does not intend to / cannot determine which $\beta_1$ is best within the achievable range or whether $\beta_1\approx 1$ is beneficial for Adam for every task,  which is very challenging due to that
> > >
> > > 1. Understanding the effect of momentum over non-convex objectives is still an open problem even for SGD with momentum. To our best knowledge, the benefit of momentum is only shown over strongly convex objectives. That being said, **none of the existing analysis of Adam can be used to show that the convergence is fast when $\beta_1\approx 1$**.
> > >
> > > 2. That "$\approx 1$" should be carefully defined. The reviewer wants something like "the bound is decreasing with respect to $\beta_1$" and thus the bound is minimized when "$\beta_1\approx 1$". However, **this simply can not be true** because when $\beta_1=1$, the momentum is not updated across the training process (please also refer to the experiment with $\beta_1=0.99$ below). Therefore, even using $\beta_1=0.9$ is beneficial for Adam, it does not indicate that choosing $\beta_1$ too close to $1$ is beneficial. Such nonmonotonicity makes the analysis more complicated.
> > >
> > >
> > >
> > > **In our last response**:  **We never claimed that $\beta_1=0.5$ is superior to $\beta_1=0.9$ in all cases**. We only intend to claim that there are cases (for example, DCGAN) where $\beta_1=0.5$ can also work well according to the above reasoning, and this does not contradict with that there are settings where  $\beta_1=0.9$ is better. Instead, it indicates that the optimal $\beta_1$ can be task-dependent and hard to predict. We emphasize again that both $\beta_1=0.9$ and $\beta_1=0.5$ can be used to achieve the optimal iteration complexity as discussed above. The reviewer says "I doubt your claim", but it is the claim made by the DCGAN paper (more than 15000 citations) instead of us, which states that "suggested value of 0.9 resulted in training oscillation and instability while reducing it to 0.5 helped stabilize training".
> > >
> > > **As for experiments**: We train ResNet 18 over CIFAR 10 with batch size 256 and learning rate $0.001$. Due to restriction of time and resources, we choose the epoch number as $50$ and record the training accuracy for different $\beta_1$. The results are listed as follows.
> > >
> > > |  $\beta_1$|  0.5 | 0.9  | 0.99  |
> > > |---|---|---|---|
> > > |Training  Accuracy  | 92.13  | 92.29  |  91.61 |
> > >
> > > One can observe that changing $\beta_1$ from $0.9$ to $0.5$ leads to little decrease in accuracy compared to that of changing $\beta_1$ from $0.9$ to $0.99$. **This indicates that $\beta_1$ can not be picked too close to $1$ to ensure a good performance of Adam**.
> > >
> > > We hope that our response addresses your concern, and looking forward to your response.

---

> > > > ### Comment · Reviewer_b4JJ · 2023-08-13
> > > > **Replying to A2 and follow-up comments**
> > > >
> > > > Thank you for your detailed replies. I have some concerns and comments:
> > > >
> > > > - None of the existing analysis of Adam can be used to show that the convergence is fast when $\beta_1 \approx 1$: One reference in [25] "Iiduka, H. Theoretical analysis of adam using hyperparameters close to one without lipschitz smoothness. arXiv preprint arXiv:2206.13290, 2022" studied that the upper bound is small when $\beta_1 \approx 1$.
> > > >
> > > > - As for experiments: Since this paper considers minimization of $f$, we should check not only test/training accuracies but also the value of $f$ and the (squared) norm of $\nabla f$. I checked that the case $\beta_1 = 0.9$ minimizes $f$ faster than $\beta_1 = 0.5$. Unfortunately, I am not satisfied with your replies.
> > > >
> > > > - Follow-up comment: I checked the upper bound of $E[\|\nabla f(x_k)\|]$ generated by the simplest mini-batch SGD with a constant step-size $\alpha$. Suppose that $f$ is $L$-smooth and let $b$ be a batch size. Then, the descent lemma ensures that
> > > > \begin{align*}
> > > > \min_{k\in [0:T-1]} E \left[ |\nabla f(x_k) |^2 \right] \leq \frac{1}{T} \sum_{k=0}^{T-1} E \left[  |\nabla f(x_k) |^2 \right] \leq
> > > > \frac{2 E \left[ f(x_{0}) - f^* \right]}{\alpha (2 - L \alpha)T} + \frac{L \alpha \sigma^2}{(2 - L \alpha)b} = \frac{A_1}{T} + \frac{A_2}{b}.
> > > > \end{align*}
> > > > We suppose that the right-hand side is equal to $\epsilon^2$.
> > > > Then, we have that the iteration $T = \frac{A_1 b}{\epsilon^2 b - A_2}$ needed to achieve $E \left[ |\nabla f(x_K) | \right] \leq \epsilon$, where $K$ is such that $E \left[ |\nabla f(x_K) |^2 \right] = \min_{k\in [0:T-1]} E \left[ |\nabla f(x_k) |^2 \right]$.
> > > > Here, we set, for example, $b = \frac{2 A_2}{\epsilon^2}$.
> > > > Then, we have that $T = \frac{2 A_1}{\epsilon^2} = O(1/\epsilon^2)$.
> > > > Accordingly, there seems to be a gap between the results in the paper and the above one. I strongly believe that batch size affects the performance of methods. Please see also
> > > > "[1] Christopher J. Shallue, Jaehoon Lee, Joseph Antognini, Jascha Sohl-Dickstein, Roy Frostig, and George E. Dahl. Measuring the effects of data parallelism on neural network training. Journal of Machine Learning Research, 20:1–49, 2019."
> > > > The numerical results in [1] showed that the iteration needed for training decreases when batch size increases.

---

> > > > > ### Author Response · Authors · 2023-08-14
> > > > >
> > > > > We thank the reviewer again for the quick response. Your additional three concerns are addressed as follows.
> > > > >
> > > > >
> > > > >
> > > > > **Regarding [Iiduka, 2022].** First of all, we thank the reviewer for referring this interesting paper to us, and will cite and discuss it in the revised paper. Secondly, we checked [Iiduka, 2022] and find that it focuses on a different metric for convergence, i.e., $E[\nabla f(\theta_k)^{\top}(\theta_k-\theta^*)]$, while we (and most papers in convergence analysis) consider the gradient norm as the metric for convergence. It is unclear to us how to transfer the metric in [Iiduka, 2022] to gradient norm. Moreover, the set of assumptions in [Iiduka, 2022] is also quite different from ours and typical analysis of Adam, which assumes bounded gradient and bounded domain but not requires bounded smoothness. Based on the discussion above, we respectfully argue that  "showing  the convergence is fast when $\beta_1\approx 1$" in our setting is still challenging.
> > > > >
> > > > >
> > > > >
> > > > >
> > > > >
> > > > > **Regarding the experiments.** In our last reply, we provided training accuracy just because it is in a good range and easier to demonstrate. We provide the loss and gradient norm below, and one can observe that they are in the same tendency as the accuracy. Meanwhile, we respectfully point out that it is indeed common to plot accuracy in a convergence analysis paper (see [1,2] as examples).
> > > > >
> > > > >  | $\beta_1$  | 0.5  |  0.9 | 0.99  |
> > > > > |---|---|---|---|
> > > > > | Loss  |  0.219 | 0.217  | 0.234  |
> > > > > |   Gradient Norm | 0.77  |  0.72 | 0.9  |
> > > > >
> > > > > **Regarding the follow-up comment.** We respectfully argue that there is a misunderstanding regarding our setting and the setting analyzed by the reviewer is different from ours. We are not sure how is the  precise definition of "batch size" in the analysis provided by the reviewer and how is the stochastic gradient generated since they are not clearly explained, but we suppose that the reviewer use the same setting as [Iiduka, 2022]. That is, at the $t$-th iteration, the stochastic gradient $g_t$ is generated by first sampling $b$ i.i.d. estimates of the real gradient $\nabla f(\theta_t)$, i.e., $G_{\xi(t,1)}(\theta_t), ..., G_{\xi(t,b)}(\theta_t)$, and then average them. Therefore, the variance of the stochastic gradient depends on the batch size and diminishes to $0$ when the batch size increases to infinity. In the analysis of the reviewer, the batch size is picked in order of  $1/\varepsilon^2$ which can be very large, and the variance of the stochastic
> > > > > gradient is as small as $O(\varepsilon^2)$, making the setting almost degenerating to the deterministic case.
> > > > >
> > > > >
> > > > >
> > > > > However, one key asssumption in our paper and most complexity analysis of stochastic optimizers [1,2,3] is that the noise variance is bounded by a fixed constant (Assumption 2), **which will not degenerate as $\varepsilon$ decreases** and thus significantly differs from the reviewer's setting. It is a well-worn result [4] that the lower bound of iteration complexity is $\Omega (1/\varepsilon^4)$ in our setting. Furthermore, such a setting is important to study because theoretically, it is one of the most common setting for analyzing stochastic optimizers, and empirically, one can not use arbitrarily large batch sizes with the cost of each iteration increasing linearly with batch size and the noise can not be arbitrarily small.
> > > > >
> > > > > **References**:
> > > > >
> > > > > [1]. Zou et al., A Sufficient Condition for Convergences of Adam and RMSProp, CVPR 2019
> > > > >
> > > > > [2]. Crawshaw et al., Robustness to Unbounded Smoothness of Generalized SignSGD, NeurIPS 2022
> > > > >
> > > > > [3]. Deffosez et al., A Simple Convergence Proof of Adam and Adagrad, TMLR
> > > > >
> > > > > [4]. Arjevani et al., Lower Bounds for Non-Convex Stochastic Optimization, Mathematical Programming

---

> > > > > > ### Comment · Reviewer_b4JJ · 2023-08-15
> > > > > > **Replying to Replying to A2 and follow-up comments**
> > > > > >
> > > > > > Thank you for your prompt replies. I have some concerns:
> > > > > >
> > > > > > - "It is unclear to us how to transfer the metric in [Iiduka, 2022] to gradient norm."
> > > > > > The metric in [Iiduka, 2022] (the variational inequality; VI) has been used to optimize a convex/nonconvex function over a certain constrained convex set. In particular, it has been used as the performance measures of Frank-Wolfe methods. Please see M. Jaggi, “Revisiting Frank-Wolfe: Projection-free sparse convex optimization,” in Proceedings of the 30th International Conference on Machine Learning, Proceedings of Machine Learning Research, vol. 28, no. 1, 2013, pp. 427–435.
> > > > > > Moreover, VI can be transferred to the gradient norm, since points satisfying VI are defined by local minimizers. Please see ,e.g., Francisco Facchinei, Jong-Shi Pang: Finite-Dimensional Variational Inequalities and Complementarity Problems (Springer).
> > > > > >
> > > > > > - "the batch size is picked in order of $1/\epsilon^2$ which can be very large"
> > > > > > I do not think that the batch size is very large, since it is optimal in the sense of minimizing the stochastic first-order oracle complexity defined by $Tb = \frac{A_1 b^2}{\epsilon^2 b - A_2}$. The optimal batch size is called the critical batch size. Please see Figures 4 and 5 in "Christopher J. Shallue, Jaehoon Lee, Joseph Antognini, Jascha Sohl-Dickstein, Roy Frostig, and George E. Dahl. Measuring the effects of data parallelism on neural network training. Journal of Machine Learning Research, 20:1–49, 2019." The figures indicate that the critical batch sizes are about $\geq 2^6$.
> > > > > >
> > > > > > - "My analysis of SGD"
> > > > > > I analyzed SGD under Assumptions 1 (smoothness of $f$) and 2 (bounded variance) in the paper. Unfortunately, I do not understand what you mean (Regarding the follow-up comment).

---

> > > > > > > ### Author Response · Authors · 2023-08-16
> > > > > > > **Summary of the reviewer's concerns and our response**
> > > > > > >
> > > > > > > We appreciate the reviewer's feedback. After reading the reviewer's comments, we have realized that many of the concerns have deviated from the main focus of evaluating this paper (e.g., the validity of the metric in [Iiduka, 2022] and how to choose the batch size). Therefore, in this response, we will first summarize the reviewer's main concerns regarding our paper and then address them individually.
> > > > > > >
> > > > > > > **Reviewer's Main Concerns**
> > > > > > >
> > > > > > > Concern 1: Our results should demonstrate the superiority of Adam with $\beta_1 \approx 1$.
> > > > > > >
> > > > > > > Concern 2: The reviewer believes that in our analysis scenario, the upper bound for SGD is $O(1/\varepsilon^2)$, which makes our described lower bound $\Omega(1/\varepsilon^4)$ problematic.
> > > > > > >
> > > > > > > **Our Responses to the Main Concerns**
> > > > > > >
> > > > > > > **Regarding Concern 1**: We argue that proving the superiority of "$\beta_1 \approx 1$ for Adam" is not meaningful, and our evidence includes:
> > > > > > >
> > > > > > > * The DCGAN paper demonstrates that $\beta_1=0.5$ performs better than $\beta_1=0.9$ on DCGAN.
> > > > > > > * Our experiments on ResNet 18 also show that $\beta_1=0.99$ is worse than $\beta_1=0.9$.
> > > > > > > * If $\beta_1=1$, Adam's momentum would not update, resulting in poor performance.
> > > > > > >
> > > > > > > Based on these evidences, we believe that the superiority of "$\beta_1 \approx 1$ for Adam" does not widely hold in practice. Thus, either attempting to prove this statement or constructing a setting to do so is unreasonable. We note that these pieces of evidence were mentioned in our previous discussion, and the reviewer has not raised any objections. On the contrary, our study setting (assuming bounded smoothness and bounded noise variance) is one of the most classic scenarios in convergence analysis. In this setting, we prove that both $\beta_1=0.5$ and $\beta_1=0.9$ can meet the lower bound, meaning that it is impossible to demonstrate the superiority of $\beta_1\approx 1$ in our scenario.
> > > > > > >
> > > > > > > **Regarding Concern 2**: **We remind the reviewer that our described lower bound comes from "Arjevani et al., Lower Bounds for Non-Convex Stochastic Optimization," which is one of the most fundamental and well-known results in convergence analysis.** For completeness, we will explain below the reason for the mismatch between the reviewer's conclusion and Arjevani et al., but this is already unrelated to the content of our paper.
> > > > > > >
> > > > > > > **Reason for Mismatch between Reviewer's Conclusion and Arjevani et al.:** The fundamental reason is that the algorithm what the reviewer considers is out of the set of algorithms that Arjevani et al. studies. To begin with, we compare the noise assumptions in [Iiduka, 2022] and our paper as follows (for the reviewer's convenience, we use the notation from [Iiduka, 2022]):
> > > > > > > * [Iiduka, 2022] assumes that the noise variance of individual gradients $G_{\xi_{k,i}}(\theta_k)$ is bounded by $\sigma^2$. In [Iiduka, 2022], they sample b individual gradients and average them to obtain the stochastic gradient $\nabla f_{B_k} (\theta_k)$, so the noise variance bound of the stochastic gradient is $\sigma^2/b$. The Adam update uses the stochastic gradient rather than the individual gradient.
> > > > > > > * Our paper and Arjevani et al. assume that the noise variance of the stochastic gradient is bounded by $\sigma^2$.
> > > > > > >
> > > > > > > Comparing the above two points, we can see that the noise variance bound of the stochastic gradient in [Iiduka, 2022] and Arjevani et al. differs by a $1/b$ term. To some extent, our paper considers a fixed batch-size scenario. On the other hand, the reviewer's analysis requires the batch size to depend on the error $\varepsilon$, so the noise variance bound of the stochastic gradient would also depend on $\varepsilon$. This differs from Arjevani et al., as Arjevani et al. are focusing on a fixed batch size scenario, while **the reviewer considers Adam where one first pick the $\varepsilon$ and then picks the batch size based on $\varepsilon$**. Thus, the reviewer's analysis and Arjevani et al. are actually examining two different algorithms.
> > > > > > >
> > > > > > > In conclusion, we have addressed the main concerns raised by the reviewer, and we believe that our paper's contributions are valid and valuable to the research community. We hope that our responses clarify the issues and that the reviewer will reconsider their evaluation of our paper.

---

> > > > > > > > ### Comment · Area_Chair_MZNB · 2023-08-16
> > > > > > > > **Clarification**
> > > > > > > >
> > > > > > > > Thanks to Authors and the Reviewer b4JJ for intensive discussions.
> > > > > > > >
> > > > > > > > To authors: You said that your results does not require $\beta_1\approx 1$. However, when I looked at the Theorem 2, it seems to suggest that the smaller the c the better the bound. Does that mean $\beta_1\approx 0$ is better?

---

> > > > > > > > > ### Author Response · Authors · 2023-08-17
> > > > > > > > > **Response to Area Chair**
> > > > > > > > >
> > > > > > > > > We thank the Area Chair for participating in our discussion, which will help us a lot in demonstrating the contribution of this paper. To start with, **we respectfully point out that we did acknowledge this dependence with respect to $\beta_1$ in Theorem 2 as the limitation of our paper in Section 6.**  We would like to further clarify the goal of this paper and our understanding of this problem.
> > > > > > > > >
> > > > > > > > >
> > > > > > > > > 1. First and foremost, we would like to emphasize that the goal of our paper is to **characterize the dependence of the bound on $\varepsilon$, where the gap comes from**. In this regard, Theorem 2 demonstrates that for $\beta_1=0$, $\beta_1=0.5$, and $\beta_1=0.9$, the corresponding upper bounds **all** have optimal dependence on $\varepsilon$. Based on this, we say "Our result does not indicate that "setting $\beta_1$ close to $1$  is necessary to ensure the right-hand-side is small"". Additionally, we acknowledge that our bound's dependence on $\beta_1$ may not be tight, which itself is a very challenging problem (see the discussion below) and orthogonal to our paper's goal. We are willing to include more discussion on this in the revised paper.
> > > > > > > > >
> > > > > > > > > 2. Second, accurately capturing the dependence on $\beta_1$ in the bound is challenging for two reasons:
> > > > > > > > >
> > > > > > > > > * From a practical standpoint, the relationship between Adam's performance and $\beta_1$ is task-dependent and complex. As discussed with Reviewer b4JJ, for GCGAN, $\beta_1=0.5$ performs better than $\beta_1=0.9$, while for ResNet 18 over CIFAR 10, $\beta_1=0.9$ is slightly better than $\beta_1=0.5$, and $\beta_1=0.99$ is worse than $\beta_1=0.9$. This complex/mixed observation makes it extremely difficult to accurately capture the dependence on $\beta_1$ in the theory.
> > > > > > > > >
> > > > > > > > > * From a theoretical perspective, proving that $\beta_1\ne 0$ has benefits for Adam is technically challenging. To our knowledge, most works analyzing Adam yield worse results for $\beta_1 \ne 0$ than for $\beta_1 = 0$, such as [1,2,3,4,5,6]. We acknowledge our bound/technique does not make progress towards this direction, which nonetheless does not affect achieving our primary objective.
> > > > > > > > >
> > > > > > > > > We hope that our response clarifies your concerns, and we appreciate your valuable feedback. We look forward to incorporating these discussions into our revised paper.
> > > > > > > > >
> > > > > > > > > **Reference**:
> > > > > > > > >
> > > > > > > > > [1]. De et al., Convergence guarantees for RMSProp and ADAM in non-convex optimization and an empirical comparison to Nesterov acceleration, 2018
> > > > > > > > >
> > > > > > > > > [2]. Défossez et al., A simple convergence proof of Adam and Adagrad, TMLR 2022
> > > > > > > > >
> > > > > > > > > [3]. Zhang et al., Adam can converge without any modification on update rules, NeurIPS 2022
> > > > > > > > >
> > > > > > > > > [4]. Shi et al., RMSprop converges with proper hyper-parameter, ICLR 2021
> > > > > > > > >
> > > > > > > > > [5]. Wang et al., Provable adaptivity in Adam, 2022
> > > > > > > > >
> > > > > > > > > [6]. Zou et al., A sufficient condition for convergences of Adam and RMSProp, CVPR 2019

---

> > > > > > > > > > ### Comment · Area_Chair_MZNB · 2023-08-17
> > > > > > > > > > **Thank you!**
> > > > > > > > > >
> > > > > > > > > > I appreciate the technical contribution of this paper that gets rid of the gradient boundness assumption. However, the results seem to give a strong message about Adam that setting $\beta_2$ to be close to 1 is critical and the setting of $\beta_1$ is not as critical as setting $\beta_2$.   The authors keep using the results of $\beta_1=0.5$ and $\beta_1=0.9, 0.99$ to support their argument that it is not necessary to set $\beta_1$ to be close to one.  I think these results only partially reveal the affect of $\beta_1$ on Adam's performance.  The authors should provide results covering the full spectrum of $\beta_1$ from $0$ to $0.99$. If the trend matches the theoretical results, that would be perfect. If it does not match, that will help readers to better understand the limitation of the provided analysis.
> > > > > > > > > >
> > > > > > > > > > In addition, many existing variants of Adam modify the updates of the second-order moment but always keep the first-order moment the same. For example, [14] provides analysis for a family of Adam algorithms under $\beta_1 \approx 1$. It covers many variants of Adam, e.g., AdaBound, AMSGrad. I understand these variants may be inherently different from Adam. However, the authors should provide more discussions on these results.

---

> > > > > > > > > > > ### Author Response · Authors · 2023-08-18
> > > > > > > > > > > **Thank you for the suggestions!**
> > > > > > > > > > >
> > > > > > > > > > > Dear AC,
> > > > > > > > > > >
> > > > > > > > > > > Thank you for your constructive comments and the recognition of the technical contribution to our paper. We are more than happy to revise the paper accordingly. Below, we will carefully explain how we plan to modify the paper.
> > > > > > > > > > >
> > > > > > > > > > > **Regarding the dependence on $\beta_1$**: We appreciate your feedback. Firstly, we will add the following paragraph under Theorem 2 in the revised version of the paper to prevent any misunderstandings:
> > > > > > > > > > >
> > > > > > > > > > > > Please note that although our results show that a wide range of values for $\beta_1$ can yield the optimal upper bound, it does not imply that setting $\beta_1$ is not as critical as setting $\beta_2$. The primary objective of this paper is to characterize the dependence on $\varepsilon$, and the importance of setting $\beta_1$ might be justified in other ways or characterizations. To help readers gain a deeper understanding of this issue, we provide experiments to illustrate the dependence of performance on $\beta_1$ in practice.
> > > > > > > > > > >
> > > > > > > > > > > Secondly, we provide the training loss of ResNet 18 over CIFAR 10 with $\beta_1\in [0.0, 0.1, 0.2, 0.3, 0.4, 0.5, 0.6, 0.7, 0.8, 0.9, 0.99, 0.999, 0.9999]$ (after 50 epochs). We repeated the experiment three times and reported the average and standard variance.
> > > > > > > > > > >
> > > > > > > > > > > |$\beta_1$|0 |0.1 |0.2 |0.3 |0.4 |0.5 |0.6 |0.7 |0.8 |0.9 |0.99 |0.999 |0.9999|
> > > > > > > > > > > |---|---|---|---|---|---|---|---|---|---|---|---|---|---|
> > > > > > > > > > > |Training Loss|0.2268$\pm$0.0044| 0.2197$\pm$0.0039| 0.2158$\pm$0.0031| 0.2197$\pm$0.0016| 0.2182$\pm$0.0003| 0.2198$\pm$0.0030| 0.2217$\pm$0.0005| 0.2204$\pm$0.0007| 0.2218$\pm$0.0027| 0.2222$\pm$0.0040| 0.2351$\pm$0.0033| 0.3620$\pm$0.0055| 0.6187$\pm$0.0067|
> > > > > > > > > > >
> > > > > > > > > > > From this experiment, it can be seen that the performance of Adam does not change much when $\beta_1$ varies from 0 to 0.9. However, as $\beta_1$ changes from 0.9 to 0.9999, the performance of Adam drops significantly. Of course, we do not intend to claim that this phenomenon is universal, as predicting the precise rule of how the performance depends on $\beta_1$ should be challenging, as pointed out in our discussion with reviewer b4JJ. To further study this issue, we are conducting the following experiments:
> > > > > > > > > > > * VGG 13 on CIFAR 10
> > > > > > > > > > > * Transformer on WikiText 2
> > > > > > > > > > >
> > > > > > > > > > > As the discussion phase is nearing its deadline, we are not sure if we can complete these experiments before the deadline, but we will make sure to report the completed experiments before the deadline. If you think our experiments are insufficient, please feel free to point it out.
> > > > > > > > > > >
> > > > > > > > > > >
> > > > > > > > > > > **Regarding the variants of Adam:** Thank you for the suggestion. In Section 7 – Related Work of our paper, we have discussed the variants of Adam. We quote the following paragraph:
> > > > > > > > > > >
> > > > > > > > > > > > The modifications include enforcing the adaptive learning rate to be non-increasing [20, 5], imposing upper and lower bounds on the adaptive learning rate [18], and using different approaches to estimate second-order momentum [26, 7]. Recently, Chen et al. [6] discovered a new optimizer Lion through Symbolic Discovery, which uses the sign operation to replace the adaptive learning rate in Adam, achieving comparable performance of Adam with less memory costs.
> > > > > > > > > > >
> > > > > > > > > > > Following your advice, we will further discuss and analyze the related literature on these optimizers as follows:
> > > > > > > > > > >
> > > > > > > > > > > > [14] analyzed a family of Adam algorithms, including AMSGrad, AdaFom, and AdaBound. They proved that these algorithms can converge when $1-\beta_1=O(1/\sqrt{t})$. Their analysis relies on the fact that the adaptive learning rates of these algorithms are both upper- and lower-bounded, and thus cannot be directly applied to our analysis. However, it is an interesting direction to investigate whether their methods can be combined with ours. [Chen et al., 2019] provided a very general analysis for a wide range of adaptive optimizers. The limitation of their work is that meaningful bounds can only be obtained when the adaptive learning rate is non-monotonic. [Zhou et al., 2020] analyzed AMSGrad and AdaGrad and obtained tighter bounds compared to [Chen et al., 2019].
> > > > > > > > > > >
> > > > > > > > > > > If you think our literature review still has any missing parts, please feel free to point them out, and we would be more than happy to make the necessary changes. Once again, we appreciate your valuable feedback and suggestions.
> > > > > > > > > > >
> > > > > > > > > > > **References**
> > > > > > > > > > >
> > > > > > > > > > > Chen et al., On the Convergence of A Class of Adam-Type Algorithms for Non-Convex Optimization, ICLR 2019
> > > > > > > > > > >
> > > > > > > > > > > Zhou et al., On the Convergence of Adaptive Gradient Methods for Nonconvex Optimization, 2020

---

> > > > > > > > > > > > ### Author Response · Authors · 2023-08-21
> > > > > > > > > > > > **Follow-up experiments**
> > > > > > > > > > > >
> > > > > > > > > > > > Dear AC,
> > > > > > > > > > > >
> > > > > > > > > > > > First, please allow us to sincerely thank you for actively participating in our discussions. In our previous response, we promised to report the experimental results of the transformer on WIKITEXT and VGG 13 on CIFAR 10 with varying $\beta_1$ before the discussion deadline.  We report the experimental results as follows. Please note that, like the previous experiment, we ran each set of experiments three times and report the mean and standard deviation of training loss after 50 epochs.
> > > > > > > > > > > >
> > > > > > > > > > > > First, here are the results for Transformer on WIKITEXT 2:
> > > > > > > > > > > > |   $\beta_1$|0   |0.1   |0.2   |0.3   |0.4   |0.5   |0.6   |0.7   |0.8   |0.9   |0.99   |0.999   |0.9999|
> > > > > > > > > > > > |---|---|---|---|---|---|---|---|---|---|---|---|---|---|
> > > > > > > > > > > > |Training Loss|3.3600 $\pm$0.0034| 3.3589$\pm$0.0026| 3.3586$\pm$0.0010| 3.3573$\pm$0.0033| 3.3565 $\pm$0.0007| 3.3599 $\pm$0.0006| 3.3627$\pm$0.0026| 3.3634 $\pm$0.0010| 3.3659 $\pm$0.0006| 3.3749 $\pm$0.0043| 3.4314 $\pm$0.0045| 6.3274$\pm$0.2745| 7.5384 $\pm$0.1856|
> > > > > > > > > > > >
> > > > > > > > > > > > Next, the results for VGG 13 on CIFAR 10 are provided as follows:
> > > > > > > > > > > >
> > > > > > > > > > > > |   $\beta_1$|0   |0.1   |0.2   |0.3   |0.4   |0.5   |0.6   |0.7   |0.8   |0.9   |0.99   |0.999   |0.9999|
> > > > > > > > > > > > |---|---|---|---|---|---|---|---|---|---|---|---|---|---|
> > > > > > > > > > > > |Training Loss| 0.1416$\pm$0.0028| 0.1605$\pm$0.0274| 0.1428$\pm$0.0008|0.1453 $\pm$ 0.0004|0.1391$\pm$0.0011| 0.1421$\pm$0.0013| 0.1387$\pm$0.0002|  0.1457$\pm$0.0016| 0.1417$\pm$0.0005|  0.1419$\pm$0.0019| 0.1551$\pm$0.0018|  0.3497$\pm$0.0298| 0.6645$\pm$0.0227|
> > > > > > > > > > > >
> > > > > > > > > > > > One can observe that in these settings, the performance is similar to that of ResNet 18 on CIFAR10: the changes in performance when $\beta_1$ varies from 0 to 0.9 are relatively small, but there is a dramatic performance drop when $\beta_1$ goes from 0.9 to 0.9999. We will improve these experiments by running more epochs and using more random seeds, and incorporate them into the revised paper to illustrate (part of) the role of $\beta_1$ in real-world tasks.

---

> > > > > > > > ### Comment · Reviewer_b4JJ · 2023-08-17
> > > > > > > > **Reply to Summary of the reviewer's concerns and our response**
> > > > > > > >
> > > > > > > > Thank you for your replies. I have some concerns.
> > > > > > > >
> > > > > > > > Regarding Concern 1:
> > > > > > > > I do not understand that the numerical results for training DCGAN are evidences of the authors’ theoretical results, since optimization problems (Nash equilibrium problems for two-persons (generator and discriminator)) in training GANs are different from ones (empirical/expected risk minimization) in training DNNs. However, I understand TTUR using Adam with $\beta_1 = 0.5$ performs better than with $\beta_1 = 0.9$ for solving Nash equilibrium problems, as promised in the previous papers that considered training GANs.
> > > > > > > >
> > > > > > > > I think that the performance of Adam with $\beta_1 = 0.9$ is almost the same as with $\beta_1 = 0.99$. As AC pointed out, if the setting of $\beta_1 \approx 0$ leads to the finding such that the upper bound becomes small, then it would be nicer. Because the results in the paper present a new setting of $\beta_1 \approx 0$ (please see also my first review comments). If so, numerical results are needed to compare Adam with $\beta_1 \approx 1$ with Adam with $\beta_1 \approx 0$ (proposed method).
> > > > > > > >
> > > > > > > > Reason for Mismatch between Reviewer's Conclusion and Arjevani et al.:
> > > > > > > > I am not satisfied with your comments. I doubt the tightness of the upper bound presented by the authors, since the performance of the upper bound in the paper does not match practical results in the previous papers showing that using $\beta_1 = 0.9, 0.99$ is useful to train DNNs. Moreover, I would like to again emphasize that the performances of deep learning optimizers strongly depend on batch sizes. This is guaranteed from the previous papers showing that the number of iterations decreases when batch size increases in both theory and practice. I do not think that we need to set batch size $b$ depending on $\epsilon$.
> > > > > > > >
> > > > > > > > Therefore, I think that the authors should show that the upper bound presented in the paper matches practical results using $\beta_1 = 0.9, 0.99$. Then, I think that the results obtained from the upper bound are reliable.
> > > > > > > > Or, as seen in my above comments, if the results in the paper present a new setting of $\beta_1$ (e.g., $\beta_1 \approx 0$), then it would be good.

---

> > > > > > > > > ### Author Response · Authors · 2023-08-18
> > > > > > > > > **Response to reviewer**
> > > > > > > > >
> > > > > > > > > Thank you for your response. We appreciate your valuable feedback and have addressed your concerns below.
> > > > > > > > >
> > > > > > > > > **Regarding the numerical results**: If the reviewer does not buy in the DCGAN results, we would like to provide some results on typical classification and language tasks. In (Figure 1, [Choi et al., 2019]) (200+ citations), the authors report that RMSProp (Adam with $\beta_1$=0) can achieve comparable performance as best-tuned Adam over ResNet-32 on CIFAR-10 and Transformer on LM1B.
> > > > > > > > >
> > > > > > > > > **Regarding our experiment**: We respectfully point out that the performance gap between $\beta_1$=0.5 and $\beta_1$=0.9 is much smaller than the gap between $\beta_1$=0.9 and $\beta_1$=0.99. Therefore, if the reviewer believes that "the performance of Adam with $\beta_1$=0.9 is almost the same as with $\beta_1$=0.99", then the performance of Adam with $\beta_1$=0.5 is also almost the same as with $\beta_1$=0.9, which indeed suggests that in this setting, $\beta_1$ need not be close to 1 to make Adam perform well. Furthermore, according to the suggestion of AC, we provide the training loss of ResNet 18 over CIFAR 10 with a full spectrum of $\beta_1$ values. We repeated the experiment three times and report the average values and variance.
> > > > > > > > >
> > > > > > > > > |   $\beta_1$|0   |0.1  |0.2   |0.3   |0.4  |0.5   |0.6   |0.7  |0.8   |0.9   |0.99  |0.999   |0.9999|
> > > > > > > > > |---|---|---|---|---|---|---|---|---|---|---|---|---|---|
> > > > > > > > > |Training Loss| 0.2268$\pm$0.0044| 0.2197$\pm$0.0039| 0.2158$\pm$0.0031| 0.2197$\pm$0.0016| 0.2182$\pm$0.0003| 0.2198$\pm$0.0030| 0.2217$\pm$0.0005| 0.2204$\pm$0.0007| 0.2218$\pm$0.0027| 0.2222$\pm$0.0040| 0.2351$\pm$0.0033| 0.3620$\pm$0.0055| 0.6187$\pm$0.0067|
> > > > > > > > >
> > > > > > > > >
> > > > > > > > > As can be observed, when $\beta_1$ varies from 0 to 0.9, the performance of Adam does not change much. However, when $\beta_1$ varies from 0.9 to 0.9999, the performance of Adam drops significantly. In particular, when $\beta_1$=0.9999, the loss is three times that of $\beta_1$=0.9, indicating that a $\beta_1$ too close to 1 cannot be used in this task.
> > > > > > > > >
> > > > > > > > > Lastly, we fully understand the reviewer's concern that the choice of $\beta_1$ in Adam needs to be cautious, which we totally agree with. Our results do not indicate that $\beta_1$ in Adam can be chosen arbitrarily; rather, they suggest that a wide range of $\beta_1$ values can be used to close the theoretical gap in terms of $\varepsilon$. We believe that the importance of $\beta_1$ might be justified by analyzing other properties of the optimizer or in more specific functions. We are willing to add a paragraph below Theorem 2 in our paper to prevent readers from misunderstanding:
> > > > > > > > >
> > > > > > > > > > "Please note that although our results show that a wide range of values for $\beta_1$ can yield the optimal upper bound, it does not imply that setting $\beta_1$ is not as critical as setting $\beta_2$. The primary objective of this paper is to characterize the dependence on $\beta_1$, and the importance of setting $\beta_1$ might be justified in other ways or characterizations. To help readers gain a deeper understanding of this issue, we provide experiments to illustrate the dependence of performance on $\beta_1$ in practice."
> > > > > > > > >
> > > > > > > > > We sincerely hope that the reviewer can recognize that the goal of our paper is not to obtain a tight dependence on $\beta_1$ but to close the gap between upper bound and lower bound in terms of $\varepsilon$, which we believe has its own merit. Pursuing a tight dependence on $\beta_1$ should be considered as an orthogonal and challenging problem, which we are willing to study as future work.
> > > > > > > > >
> > > > > > > > > Regarding batch size: We agree that batch size is crucial for the optimizer, but this does not contradict the importance of our considered setting. Although we do not consider the variation of batch size, it is a classic and fundamental setting, so we believe studying this setting has its own significance. Therefore, we are puzzled by the reviewer's concern about this part and respectfully ask that is the reviewer concerned that
> > > > > > > > >
> > > > > > > > > * the setting we study is meaningless, or
> > > > > > > > >
> > > > > > > > > * our bound can not meet the lower bound?
> > > > > > > > >
> > > > > > > > > If the latter one, since our Theorem 2 has already shown that we close the gap, we kindly request the reviewer to point out the flaw in our proof.
> > > > > > > > >
> > > > > > > > > Additionally, we kindly point out that although the reviewer stated "I do not think we need to set batch size $b$ depending on $\varepsilon$", in your analysis of SGD, you chose $b=2A_2/\varepsilon^2$.

---

### Official Review · Reviewer_ncBJ · 2023-07-05

**Soundness:** 3 good
**Presentation:** 4 excellent
**Contribution:** 3 good
**Rating:** 7
**Confidence:** 3

**Summary:**

This paper analyzes the iteration complexity of Adam. It Is first pointed out that upper bounds in prior work do not match existing lower bound; the reason is that the lower bound is proved under smoothness and bounded noise variance, while prior upper bounds make more assumptions. This paper then proves a general upper bound (Theorem 1) which only requires smoothness and bounded noise variance and matches the lower bound up to a logarithmic factor. Later in Theorem 2, a refined analysis is given which further removes the logarithmic factor, and thus giving an upper bound that matches the lower bound exactly.

**Strengths:**

This paper analyzes the iteration complexity of Adam, which is a very important problem given Adam's popularity. Moreover, it is pointed out that existing upper bounds do not match lower bound, and this paper closes this gap, which is a nice contribution. The proof techniques may be of independent interest, such as the peeling-off strategy to handle the dependency between the momentum and the adaptive learning rate.

**Weaknesses:**

N/A

**Questions:**

Can SGD match the lower bound? Alternatively, can we show that Adam converges faster than SGD, which is usually observed in practice?

---

> ### Author Rebuttal · Authors · 2023-08-09
>
> We would like to thank the reviewer for the positive feedback. Your concern is dually addressed as follows.
>
> **Q**: Can SGD match the lower bound? Alternatively, can we show that Adam converges faster than SGD, which is usually observed in practice?
>
> **A**: Thanks for asking. SGD can also meet the lower bound according to [1], and thus our results show that Adam can achieve the same order of convergence rate as SGD (which is the first time to our best knowledge). We acknowledge that our result can not be used to show that Adam converges faster than SGD since both meet the lower bound, but conjecture that this is because our analysis is a worst-case analysis. In order to show that Adam converges faster than SGD, we might need to carefully model the structure of neural networks and restrict ourselves to a more specific subset of the objective function space. We leave this as a future work.
>
> **Reference**:
>
> Ghadimi et al., Stochastic First- and Zeroth-order Methods for Nonconvex Stochastic Programming, 2013

---

> > ### Comment · Area_Chair_MZNB · 2023-08-18
> >
> > Thank you for the rebuttal!

---

### Official Review · Reviewer_tjB4 · 2023-07-06

**Soundness:** 3 good
**Presentation:** 3 good
**Contribution:** 3 good
**Rating:** 6
**Confidence:** 4

**Summary:**

Adam is one of the most popular stochastic optimization algorithms especially in deep learning, the existing convergence theories do not achieve a tight upper bound that meets the lower bound. Moreover, many of them require additional assumptions, such as bounded gradient. This paper shows that Adam can achieve the tight bound of $O ( \epsilon^{-4} )$ without such additional assumptions. This result closes the gap between the practical success of Adam and the theoretical sub-optimality.

**Strengths:**

- The theoretical result is strong. Although there are many existing works that analyze the convergence of Adam, this paper is the first one that proves that Adam can converge with $\mathcal{O} ( 1 / \sqrt{T} )$, or equivalently $\mathcal{O} ( \epsilon^{-4} )$.
- Technical improvements for deriving the result are interesting and clearly explained.
- This paper is easy to read, and the presentation is also clear at least to the experts on the convergence theory of stochastic optimization.

**Weaknesses:**

**Relation to [1] is not clear**

Although I am basically positive about the result of this paper, I am not sure about how it relates to the well-known result about the non-convergence behavior of Adam by [1]. They showed that there always exits a problem in which Adam fails to converge, but the authors' result seems that Adam can always converge with the optimal rate (i.e., $\mathcal{O} ( \epsilon^{-4} )$). In my current understanding, this is because Theorem 2 requires $1 - \beta_2 = \Theta ( 1 / T )$, which means that, in order to ensure the convergence of Adam, we need to choose the hyper-parameter $\beta_2$ depending on the total number of parameter updates $T$. I think it would be better to clarify the relationship clearly in Section 7. Though the authors mention [1] in the section, the relation to their theoretical result is not clear to me. When it becomes clear in the rebuttal period, I will raise my score.

**There are no experiments**

- I think the condition of $1 - \beta_2 = \Theta ( 1 / T )$ is crucial to achieve the optimal convergence rate, so it would be better to demonstrate it experimentally (I think a toy experiment is enough).

**Minor comments**

- The notations of the output of Adam are inconsistent in Algorithm 1 ($\boldsymbol{w}_r$) and Theorem 2 ($\boldsymbol{w}_\tau$), which is a little confusing.
- Finishing a paper with Related work is not common in my opinion.
- It would be better to add equation numbers to all the equations for the ease of communication between the reviewers and the authors.

**Typos**

- line 20: uderstand -> understand
- line 198: $\boldsymbol{G_{t-2} \rightarrow G_{t-1}}$
- line 248: pratice -> practice

**References**

[1] Reddi, Sashank J., Satyen Kale, and Sanjiv Kumar. "On the Convergence of Adam and Beyond." International Conference on Learning Representations. 2018.

**Questions:**

**Relation to [1]**

As I mentioned in the weaknesses section, I would like to know the relation between the results of this paper and [1].

**What happens when using diminishing step size?**

In the analysis on SGD, diminishing step size (e.g., $\eta_t = \Theta ( 1 / \sqrt{t} )$ is often used, because it is close to practical situations. Is it easy to extend this result to the case of diminishing step size?

**Limitations:**

The limitations are mentioned in Section 6, which is clear to me.

---

> ### Author Rebuttal · Authors · 2023-08-09
>
> We would like to thank the reviewer for the constructive comment and positive feedback. The raised typos have been dually corrected, and other concerns are dually addressed below.
>
> **Q1**: Relation to [1] is not clear.
>
> **A1**: Thanks for asking. We did not include a discussion on this because there has been a existing work [2] explaining why one can still establish a convergence result of Adam given the counterexample in [1] (**please see Section 4 in [2] for details, and we will include a discussion on this in the revised paper**). The counterexample in [1] is constructed by first choosing the hyperparameters of Adam ($\beta_1$ and $\beta_2$) and then **adversarially** choosing the objective function  for the chosen $\beta_1$ and $\beta_2$. Note that the noise variance $\sigma$ of the chosen objective function has a dependence over the chosen $\beta_1$ and $\beta_2$.  One the contrary, if the objective function is fixed, then one can choose appropriate $\beta_1$ and $\beta_2$ so that Adam converges. This is exactly the setting in our paper (which is also in most optimization papers), where convergence results are provided after the configs of the objective function (including $L$ and $\sigma$) are fixed. Therefore, there is no contradiction between [1] and our result.
>
>
> **Q2**: There are no experiments.
>
> **A2**: Thanks for the suggestion. We will include the following discussion in the revised paper.
>
> 1. We conduct an experiment over the same toy example in (Figure 4, [2]) and plot the result in the pdf of the general rebuttal. Concretely, the objective function is defined as $f(x)=\frac{\sum_{i=0}^9 f_i(x)}{10}$, where $f_0(x)=(x-3)^2$ and $f_i(x)=-0.1(x-\frac{10}{3})^2$ for $ 1\le i\le 9$. In each iteration, we uniformly sample $i$ from $[0,9]$ and use $\nabla f_i$ as the gradient. We compare Adam with different schedulers of $\beta_2$, respectively $\beta_2 = 1- \frac{0.1}{t}$, $\beta_2 = 1- \frac{0.1}{t^{0.5}}$, and $\beta_2 = 1- \frac{0.1}{t^2}$ (where $t$ is the number of iteration), and observe that Adam with  $\beta_2 = 1- \frac{0.1}{t}$ maintains the fastest convergence, which aligns with our observation.
>
> 2. Also, we notice that there are real-world experiments in [3], which show that Adam with $\beta_2=1-\Theta(1/T)$ converges the fastest. We also include their result in the pdf of the general rebuttal.
>
> **Q3**: The notations of the output of Adam are inconsistent in Algorithm 1 and Theorem 2.
>
> **A3**: Thanks for pointing it out. We will revise the paper and use the notation of the output of Adam in Algorithm 1 in both two places.
>
> **Q4**: Finishing a paper with Related work is not common in my opinion.
>
> **A4**: Thanks for pointing it out. We will reorganize the paper and put Related Work after Section 3.
>
> **Q5**: Is it easy to extend this result to the case of diminishing step size?
>
> **A5**: Thank you for the question. Indeed, it is straightforward to extend the result to Adam with a diminishing learning rate by following a similar approach as in Theorem 2 (Theorem 1) and the corresponding rate will only differ from that in Theorem 2 by a $\log T$ factor. This is because, in the proof of Theorem 2, we first analyze the descent lemma for each iteration and subsequently sum them together across iterations. To accommodate a diminishing step size, we can simply replace the learning rate in the analysis of the descent lemma and make minor adjustments when summing them together. We will include this discussion in the revised version of the paper.
>
> **References**:
>
> [1]. Reddi et al., On the Convergence of Adam and Beyond, 2019
>
> [2]. Zhang et al., Adam can converge without any modification on update rules, 2022
>
> [3]. Zou et al., A Sufficient Condition for Convergences of Adam and RMSProp, 2018

---

> > ### Comment · Area_Chair_MZNB · 2023-08-18
> > **Thanks for rebuttal!**
> >
> > Regarding related work, can the authors discuss more about the relationship with [25] in terms of analysis technique?

---

> > > ### Author Response · Authors · 2023-08-19
> > > **Thank you for your suggestion!**
> > >
> > > Dear AC,
> > >
> > > We thank AC for your insightful suggestion. We are more than happy to discuss the technical differences with [25] in more detail in our paper. Specifically, we will add the following paragraph in Section 4.2 - Proof Sketch.
> > >
> > > > Recently, [25] proved that Adam converges without correcting the update. We discuss the technical differences between our work and [25] here.
> > >
> > > >  The most fundamental difference is that the assumptions in [25] are stronger: compared to our paper, [25] additionally assume that the objective function follows an $n$-sum structure, and the stochastic gradients satisfy the $L$-smooth condition (while we only assume this condition holds for full gradients). This makes many of the proof techniques in [25] inapplicable to our paper. Specifically, the proof in [25] can also be roughly divided into addressing Challenge I and Challenge II. We discuss the differences in solving each challenge respectively.
> > >
> > > >1. In addressing Challenge I, [25] and we have the following differences:
> > > > * Different surrogate learning rates: [25] use $\nu_{t-1}$ as a surrogate conditioner to disentangle the stochasticity in momentum and adaptive learning rate, while we use $\tilde{\nu}_{t}^i$.
> > > >* Different proof ideas: [25] conduct a case analysis: if the gradient norm is large, then $\nu_{t}$ and $\nu_{t-1}$ are close, allowing $\nu_t$ to be converted to $\nu_{t-1}$; if the gradient norm is small, the first-order term can be directly bounded by the product of gradient norm and update norm. Note that the claim "if the gradient norm is large, then $\nu_{t}$ and $\nu_{t-1}$ are close" requires the $n$-sum structure and $L$-smooth condition for stochastic gradients, and thus cannot be applied in our paper. Instead, our paper proves that the approximation error introduced by converting $\nu_t$ to our surrogate conditioner $\tilde{\nu}_{t}^i$ is at the "Second Order" term level, so the accumulation of approximation errors can be bounded.
> > > >2. In addressing Challenge II, due to the assumptions of the $n$-sum structure and the $L$-smooth condition for stochastic gradients, [25] directly convert the surrogate conditioner $\nu_{t-1}$ to the gradient norm and directly obtain a bound on the gradient norm. However, we do not have such assumptions, so we first obtain a bound on the sum of $E[\Vert\sqrt[4]{\tilde{\nu}_{t}^{1}} \Vert ]$ from Eq. (4), and then use Cauchy's inequality to obtain the final gradient norm bound.
> > >
> > > Once again, we appreciate your valuable suggestions. If you feel that there are still places in our paper that need improvement, please feel free to point them out!

---

### Official Review · Reviewer_oCro · 2023-07-10

**Soundness:** 2 fair
**Presentation:** 2 fair
**Contribution:** 3 good
**Rating:** 5
**Confidence:** 3

**Summary:**

This paper gives a new analysis of the Adam algorithm intended to close the gap between the upper bound of Adam's iteration complexity and the existing lower bound for first-order nonconvex optimization. The authors show that existing analysis of Adam either uses the bounded gradient assumption, achieves a suboptimal iteration complexity, or relies on the mean-squared smoothness assumption. Afterwards, they give a novel analysis of the algorithm that meets the lower bound in dependence on the desired accuracy $\epsilon$.

**Strengths:**

- The new analysis of Adam is conducted under the same assumptions as standard SGD, unlike most prior work.
- The new analysis of Adam achieves the optimal $\frac{1}{\epsilon^{4}}$ complexity with no additional log factors.
- The authors introduce several technical tools that can be helpful in the analysis of adaptive algorithms more generally, for example the stochastic surrogates used are new.

**Weaknesses:**

- My main problem with this paper is that the proof, as it is, is very complicated to check. It would be very helpful if the authors included a section in the deterministic case (no stochasticity) with their full proof in this simplified setting.
- The bounded variance condition is a bit restrictive (see [1]), can the convergence of Adam be derived under any of the more general conditions mentioned in [1]?
- (Minor typos) line 134 should be log 1/\epsilon not log \epsilon.  Line 198 $G_{t-1}$ not $G_{t-2}$. Please use \left and \right for braces in line 229.

[1] Ahmed Khaled & Peter Richtárik, Better Theory for SGD in the Nonconvex World, TMLR 2023

**Questions:**

1. Please address my concern in the weaknesses section.

**Limitations:**

N/A.

---

> ### Author Rebuttal · Authors · 2023-08-09
>
> We would like to thank the reviewer for your constructive comment and positive feedback. The raised typos have been corrected and below we dually respond to your comments.
>
> **Q1**: Proof in the deterministic case will be helpful.
>
> **A1**: Thanks for the helpful suggestion. We will take your advice and reorganize Section 4.2 by (i). first providing the proof in the deterministic case and then (ii). introducing the additional challenges in the stochastic case and how to solve it. Below we sketch the proof in the deterministic case.
>
> **Stage I.** We no longer need surrogate conditioner $\tilde{\nu_{t}^i}$ to disentangle the correlation between $m_t$ and $\nu_t$ and we only need to handle the mismatch between $m_t$ and $G_t$ in the first-order term $-\eta \langle G_t, \frac{1}{\sqrt{\nu_t}}\odot m_t\rangle$. To handle this, we first apply the definition of $m_t = (1-\beta_1) (G_t+\beta_1 G_{t-1}+ \beta_1^2 G_{t-2}+...) $, and write $$-\eta \langle G_t, \frac{1}{\sqrt{\nu_t}}\odot m_t \rangle= -\eta (1-\beta_1)( \langle G_t, \frac{1}{\sqrt{\nu_t}}\odot G_t\rangle + \beta_1 \langle G_t, \frac{1}{\sqrt{\nu_t}}\odot G_{t-1}\rangle+\beta_1^2\langle G_t, \frac{1}{\sqrt{\nu_t}}\odot G_{t-2}\rangle+...).$$
>
> We proceed by approximating $G_{t-i}$ by $G_t$ in the above equation, and have
>
> $$-\eta \langle G_t, \frac{1}{\sqrt{\nu_t}}\odot m_t \rangle= -\eta (1-\beta_1)( \langle G_t, \frac{1}{\sqrt{\nu_t}}\odot G_t\rangle + \beta_1 \langle G_{t-1}, \frac{1}{\sqrt{\nu_t}}\odot G_{t-1}\rangle+\beta_1^2\langle G_{t-1}, \frac{1}{\sqrt{\nu_t}}\odot G_{t-2}\rangle+...) +\text{Error Term}.$$
>
> In the right-hand-side of the above equation, all terms except the "Error Term" (which is in the same order as the second-order term in the descent lemma) are negative. Applying the above equation to the descent lemma and summing it over iterations gives a bound of $\sum_{t=1}^T\sum_{l=1}^d \frac{\vert G_{t,l} \vert^2}{\sqrt{\nu_{t,l}}}$.
>
> **Stage II.** Based on the observation that $(1-\beta_2)G_{t,l}^2= \nu_{t,l}-\beta_2 \nu_{t-1,l}$, we have $\frac{\vert G_{t,l} \vert^2}{\sqrt{\nu_{t,l}}} = \Omega (\sqrt{\nu_{t,l}}-\sqrt{\beta_2} \sqrt{\nu_{t-1,l}})$, which can transfer the bound of $\sum_{t=1}^T\sum_{l=1}^d \frac{\vert G_{t,l} \vert^2}{\sqrt{\nu_{t,l}}}$ into a bound of $\sum_{t=1}^T\sum_{l=1}^d \sqrt{\nu_{t,l}}$. Due to $$ (\sum_{t=1}^T\sum_{l=1}^d \frac{\vert G_{t,l} \vert^2}{\sqrt{\nu_{t,l}}})(\sum_{t=1}^T\sum_{l=1}^d \sqrt{\nu_{t,l}})\ge (\sum_{t=1}^T \sum_{l=1}^d  \vert G_{t,l} \vert )^2,$$ applying bounds of $\sum_{t=1}^T\sum_{l=1}^d \frac{\vert G_{t,l} \vert^2}{\sqrt{\nu_{t,l}}}$ and $\sum_{t=1}^T\sum_{l=1}^d \sqrt{\nu_{t,l}}$ provides the estimation of gradient norm and concludes the proof.
>
> **Remark**: Compared to the stochastic case, Stage I is much more simplified and no longer needs to disentangle the randomness in $m_t$ and $\nu_t$, and Stage II is also much more simplified when transforming the bound of $\sum_{t=1}^T\sum_{l=1}^d \frac{\vert G_{t,l} \vert^2}{\sqrt{\nu_{t,l}}}$ into a bound of $\sum_{t=1}^T\sum_{l=1}^d \sqrt{\nu_{t,l}}$.
>
> **Q2**: Can the assumption on noise be weakened to more general conditions mentioned in [1]?
>
> **A2**: Thanks for asking. A brief answer is that the proof can be extended to the more general noise assumption "$E \Vert O_f(w,z) -\nabla f(w) \Vert^2 \le \sigma_1^2 \Vert \nabla f(w) \Vert^2+\sigma_0^2$" (called "affine noise variance assumption" in [Faw et al., 2022]), but we are not sure if it can be further generalized under the noise assumption "$E \Vert O_f(w,z) -\nabla f(w) \Vert^2 \le \sigma_2^2 (f(w)-f^*)+\sigma_1^2 \Vert \nabla f(w) \Vert^2+\sigma_0^2$".
>
> Below we sketch how to extend the proof under the affine noise variance assumption (**we will provide detailed proof in the appendix of the revised paper**). The core difference of the proofs under different noise assumptions is the way to estimate the first-order term in Stage I. We leverage an auxiliary function $\xi_t=\sum_{l=1}^d\frac{\vert G_{t,l} \vert^2 }{\sqrt{\tilde{\nu_{t,l}}}}$ to handle the affine variance noise, where $\tilde{\nu_t}$ is defined as $\tilde{\nu_t}=\beta_2\nu_{t-1}+(1-\beta_2) \sigma_0^2$.
>
> For simplicity, we focus on the case where $\beta_1=0$, i.e., no momentum.  To disentangle the stochasticity between $g_t$ and $\nu_t$, we use $\tilde{\nu_t}=\beta_2\nu_{t-1}+(1-\beta_2) \sigma_0^2$ to approximate $\nu_t$ and thus the first-order term can be written as $$E [\langle G_t,-\eta \frac{1}{\sqrt{\nu_t}}\odot g_t \rangle]=-\eta E [\langle G_t, \frac{1}{\sqrt{\tilde{\nu}_t}}\odot G_t \rangle]+E [\langle G_t,\eta (\frac{1}{\sqrt{\tilde{\nu}_t}}-\frac{1}{\sqrt{\nu_t}})\odot g_t \rangle].$$ The first term on the right-hand side is the main term (which is negative) and we need to tackle the first term. By leveraging a similar routine as the proof of Lemma 1, we can arrive at the following estimation of the approximation error.
>
> $$\textbf{Approximation Error} \le  \frac{1}{4}\eta E [\langle G_t, \frac{1}{\sqrt{\tilde{\nu_t}}}\odot G_t \rangle]+O(\text{Second Order Term})+O(\sum_{l=1}^dE(1-\beta_2)\frac{ g_{t,l} ^2 G_{t,l}^2}{(\sqrt{\nu_{t,l}}+\sqrt{\tilde{\nu_{t,l}}})^2\sqrt{\tilde{\nu_{t,l}}}} ).$$
> In the right-hand-side of the above equation, the last term takes close resemblance to $\frac{\xi_{t-1}}{\sqrt{\beta_2}}-\xi_t$ (after expansion), and indeed can be directly bounded by $O(\frac{\xi_{t-1}}{\sqrt{\beta_2}}-\xi_t)$ plus some error term by direct calculation (recall that $\xi_t=\sum_{l=1}^d\frac{\Vert G_{t,l} \Vert^2 }{\sqrt{\tilde{\nu_{t,l}}}}$). The sum of $O(\frac{\xi_{t-1}}{\sqrt{\beta_2}}-\xi_t)$ across t gives $(\frac{1}{\sqrt{\beta_2}}-1)O(\sum_{t=1}^T \xi_t)$, which is small than the sum of the main term in the first-order term $-\eta E [\langle G_t, \frac{1}{\sqrt{\tilde{\nu}_t}}\odot G_t \rangle]$ when $\beta_2$ is close to $1$, and thus we succeed to control the approximation error. The rest of the proof should be the same as under the original noise assumption.

---

> > ### Comment · Area_Chair_MZNB · 2023-08-18
> > **Thank you for providing detailed rebuttal!**
> >
> > We will take this into account!

---

### Author Rebuttal · Authors · 2023-08-09

We thank ACs, SACs, PCs, and reviewers for the efforts and time spent in handling our paper. According to the suggestions of Reviewer oCro and Reviewer b4JJ, we include several experiments to support our theoretical claims. The plots can be found in the attached pdf file. Specifically:

1. We run Adam with different schedulers of $\beta_2$ over a toy example proposed in [1];

2. We adapt a figure from [2], which train ResNet 18 over CIFAR-100 using Adam with different schedulers of $\beta_2$.

All of the experiments indicate that Adam with scheduler $\beta_2=1-\Theta(1/T)$ maintains the fastest convergence, which supports our Theorem 3.

**References**

[1]. Zhang et al., Adam Can Converge Without Any Modification On Update Rules, 2022

[2]. Zou et al., A Sufficient Condition for Convergences of Adam and RMSProp, 2018

---

### Decision · Program_Chairs · 2023-09-21

**Decision:**

Accept (poster)

**Comment:**

The paper tries to develop a proof of Adam for non-convex problems under only two standard assumptions bounded gradient and smoothness of the objective function. It either removes bounded gradient or stochastic gradient assumption or the individual smoothness assumption and manages to prove the standard convergence rate.

There was intensive discussion between the authors and one reviewer about the setting of  $\beta_1$. The result seems to suggest that the smaller $\beta_1$ the better the convergence, which seems to contradict to empirical results and prior works. The authors agree to provide more empirical results to justify the limitations of the theoretical results regarding the value of $\beta_1$. Given the contribution and discussion, AC thinks the paper makes enough contributions to be published at NeurIPS conference. The new proof techniques might inspire new works for analyzing Adam algorithms. The authors are encouraged to add more empirical results about the $\beta_1$ issue, including generalization and training curves.